# Perception Encoder: The best visual embeddings are not at the output of the network

**Daniel Bolya**[1,*]  **Po-Yao Huang**[1,*]  **Peize Sun**[1,*]  **Jang Hyun Cho**[1,2,*,†]  **Andrea Madotto**[1,*]  **Chen Wei**[1]  **Tengyu Ma**[1]  **Jiale Zhi**[1]  **Jathushan Rajasegaran**[1]  **Hanoona Rasheed**[3,†]  **Junke Wang**[4,†]  **Marco Monteiro**[1]  **Hu Xu**[1]  **Shiyu Dong**[1]  **Nikhila Ravi**[1]  **Daniel Li**[1]  **Piotr Dollár**[1]  **Christoph Feichtenhofer**[1]

[1]Meta  [2]UT Austin  [3]MBZUAI  [4]Fudan University
[*]Joint first author  [†]Work done during internships at Meta

## Abstract

We introduce Perception Encoder (PE), a family of state-of-the-art vision encoders for image and video understanding. Traditionally, vision encoders have relied on a variety of pretraining objectives, each excelling at different downstream tasks. Surprisingly, after scaling a carefully tuned image pretraining recipe and refining with a robust video data engine, we find that contrastive vision-language training *alone* can produce strong, general embeddings for all of these downstream tasks. There is only one caveat: *these embeddings are hidden within the intermediate layers of the network*. To draw them out, we introduce two alignment methods: language alignment for multimodal language modeling, and spatial alignment for dense prediction. Together, our PE family of models achieves state-of-the-art results on a wide variety of tasks, including zero-shot image and video classification and retrieval; document, image, and video Q&A; and spatial tasks such as detection, tracking, and depth estimation. We release our models, code, and novel dataset of synthetically and human-annotated videos: https://github.com/facebookresearch/perception_models

## 1  Introduction

For the last decade in computer vision, pretrained vision encoders have been the core building block for most applications requiring *perception*. From million-scale ImageNet [25] pretrained convolutional networks [41, 60, 79, 121, 128] to billion-scale web-pretrained transformers [19, 24, 28, 53, 127, 156], the dominant strategy in vision has been to adapt large-scale pretrained encoders to downstream tasks.

Today, these pretraining objectives come in several flavors: vision-language contrastive losses [103, 158] learn a global vision and language embedding well-suited for zero-shot classification and retrieval as well as provide vision-language alignment for open-world [68, 92] and generative tasks [105, 111]; captioning losses [36, 134] learn to predict image descriptions using a language decoder, which transfers well to downstream multimodal language model (MLLM) tasks; and spatially self-supervised losses [43, 96] learn dense spatial correspondences without language supervision, making them useful for tasks requiring precise localization like object detection. Many works are now attempting to combine two or more of these techniques in different ways [19, 33, 34, 36, 44, 88, 107, 156]. While many have been successful, the complexity of these strategies grows exponentially with number of use cases, which can make scaling difficult. There has not yet been shown a *single, simple, and easily scalable* pretraining technique that can learn state-of-the-art features for all downstream tasks.

In this work, we discover that *global vision-language contrastive learning alone* can be one such approach. We begin by building $PE_{core}$ (Fig. 1, left), a large-scale contrastively pretrained model with state-of-the-art zero-shot performance on *both* image and video (§2). To accomplish this, we first focus on developing a strong *image-only* contrastive pretraining recipe to extract general knowledge from billion-scale image-text data (§2.1). We then use the resulting model as a frame-based encoder to develop a *video* data engine (§2.2) for generating well-aligned video captions. Finetuning on this synthetic video-text data substantially improves performance on *both image and video* classification

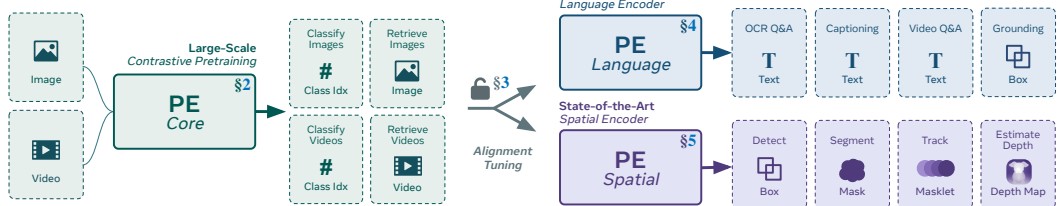

Figure 1: **Perception Encoder (PE)** is a family of large-scale vision encoder models with state-of-the-art performance on a large variety of vision tasks. By using a robust contrastive pretraining recipe and finetuning on synthetically aligned videos, PE not only outperforms all existing models on classification and retrieval (§2), but it also internally produces strong, general features that *scale* for downstream tasks (§3). PE unlocks the ability for large-scale contrastive pretraining to transfer to downstream tasks with alignment tuning to capitalize on those general features (§4, §5).

and retrieval tasks. Finally, we scale our robust image pretraining and well-aligned video finetuning strategy to 2B parameters to produce $PE_{core}G$ (§2.3), a single unified encoder that outperforms SigLIP2 [135] on zero-shot image tasks and InternVideo2 [143] on most zero-shot video tasks.

After analyzing the performance of $PE_{core}G$, we found a surprising result: *inside the model were specific features aligned to OCR, VQA, grounding, detection, depth estimation, and tracking* (§3). Compared to the state-of-the-art models with captioning [36] and spatially self-supervised [96] pretraining, our contrastive encoder has specific layers that, when used as frozen features, matches or exceeds the performance of the other two pretraining techniques *on tasks they should be the best at*. The only problem is—these features exist at *different layers* for each task.

By exploiting this phenomenon with *alignment tuning* (Fig. 1, right), we show it is possible to align these features to the end of the network to create state-of-the-art encoders for downstream MLLM (§4) and spatial (§5) tasks—all following the same easily scalable contrastive pretraining. Thus, Perception Encoder unlocks the potential to scale one simple pretraining method to solve many downstream vision tasks. We will release our models, code, and novel PE Video Dataset of 1M high-quality stock footage videos and 120K human-refined captions.

## 2    Perception Encoder: *Core*

To build Perception Encoder (PE), we start by training a large-scale, robust, and performant vision-language contrastive model for image *and video*. We have two objectives: to enhance the scalability and data efficiency of contrastive training, and to create a unified model for image and video.

We decouple image and video training into two stages. We first develop a strong *image* pretraining recipe (§2.1) with several regularization techniques to create a robust starting point. Then we use the resulting image model as a frame encoder to develop a *video data engine* (§2.2) supported by our novel human-refined video-text dataset to generate aligned captions for video clips. Finally, we train the image encoder on the resulting aligned video data (§2.3). Using our data engine design, this short training step substantially improves *both* image and video performance.

### 2.1    Robust Image Pretraining

In the first stage of pretraining, we want to learn as much visual information as possible from a large set of image-text data with high regularization, stability, and training efficiency in mind.

**Setup.** We track our changes with OpenCLIP [50] ViT-L/14 at 224 resolution as a baseline (Fig. 2.1). We fix a training budget of around 1T GFLOPs (*i.e.*, a ZFLOP), and ablate on a fixed 2.3B noisy image-text dataset curated using the MetaCLIP [150] text-only curation pipeline, and start by training for 12B samples seen. To assess *generality*, we report ImageNet val [25] zero-shot classification results as well as an average of 6 common robustness metrics: ImageNet val [25], ImageNet v2 [109], ObjectNet [4], ImageNet Adversarial [46], ImageNet Rendition [45], and ImageNet Sketch [140].

**Training.** Motivated by [69, 70, 77, 128, 133], we begin by improving training efficiency with *progressive resolution* (Fig. 2.2). By evenly splitting the baseline 12B sample run into 98, 154, and 224 resolution stages (4B per stage), we half training FLOPs while maintaining performance. We then use the extra budget to double global *batch size* (Fig. 2.3) from 32K to 64K, increasing total samples from 12B to 24B. This makes hard negatives more probable, increasing the

"task difficulty" of CLIP. Finally, we switch from AdamW to *LAMB* [154] (Fig. 2.4), which allows us to stably increase learning rate from $5 \times 10^{-4}$ to $2 \times 10^{-3}$ and better fit the CLIP objective. Overall, these changes improve +1.0% on ImageNet val and a similar +1.6% on robustness.

**Modeling.** To assist with scalability [35, 128], we add a *higher resolution* (Fig. 2.5) stage at 336 pixels. To keep FLOPs the same, we adjust the schedule to 10B samples at 98 resolution, 8B at 154, 4B at 224, and 2B at 336. To improve extrapolation, we also add *2D RoPE* [124] (Fig. 2.6) to each attention layer, keeping the original position embedding. Finally, we follow [158] in constructing the CLIP embedding using an *attention pooling* transformer block (Fig. 2.7). Surprisingly, we found keeping the class token as an input to this block is important for small model performance. These changes improve ImageNet val by +1.1% but robustness threefold, by +3.2%.

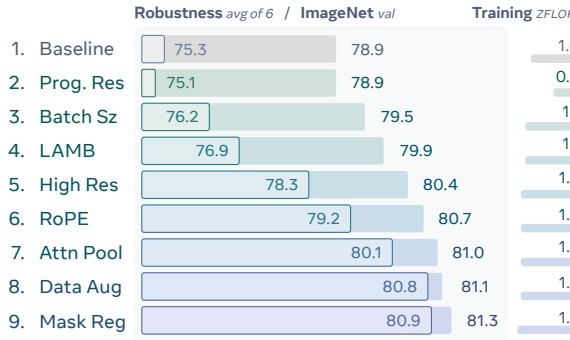

Figure 2: **Robust Image Pretraining.** We tune our pretraining recipe (§2.1) to maximize performance on a fixed set of data, starting with an OpenCLIP [50] ViT-L/14 model. We report cumulative zero-shot classification results for each modification. The inner bars show robustness evaluation, calculated as the average of 6 robustness benchmarks [4, 25, 45, 46, 109, 140], and the outer bars show ImageNet val [25] alone. Several changes significantly improve robustness, indicating that ImageNet val scales more with data, while robustness can scale with refined training techniques.

**Regularization.** Despite training on billions of samples, we find *data augmentation* (Fig. 2.8) still important. Adding heavy random cropping, brightness/saturation jitter, and horizontal flip generally improves robustness without adverse downstream effects (e.g., for OCR). Finally, we add *mask regularization* (Fig. 2.9) by duplicating and masking 1/16th of the input batch. At the output, the masked tokens are aligned to their unmasked counterparts by maximizing cosine similarity. Together, these regularization changes improved ImageNet val by +0.3% and robustness by +0.8%.

Overall, our recipe improves ImageNet val by +2.4% and robustness by a significant +5.6% while keeping FLOPs similar and maintaining or improving scaling behavior (see Appendix C.1).

## 2.2 Bootstrapping a Video Data Engine with Perception Encoder

Our next step is to extend the image-only encoder to video. Unlike web-scale image-text data, which comes in many cases with human-generated descriptive alt-text information, videos with aligned language annotation are inherently scarce and often low quality. Inspired by the recent success of image data engines [57, 63, 94, 108, 149], we address the lack of high quality aligned video captions by developing a robust video data engine to generate them. Our approach (Fig. 3) represents the first large-scale exploration of this kind.

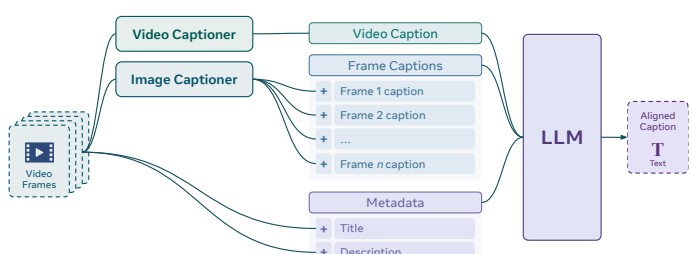

Figure 3: **Video Data Engine.** We use a PE-based video captioner for video-level captions and an existing image captioner [80] on sampled frames. We use these components along with the video metadata to synthesize short captions with a text-only LLM [80].

**Video Data Engine.** We build our data engine in 3 parts: (1) we construct a video captioning model using an early image-only version of PE as a frame-level encoder and Llama [80] as the language decoder. We train with the PLM [21] MLLM training recipe and data mix. In total, the mix consists of 64.7M images and videos covering natural images, charts, documents, exocentric and egocentric videos. (2) to further boost captioning performance, we collect a set of 265K videos (part of which we release as PE Video Dataset, see Appendix A.1), caption them with our base video captioner, and ask human raters to refine the captions. We then finetune our video captioner with this human refined data, significantly improving captioning quality (see Appendix C.2). (3) finally, we synthesize

the aligned video captions by incorporating captions from our video captioner, Llama 3.2 [80] as a per-frame image captioner, and the existing title and description metadata (Fig. 3) summarized with a Llama 3.3 70B text model (see Appendix A.2.4 for prompts).

**Video Training.** We use the resulting data engine to generate information-dense captions for a diverse set of 22M videos, with which we finetune the image-only PE model. To do so, we use PE as an *frame-level encoder*: for each video, we uniformly sample 8 frames, extract the CLIP embedding for each, and average pool to obtain a single video embedding for text embedding alignment. Despite its simplicity, we find this technique produces a strong joint image-video encoder.

**Ablations.** In Tab. 1, we ablate the impact of each component of the video data engine by finetuning an intermediate image-only PE$_{core}$ checkpoint on the recaptioned videos. Compared to the image-only baseline encoder (first row), our video data engine significantly enhances zero-shot classification and retrieval performance for both image (72.6→78.2) and video (50.9→61.6). Notably, using video-level and frame-level captions provides significant improvements over relying solely on metadata such as video title and description (second row), highlighting the importance of building a robust video data engine to compensate for noise in web videos.

| Title | Description | Video Caption | Frame Caption | *Average Image* | ImageNet *val* [25] | ImageNet *v2* [109] | ObjectNet *IN Classes* [4] | MS-COCO *txt→img* [74] | MS-COCO *img→txt* [74] | *Average Video* | Kinetics *400* [54] | Kinetics *600* [54] | MSR-VTT *txt→vid* [151] | MSR-VTT *vid→txt* [151] |
|---|---|---|---|---|---|---|---|---|---|---|---|---|---|---|
| | | | | 72.6 | 83.3 | 77.8 | 85.8 | 49.4 | 66.8 | 50.9 | 69.7 | 68.4 | 38.0 | 27.3 |
| ✓ | ✓ | | | 75.4 | 83.2 | 78.2 | 87.1 | 47.3 | 66.0 | 56.0 | 74.1 | 73.5 | 39.0 | 37.3 |
| ✓ | ✓ | ✓ | | 78.2 | 83.5 | 78.4 | 86.8 | 56.0 | 74.3 | 60.9 | 73.8 | 73.4 | 47.6 | 48.8 |
| ✓ | ✓ | ✓* | ✓ | 78.1 | 83.7 | 79.0 | 87.7 | 54.1 | 73.0 | 60.9 | 75.4 | 75.1 | 46.7 | 46.5 |
| ✓ | ✓ | ✓ | ✓ | 78.2 | 83.7 | 79.0 | 87.5 | 54.6 | 73.2 | 61.6 | 75.8 | 75.5 | 47.4 | 48.1 |

Table 1: **Video Data Engine Ablation.** We ablate our video data engine in Fig. 3 by finetuning on an in-development image-only version of PE by averaging the frame embeddings to create a single video CLIP embedding. Video captions are generated by our captioner trained with or without (✓*) human-refined data. Frame captions are generated by the Llama 3.2 vision model [80]. Taken together, the result is a huge boost to *both* image and video zero-shot performance. See Appendix C.2 for more ablations and scaling behavior.

## 2.3 A Unified Encoder for Image and Video

Using a robust, scalable image pretraining recipe and video-pretraining data recaptioned by the proposed video data engine, in this section we present **PE$_{core}$**, a unified image-and-video encoder.

**Model Architecture**. To capitalize on the promising scaling behavior observed in §2.1, we scale the largest PE$_{core}$ model to 2B parameters (G scale). Tab. 2 shows the detailed model configuration of the vision and text transformers and the dimension of the output clip embedding space.

| Scale | Tower | Params | Width | Depth | MLP | Heads | CLIP Dim |
|---|---|---|---|---|---|---|---|
| B | Vision | 0.09B | 768 | 12 | 3072 | 12 | 1024 |
| | Text | 0.31B | 1024 | 24 | 4096 | 16 | |
| L | Vision | 0.32B | 1024 | 24 | 4096 | 16 | 1024 |
| | Text | 0.31B | 1024 | 24 | 4096 | 16 | |
| G | Vision | 1.88B | 1536 | 50 | 8960 | 16 | 1280 |
| | Text | 0.47B | 1280 | 24 | 5120 | 20 | |

Table 2: **PE Model Configurations.**

**Model Training**. We train PE$_{core}$ in three stages:

1. *Image pretraining.* We scale up image pretraining data to 5.4B publicly available image alt-text pairs curated with MetaCLIP [150] and a total of 86B samples seen to ensure convergence (58B for B and L). We use a global batch size of 131K, with progressive resolution from 98 to up to 448 depending on the model.

2. *Image and video finetuning.* Following the initial pretraining, we subsequently finetune the model at max resolution with a short schedule for 50M samples on the image pretraining data (as cooldown) followed by 22M samples on the recaptioned videos with a smaller learning rate and batch size. The video captions are produced using the proposed video data engine (§2.2). For each video clip, we uniformly sample 8 frames, encode them, take their average to produce a single video embedding, and align them with the corresponding video captions using the same contrastive objective in image training.

3. *Smaller model distillation.* We distill the 2B model (G scale) into smaller contrastive pretrained models at B and L scales under their final resolutions, using a short *finetuning* schedule that covers approximately 4B samples seen (∼8% of the pretraining schedule) with a lower learning rate. We still perform stages 1 and 2 for small models (see Appendix C.3).

Detailed training configurations and setups are listed in Appendix B.1.1.

Table 3 (Zero-Shot Image Results):

| Model | Encoder Params | Resolution | Data | *Avg Class.* | ImageNet *val* [25] | ImageNet *v2* [109] | ObjectNet *IN Classes* [4] | ImageNet *Adversarial* [46] | ImageNet *Renditions* [45] | ImageNet *Sketch* [140] | *Avg Fine.* | Flowers *Oxford* [95] | Cars *Stanford* [58] | Aircrafts *FGVC* [86] | Countries *211* [130] | Scenes *SUN397* [148] | Satellite *RESISC* [20] | *Avg Retrieval* | MS-COCO *txt→img* [74] | MS-COCO *img→txt* [74] | Flickr-30k *txt→img* [155] | Flickr-30k *img→txt* [155] |
|---|---|---|---|---|---|---|---|---|---|---|---|---|---|---|---|---|---|---|---|---|---|---|
| SigLIP-B/16† [158] | 0.1B | 224 | 10B | 69.9 | 76.2 | 69.5 | 70.7 | 45.1 | 90.2 | 67.9 | 61.8 | 85.2 | 90.8 | 44.0 | 15.9 | 70.0 | 64.6 | 69.8 | 47.2 | 64.5 | 77.9 | 89.6 |
| SigLIP2-B/16† [135] | 0.1B | 224 | 10B | 73.1 | 78.2 | 71.4 | 73.6 | 55.0 | 91.7 | 68.9 | 66.2 | 85.7 | 93.4 | 54.8 | 19.2 | 72.7 | 71.1 | 73.7 | 52.1 | 68.9 | 80.7 | 93.0 |
| **PE_core B** | 0.1B | 224 | 5.4B | 73.2 | 78.4 | 71.7 | 71.9 | 62.4 | 88.7 | 66.1 | 68.8 | 86.5 | 92.1 | 57.0 | 30.5 | 74.0 | 72.7 | 74.3 | 50.9 | 71.0 | 80.8 | 94.4 |
| SigLIP-L/16† [158] | 0.3B | 384 | 10B | 80.7 | 82.1 | 75.9 | 80.9 | 76.5 | 95.0 | 73.6 | 67.1 | 89.4 | 94.8 | 53.2 | 24.7 | 72.5 | 67.9 | 74.7 | 52.8 | 70.5 | 82.6 | 92.9 |
| SigLIP2-L/16† [135] | 0.3B | 384 | 10B | 83.3 | 83.1 | 77.4 | 84.4 | 84.3 | 95.7 | 75.5 | 72.5 | 90.0 | 95.8 | 67.0 | 31.6 | 74.8 | 75.5 | 76.7 | 55.3 | 71.4 | 85.0 | 95.2 |
| **PE_core L** | 0.3B | 336 | 5.4B | 83.9 | 83.5 | 77.9 | 84.7 | 89.0 | 95.2 | 73.4 | 74.6 | 87.2 | 93.7 | 67.8 | 45.6 | 77.4 | 75.7 | 78.8 | 57.1 | 75.9 | 85.5 | 96.6 |
| DFN-H+† [32] | 0.6B | 378 | 5B | 81.6 | 84.3 | 78.3 | 79.6 | 79.6 | 93.6 | 73.3 | 75.2 | 91.6 | 96.0 | 72.5 | 37.9 | 77.4 | 75.9 | 75.8 | 55.6 | 71.8 | 82.1 | 93.6 |
| InternVL-C [19] | 5.5B | 224 | 5B | 82.5 | 83.2 | 77.3 | 80.6 | 83.8 | 95.7 | 74.3 | 69.9 | 85.8 | 94.4 | 53.3 | 35.1 | 76.3 | 74.4 | 78.6 | 58.6 | 74.9 | 85.0 | 95.7 |
| EVA 18B [127] | 17.5B | 224 | 2B | 83.6 | 83.8 | 77.9 | 82.2 | 87.3 | 95.7 | 74.7 | 73.1 | 86.0 | 94.9 | 59.7 | 43.1 | 77.7 | 76.9 | 77.5 | 56.2 | 73.6 | 83.3 | 96.7 |
| SigLIP2-g-opt† [135] | 1.1B | 384 | 10B | 86.2 | 85.0 | 79.8 | 88.0 | 90.5 | 96.6 | 77.4 | 75.6 | 91.5 | 95.9 | 73.6 | 40.1 | 76.3 | 75.9 | 78.0 | 56.1 | 72.8 | 86.0 | 95.4 |
| **PE_core G** *(image only)* | 1.9B | 448 | 5.4B | 86.0 | 85.2 | 80.2 | 87.1 | 91.2 | 96.1 | 76.1 | 78.2 | 91.0 | 94.6 | 76.7 | 57.3 | 77.5 | 71.8 | 74.9 | 53.1 | 70.9 | 81.6 | 93.9 |
| **PE_core G** | 1.9B | 448 | 5.4B | 86.6 | 85.4 | 80.2 | 88.2 | 92.6 | 96.5 | 76.5 | 79.4 | 91.4 | 94.7 | 78.2 | 57.6 | 78.5 | 75.8 | 78.9 | 58.1 | 75.4 | 85.7 | 96.2 |

Table 3: **Zero-Shot Image Results.** Image zero-shot performance of PE$_{core}$ compared to the state-of-the-art. Across all model sizes, PE$_{core}$ obtains state-of-the-art results across general classification, retrieval, and finegrained classification. †Re-evaluated: DFN by [127]; SigLIP and SigLIP2 by us with the same benchmark settings if not reported in [135] (see Appendix B.1.2).

Table 4 (Zero-Shot Video Results):

| Model | Encoder Params | Resolution | # Frames | Video Data | *Avg Class.* | Kinetics *400* [54] | Kinetics *600* [54] | Kinetics *700* [54] | UCF *101* [123] | HMDB *51* [61] | *Avg Retrieval* | MSR-VTT *txt→video* [151] | MSR-VTT *video→txt* [151] | MSVD *txt→video* [13] | MSVD *video→txt* [13] | ActivityNet *txt→video* [10] | ActivityNet *video→txt* [10] |
|---|---|---|---|---|---|---|---|---|---|---|---|---|---|---|---|---|---|
| CLIP4CLIP [82] | 0.1B | 224 | 12 | n/a | - | - | - | - | - | - | - | 32.0 | - | 38.5 | - | - | - |
| SigLIP2-B/16† [135] | 0.1B | 224 | 8 | n/a | 57.3 | 58.7 | 55.0 | 48.4 | 82.0 | 42.3 | 39.9 | 38.5 | 30.1 | 49.0 | 67.2 | 28.6 | 25.8 |
| **PE_core B** | 0.1B | 224 | 8 | 22M | 63.9 | 65.6 | 65.1 | 55.8 | 84.6 | 48.2 | 49.9 | 47.6 | 47.3 | 50.4 | 76.7 | 39.0 | 38.4 |
| UMT-L [66] | 0.3B | 224 | 8 | 25M | - | - | - | - | - | - | 47.1 | 40.7 | 37.1 | 49.0 | 74.5 | 41.9 | 39.4 |
| SigLIP2-L/16† [135] | 0.3B | 384 | 8 | n/a | 64.1 | 65.3 | 62.5 | 56.8 | 86.7 | 49.3 | 44.7 | 41.5 | 31.4 | 53.7 | 74.2 | 35.9 | 31.5 |
| **PE_core L** | 0.3B | 336 | 8 | 22M | 71.4 | 73.4 | 72.7 | 63.5 | 87.1 | 58.5 | 54.8 | 50.3 | 50.1 | 57.2 | 82.4 | 46.4 | 42.1 |
| InternVL-C [19] | 5.5B | 224 | 8 | n/a | - | 69.1 | 68.9 | 60.6 | - | - | - | 44.7 | 40.2 | - | - | - | - |
| InternVideo2 [143] | 1.0B | 224 | 8 | 102M | 70.7 | 73.1 | 72.8 | 64.9 | 88.8 | 53.9 | 59.9 | 51.9 | 50.9 | 58.1 | 83.3 | 60.4 | 54.8 |
| SigLIP2-g-opt† [135] | 1.1B | 384 | 8 | n/a | 68.2 | 69.8 | 67.0 | 61.8 | 90.7 | 51.8 | 46.6 | 43.1 | 34.2 | 55.8 | 74.6 | 38.3 | 33.4 |
| **PE_core G** *(image only)* | 1.9B | 448 | 8 | n/a | 70.9 | 73.1 | 72.2 | 64.3 | 89.5 | 55.5 | 47.6 | 44.3 | 35.2 | 54.3 | 73.9 | 41.4 | 36.3 |
| **PE_core G** | 1.9B | 448 | 8 | 22M | 74.8 | 76.9 | 76.1 | 69.1 | 90.7 | 61.1 | 58.7 | 51.2 | 49.9 | 59.7 | 85.4 | 54.7 | 51.2 |

Table 4: **Zero-Shot Video Results.** Video performance of PE$_{core}$ compared to recent video and image encoders. PE$_{core}$ obtains state-of-the-art in video classification and comparable performance on retrieval benchmarks while using only 22M videos. †SigLIP2 evaluated by us (see Appendix B.1.2).

**Zero-Shot Image Results.** In Tab. 3, we present PE$_{core}$'s performance on zero-shot image benchmarks for classification and retrieval *vs.* the strongest open models, including SigLIP2 [135]. PE$_{core}$ outperforms all other contrastive models across the board on all zero-shot tasks, including the highly competitive average of zero-shot ImageNet robustness metrics [4, 25, 45, 46, 109, 140]. This marks a significant achievement, as we are the first to accomplish this in over 3 years without access to Google's internal JFT-3B [28] or WebLI [17] datasets. And *at the same time*, PE$_{core}$ also exceeds the existing state-of-the-art on image-text retrieval and significantly improves on fine-grained classification—the first to simultaneously hold state-of-the-art on all common zero-shot categories.

Notably, this dominant *image* performance is made possible by our video finetuning. Compared to image only, the video finetuned PE$_{core}$G obtains +0.6% general classification, +1.2% fine-grained classification, and a significant +4.0% boost on retrieval. Thus, well-aligned video text data does not just improve video performance—it creates a strictly better model for both videos *and* images.

**Zero-Shot Video Results.** We present video results in Tab. 4. Our base image encoder already outperforms all other image-only encoders on both zero-shot classification and retrieval, including SigLIP2-g-opt. With video finetuning, PE$_{core}$G significantly outperforms even native video models that use full temporal attention on video classification, and it nearly matches the state-of-the-art on video retrieval despite being a simple frame-level encoder. This result underscores the importance of our video data engine, resulting in +3.9% on average zero-shot video classification, and a massive +11.1% on video retrieval. Moreover, PE$_{core}$ does this with fewer videos compared to other video-based approaches like InternVideo2 [143], highlighting the benefits of a joint image-video encoder.

See Appendix C.4 for additional zero-shot and probing results.

# 3 General Features in a Contrastive Disguise

PE$_{core}$ has strong results on zero-shot classification and retrieval, but these are tasks contrastive encoders specialize in. More important is whether or not this strong performance *generalizes* to downstream tasks. To find out, we compare PE$_{core}$G to state-of-the-art models for other pretraining techniques: captioning (AIMv2-3B [29]) and self-supervised learning (DINOv2-g [96]).

**Layerwise Feature Analysis.** We perform frozen feature analysis of each encoder in Fig. 4 for several downstream benchmarks in 3 categories: classification, language modeling, and spatial tasks. For classification, we probe each model using a randomly initialized cross attention transformer block. For language alignment, we learn a projector and finetune a decoder-only LLM (see §4), and for spatial tasks we train with several different decoders (ViTDet [71] Mask-RCNN [42] with Absolute Win [7] for detection, DPT [106] for depth, and zero-shot feature correspondence for tracking [51]). For each experiment, we sweep over the layers of the model as the optimal features are not necessarily the last. In each case, we use an equivalent image size (window size for detection) of $32 \times 32$ tokens. In each plot, we normalize performance by the maximum and minimum performance across models on that task.

**General Features in Disguise.** This analysis reveals several insights. First, as expected, AIMv2 performs well at classification and the best at visual Q&A language tasks. Similarly, DINOv2 performs the well on spatial tasks like detection, depth, and even grounding through an LLM. Then as already established by other works: DINOv2 performs poorly on OCR tasks [131]. But interestingly, its performance *peaks in the middle of the network* and then drops by the end. And so do the others on several tasks

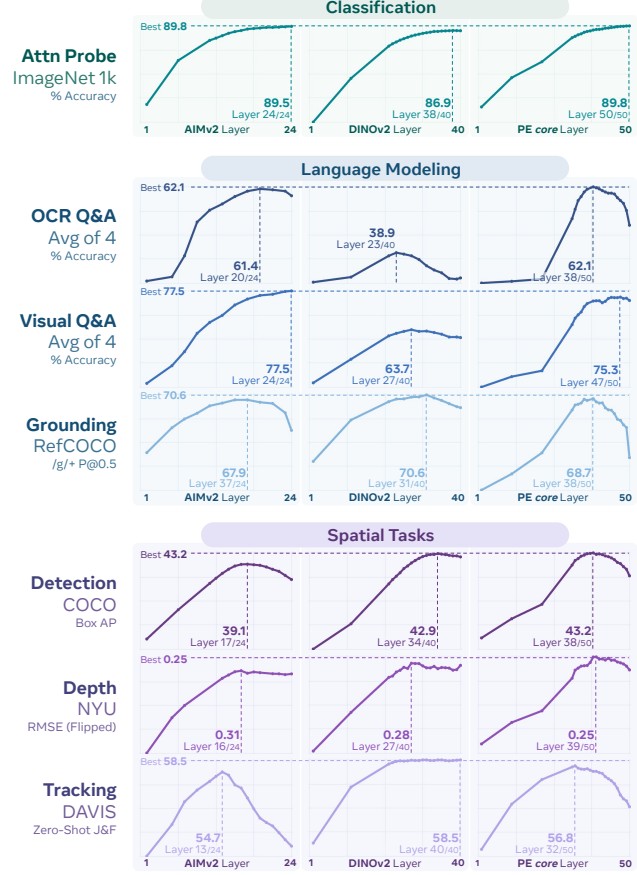

Figure 4: **Frozen Feature Layer Analysis** of different pretraining methods: captioning (AIMv2-3B [36], left), spatially self-supervised (DINOv2-g [96], middle), and our contrastive recipe (PE$_{core}$G, right). Vertical lines denote the best layer and horizontal lines the best performance across models. AIMv2 excels at language but not spatial, and DINOv2 excels at spatial but not language. But, *intermediate layers* of PE$_{core}$G perform well on *both* language and spatial tasks.

(AIMv2: tracking, grounding, detection; DINOv2: VQ&A, grounding). PE$_{core}$ exhibits similar behavior, but with unexpected results: *it can perform well on all tasks, often matching or exceeding the leading models*. Remarkably, PE has intermediate layers that perform near to or on par with AIMv2 for language tasks and DINOv2 for spatial tasks, despite being trained with a global contrastive loss. Depth estimation is particularly noteworthy, as contrastive encoders are not typically considered state-of-the-art in that area. In fact, CLIP models are notorious for poor spatial performance [107].

**An Alignment Problem.** However, PE$_{core}$'s strong general performance diminishes rapidly towards the end of the network, such as for LLM-based grounding. This behavior is less pronounced the closer the downstream task is to the pretraining method, suggesting an *alignment problem*. Thus, a well-tuned large-scale contrastive model can learn general embeddings in the process of fitting its objective, *but it fails to output them*. We address this issue with alignment tuning in §4 and §5 and analyze why our CLIP model has these general features and its scaling behavior in Appendix C.5.

**Analysis.** The finding that pure CLIP models possess features which match the performance of state-of-the-art pretraining methods in their specialized domains is new. In fact, recent work [30] has shown the opposite—that CLIP models fail to scale on downstream tasks. We next investigate how our approach yields these results.

To start, we perform layerwise frozen feature analysis on COCO detection. PE$_{core}$ was particularly "peaky" on this task in Fig. 4, with its best layer on par with DINOv2, but last layer significantly worse. We already ablate each change we made from vanilla CLIP in Fig. 2 using a ViT-L/14 model. So to retrace our steps, we run frozen feature analysis on those checkpoints. For efficiency, we use a lower resolution and only sample even layers for this experiment. In Fig. 5, we report COCO box mAP for the last and best layers for each cumulative ablation, along with the index of the best layer. Further, we plot the layerwise mAP for each change in Fig. 6.

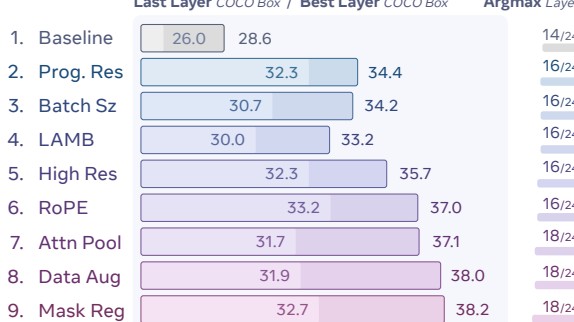

Figure 5: **The Downstream Effects of Robust Pretraining.** The ViT-L/14 checkpoints from Fig. 2 evaluated as frozen features on COCO [74] using Mask R-CNN [42]. We report the last layer performance, best layer performance, and the best layer's index.

Surprisingly, the simple changes we made to CLIP pretraining in §2.1 overall improved the best layer's performance by *almost 10 mAP*! Some improvements are expected like with high resolution (5) and RoPE (6), but unexpectedly data augmentation (8) and *especially* progressive resolution (2) help considerably. It is possible that contrastive pretraining overfits to a specific resolution through "global tokens" [23], thus changing the resolution during training forces the model to be more robust.

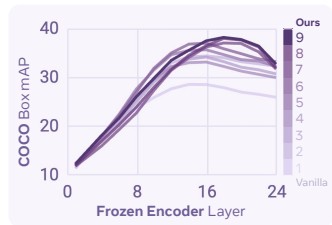

Figure 6: **Layer Analysis** of Fig. 5.

Next, both progressive resolution (2) and attention pooling (7) move the argmax layer deeper into the network (rightmost column of Fig. 5). Attention pooling in particular alters the whole shape of the layerwise performance curve (Fig. 6). Finally, some changes *reduced* performance: increasing the batch size (3) and using LAMB with a high learning rate (4). Both help fit the CLIP loss better, which after a point may not improve the general features. Moreover, while the best layer improved significantly, the last layer performance stagnated after (2). This suggests that constructing the CLIP token requires a specialized decoder. Yet, this does not prevent the model from learning general features—just outputting them.

**Scaling Behavior.** Evidently, our robust recipe can enable contrastive pretraining to produce general features. But, does it scale? In Fig. 7, we answer this by performing frozen feature analysis across S/14, B/14, and L/14 models trained with the same schedule with either the vanilla CLIP recipe or our recipe (see Fig. 14). Immediately, we see a stark contrast between their scaling behaviors: while the vanilla recipe quickly plateaus at L scale (300M), the best layer of our robust pretraining recipe demonstrates scaling to G scale (2B)—despite being trained with a decidedly non-spatially aligned global contrastive loss. Though note this is the *best* layer. The *last* layer still stagnates for both. Thus, CLIP loss obfuscates its general features even with our recipe, placing them several layers deep.

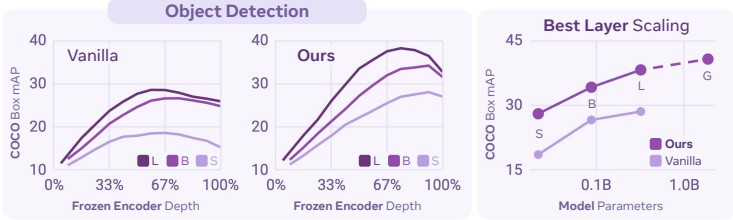

Figure 7: **The Downstream Scalability of Robust Pretraining.** Left: frozen feature layer analysis of the S/14, B/14, and L/14 models from Fig. 14 using the same setup as Fig. 5. Right: scaling behavior of the *best layer* for each model. Note: G has a different schedule. See Appendix C.5 for more.

# 4 Perception Encoder: *Language Alignment*

In §3 we have seen that $PE_{core}$ already possesses useful features for Multimodal Large Language Models (MLLMs), but those features are not aligned to the end of the network. In this section, we *lift* these features through *alignment tuning* to construct a new, MLLM-specialized encoder: $PE_{lang}$.

**Alignment Method.** Aligning a vision encoder to an LLM is relatively straightforward. We follow the approaches of [18, 21, 36], where the vision encoder is *unfrozen* and finetuned as part of an MLLM. In our case, we align $PE_{core}$ to a pretrained Llama3.2 3B text-only decoder with both the encoder and decoder unfrozen, connected with a 2-layer MLP. We discard the last 3 layers of $PE_{core}$, as suggested by [18] and regularize the encoder with LayerScale [132] and DropPath [49]. We train with next token prediction on 70M total samples across OCR Q&A, Captioning, Visual Q&A, and Video Q&A (following [21]), and finally extract the vision encoder only as $PE_{lang}$. More training details are available in Appendix B.2 and ablations of this recipe are conducted in Appendix D.1.

**Effects.** In Fig. 8, we conduct the same layerwise analysis in §3 on the resulting $PE_{lang}G$ compared to $PE_{core}G$. Across all categories, the best layer for the aligned model is the last, no matter the performance of the original checkpoint. Notably, our $PE_{lang}$ training mix did *not* contain grounding data, which means that this significantly lifted grounding performance is entirely due to the strong intermediate grounding features in $PE_{core}$ now being aligned to the end of the network. Moreover, specific domains such as OCR Q&A that *were* represented in the training mix see a significant boost to performance compared to even the best layer of $PE_{core}$, which was already strong. Thus, with an order of magnitude fewer samples compared to pretraining, we were able to *language align* $PE_{core}G$ to create a single, strong encoder for all MLLM tasks.

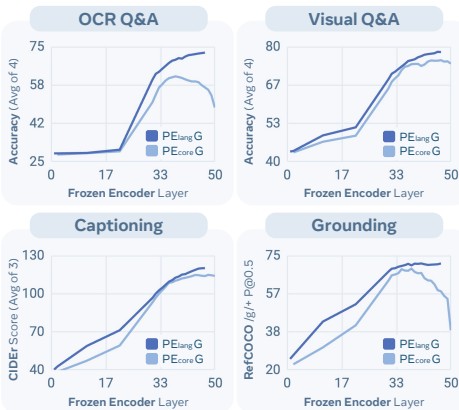

Figure 8: **Language Alignment** lifts the strong performance of $PE_{core}$ (§3) to the end.

**Results.** In Tab. 5, we compare $PE_{lang}$ to existing encoders with good language alignment. To benchmark, we plug each model into a fresh 2-layer MLP and Llama 3.1 8B decoder. The encoder is frozen the rest are are finetuned on 2.6M visual Q&A pairs (see Appendix B.2). We evaluate each encoder at native resolution unless otherwise noted. Despite using a different LLM than during alignment, $PE_{lang}$ significantly outperforms all other models across all scales, resolutions, and tasks. Results with tiling and different LLM decoders are available in Appendix D.4. In all cases, $PE_{lang}$ exhibits *generality*. That is, it outperforms other models no matter the resolution, decoder, or task.

| | OCR / Chart / Doc. Q&A | | | | | Visual Q&A | | | | | Captioning | | | | | | Video | | | | | |
| Model | Avg. OCR QA | ChartQA Acc. [162] | DocVQA Acc. [89] | Info. QA Acc. [90] | AI2D Acc. [56] | Avg. VQA | TextVQA Acc. [122] | OK-VQA Acc. [115] | POPE Acc. [72] | VQAv2 Acc. [39] | Avg. Cap. | Flicker CIDEr [155] | COCO CIDEr [74] | No Cap CIDEr [1] | Avg. Ground. RefCOCO/g/+ [55] | Avg. Video | VideoMME Acc. [37] | STAR Acc. [145] | TGIF-QA Acc. [52] | EgoSchema Acc. [87] | MVBench Acc. [67] | PerceptionTest Acc. [102] |
|---|---|---|---|---|---|---|---|---|---|---|---|---|---|---|---|---|---|---|---|---|---|---|
| *576 Tokens per Image* | | | | | | | | | | | | | | | | | | | | | | |
| CLIP-L [103] | 53.5 | 61.7 | 49.5 | 32.8 | 70.1 | 72.7 | 60.7 | 63.9 | 87.3 | 78.9 | 113.3 | 92.0 | 132.9 | 115.0 | 65.0 | 54.2 | 46.3 | 52.1 | 68.6 | 57.4 | 48.5 | 52.3 |
| AIMv2-L Distill [36] | 53.7 | 61.1 | 49.4 | 31.5 | 72.7 | 74.1 | 62.8 | 64.8 | 88.3 | 80.3 | 117.8 | 94.7 | 137.5 | 121.2 | 62.6 | 53.8 | 44.3 | 52.4 | 65.0 | 57.4 | 50.0 | 53.6 |
| SigLIP2-so400M [135] | 58.9 | 69.0 | 58.3 | 35.2 | 73.1 | 76.8 | 69.8 | **67.2** | 88.7 | 81.6 | 116.5 | 92.1 | 137.7 | 119.8 | 67.4 | 54.5 | 45.5 | 53.1 | 67.2 | 57.6 | 49.3 | 54.5 |
| SigLIP2-g-opt [135] | 56.2 | 63.1 | 55.3 | 34.0 | 72.4 | 77.0 | 70.3 | 66.7 | 89.6 | 81.6 | 117.7 | 94.9 | 137.8 | 120.3 | 66.5 | 53.9 | 46.2 | 53.9 | 66.6 | 53.8 | 48.5 | 54.7 |
| **$PE_{lang}$ G**† | 66.9 | 76.8 | 73.6 | 41.1 | 76.1 | 76.2 | 68.5 | 66.0 | 89.1 | 81.3 | 119.7 | 96.1 | 139.6 | 123.4 | 68.9 | 58.1 | 48.7 | 58.9 | 70.5 | 61.8 | 52.7 | 55.9 |
| *1024 Tokens per Image* | | | | | | | | | | | | | | | | | | | | | | |
| InternViT2.5-L [18] | 60.6 | 74.1 | 59.2 | 35.9 | 73.1 | 74.2 | 65.4 | 64.4 | 87.6 | 79.6 | 112.3 | 88.4 | 133.7 | 114.9 | 66.9 | 50.6 | 45.2 | 44.8 | 62.7 | 54.2 | 46.0 | 50.5 |
| SigLIP2-so400M [135] | 63.3 | 72.1 | 69.3 | 39.0 | 72.7 | 77.9 | 74.8 | 66.0 | 89.0 | **81.8** | 117.4 | 93.5 | 138.3 | 120.2 | 69.6 | 55.8 | 46.2 | 55.4 | 67.0 | **62.0** | 50.0 | 54.5 |
| **$PE_{core}$ L** | 59.4 | 68.7 | 62.5 | 36.6 | 69.7 | 74.7 | 67.7 | 64.3 | 88.3 | 78.7 | 112.7 | 89.6 | 133.4 | 114.9 | 59.7 | 50.9 | 41.7 | 51.2 | 61.6 | 52.6 | 47.4 | 50.6 |
| **$PE_{lang}$ L** | 71.1 | 81.0 | 81.9 | 46.4 | 75.0 | 77.1 | 73.0 | 65.5 | 89.3 | 80.8 | 117.3 | 94.3 | 137.3 | 120.1 | 70.5 | 56.5 | 47.0 | 57.2 | 68.0 | 59.8 | 52.3 | 54.7 |
| DINOv2-g [96] | 30.0 | 19.6 | 14.7 | 24.2 | 61.5 | 61.0 | 19.3 | 60.4 | 88.6 | 75.8 | 109.4 | 86.5 | 131.6 | 110.1 | 64.9 | 49.5 | 39.7 | 52.1 | 60.1 | 46.8 | 47.4 | 50.8 |
| AIMv2-3B [36] | 48.9 | 40.5 | 53.9 | 33.9 | 67.2 | 73.0 | 64.1 | 64.0 | 85.2 | 78.9 | 115.7 | 93.8 | 135.2 | 118.1 | 36.1 | 54.6 | 45.1 | 54.5 | 66.7 | 55.4 | 51.7 | 54.3 |
| InternViT2.5-6B [18] | 59.9 | 72.3 | 59.4 | 35.2 | 72.5 | 75.5 | 68.9 | 64.9 | 88.2 | 80.2 | 115.0 | 92.2 | 136.3 | 116.3 | 68.0 | 49.6 | 44.5 | 47.0 | 62.6 | 45.8 | 48.9 | 48.5 |
| **$PE_{core}$ G** | 60.8 | 69.9 | 65.4 | 36.7 | 71.1 | 73.3 | 65.9 | 60.7 | 88.4 | 78.0 | 112.5 | 91.6 | 133.6 | 112.4 | 66.6 | 52.0 | 42.3 | 53.1 | 62.9 | 51.4 | 48.8 | 53.6 |
| **$PE_{lang}$ G** | **72.4** | **80.5** | **84.4** | **48.3** | **76.4** | **78.1** | **75.2** | 65.4 | **90.1** | 81.8 | 120.1 | **96.6** | 140.0 | 123.6 | **71.3** | **58.0** | 48.0 | 60.1 | 69.4 | 62.0 | 52.4 | 56.0 |

Table 5: **MLLM Results.** We benchmark $PE_{lang}$ *vs.* other frozen vision encoders with Llama 3.1-instruct 8B [80] as the LLM. $PE_{lang}$ shows strong performance across all benchmarks, outperforming much larger models. †Interpolated without extra training. See Appendix D.4 for more results.

# 5 Perception Encoder: *Spatial Alignment*

Unlike for language alignment with an MLLM, the best way to spatially align a model is not obvious. However, the path becomes clear when we study an apparent dichotomy in §3 for $PE_{core}$: higher level spatial tasks like detection and depth estimation perform optimally around layer 40, while low level tasks like tracking perform the best at around layer 30. Upon analyzing the features directly (see Appendix E.1), we find that *locality* begins to deteriorate starting at layer 33 due to global tokens [23].

**Alignment Method.** Following these insights, we design our spatial alignment method with two goals in mind: (1) keep the high level features around layer 40 in tact while (2) improving the locality of the features for lower level tasks. To address (1), we simply finetune $PE_{core}$ using *its own frozen layer 41 features* as a teacher with heavy regularization (DropPath [49], LayerScale [132], 75% masking [144]). Then, we enforce spatial correspondence for (2) using SAM 2.1 [108] *mask logits*. That is, unlike [44, 107, 116], we do not directly use SAM features but instead sample $32{\times}32$ points in a grid and concatenate the SAM 2.1 mask logit for each into a single feature map. As shown in Appendix Fig. 19, this provides features with strong locality. See Appendix B.3.1 for training details.

**Effects.** In Fig. 9, we compare layerwise performance of the original $PE_{core}G$ checkpoint compared to aligning to the teachers described above. We denote aligning to *both* teachers as $PE_{spatial}G$. Aligning to $PE_{core}G$ layer 41 alone performs generally well on all tasks, but has lackluster performance on tracking, where percise locality is necessary to define boundaries. In contrast, aligning to SAM 2.1 mask logits lowers last layer performance on every task *but* tracking. Thus, the optimal approach is to combine both teachers. As a result, $PE_{spatial}G$ not only lifts the features for all tasks to the end of the network, but it also improves over self-alignment, especially on tracking and semantic segmentation. Notably, $PE_{spatial}G$'s tracking performance is lower than the SAM-aligned model, but it is still ahead of other methods while being generally good, see results.

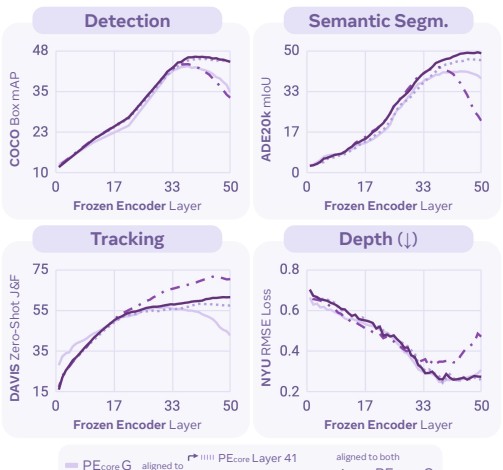

Figure 9: **Spatial Alignment** of $PE_{core}G$.

**Last Layer Visualization.** In Fig. 10, we visualize last layer features for $PE_{core}G$ and the 3 aligned models, with similar colors denoting similar features. In the first column, we see why the last layer performance of $PE_{core}$ is so poor: while it contains information about the salient objects, it seems to have lost spatial coherence. Aligning to the model's own layer 41 features fixes this, but its spatial quality is lacking. In contrast, the model aligned to SAM 2.1 mask logits has great locality, but no semantics (e.g., low similarity between cats in row 1 and cows in row 2). $PE_{spatial}$ retains the semantics of $PE_{core}$ while producing high quality spatial features.

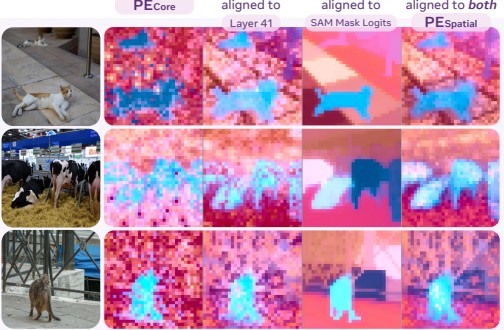

Figure 10: **Last Layer Visualization** using PCA (see Appendix B.3.2). More in Appendix E.4.

| Encoder | Tracking DAVIS (↑) [101] | | | Segmentation ADE20k (↑) [164] | | | Depth NYU (↓) [120] | | |
|---|---|---|---|---|---|---|---|---|---|
| | *Best* | *Last* | *Idx* | *Best* | *Last* | *Idx* | *Best* | *Last* | *Idx* |
| SigLIP-so400M [158] | 48.7 | 36.3 | 16/27 | 40.1 | 38.3 | 22/27 | .339 | .369 | 21/27 |
| SigLIP2-so400M [135] | 51.4 | 45.3 | 15/27 | 44.0 | 42.9 | 24/27 | .306 | .329 | 25/27 |
| DINOv2-L [96] | 58.7 | 58.2 | 23/24 | 47.3 | 47.3 | 24/24 | .297 | .308 | 23/24 |
| DINOv2-g [96] | 58.5 | 58.5 | 40/40 | 48.7 | 48.4 | 37/40 | .279 | .290 | 27/40 |
| **$PE_{core}G$** | 56.8 | 42.8 | 32/50 | 41.5 | 38.6 | 44/50 | **.249** | .309 | 39/50 |
| **$PE_{spatial}G$** | **61.5** | **61.5** | 50/50 | **49.3** | **48.9** | 49/50 | .262 | **.275** | 46/50 |

Table 6: **Frozen Dense Prediction** for the best and last layers of each model. Details in Appendix B.3.3.

| Encoder | LVIS [40] $AP_{box}$ | $AP_{mask}$ | COCO [74] $AP_{box}$ | $AP_{mask}$ |
|---|---|---|---|---|
| MetaCLIP-G [150] | 45.1 | 41.9 | 53.2 | 46.7 |
| SigLIP2-so400M [135] | 49.3 | 45.6 | 56.0 | 49.4 |
| SigLIP2-g-opt [135] | 52.9 | 48.5 | 57.1 | 50.2 |
| DINOv2-L [96] | 46.7 | 43.5 | 55.7 | 49.0 |
| DINOv2-g [96] | 51.5 | 47.3 | 57.2 | 50.0 |
| **$PE_{core}G$** | 51.9 | 47.9 | 57.0 | 49.8 |
| **$PE_{spatial}G$** | **54.2** | **49.3** | **57.8** | **50.3** |

Table 7: **End-to-End Detection** using Mask R-CNN [42]. Details in Appendix B.3.4.

**Results.** In Tab. 6, we compare performance on dense tasks with a frozen encoder with a fixed 448 resolution, reporting both best layer performance and last layer performance. Across the board, $PE_{spatial}G$ outperforms other state-of-the-art models, with its features well aligned to the last layer. In Tab. 7, the same is true when end-to-end finetuning for detection on both LVIS [40] and COCO [74]

| Encoder | Params | Detector | COCO AP$_{box}$ |
|---|---|---|---|
| SwinV2-G [78] | 3.0B | HTC++ [14] | 62.5 |
| Swin-L [77] | 0.3B | DINO [159] | 63.2 |
| InternImage-G [142] | 3.0B | DINO [159] | 65.3 |
| EVA02-L [34] | 0.3B | CoDETR [165] | 65.9 |
| **PE$_{spatial}$G** | 1.9B | DETA [97] | **66.0** |

Table 8: **SOTA Setting Detection** on COCO val. Recipe in Appendix B.3.5.

with a fixed 1024 resolution using Mask-RCNN [42] and ViTDet [71]. Finally, in Tab. 8, we provide a system-level comparison *vs.* the absolute state-of-the-art on COCO val2017. With only Object365 [117] as extra detection data, $PE_{spatial}G$ can match the performance of more complex models tuned for detection, while only using a simple DETR-style decoder [11, 97]. $PE_{spatial}G$ marks the first general, contrastively pretrained model to accomplish this.

# 6 Related Work

Vision-language pretrained models have served as foundation for zero-shot image classification and image-text retrieval [50, 103, 114], open-vocabulary detection [62, 92, 93] and segmentation [22, 27], and multimodal large language models (MLLMs) [3, 5, 76, 91, 98, 131]. PE iterates on this paradigm.

**Contrastive Language-Image Pretraining.** The early works of Virtex [26], ICMLM [112], and ConVIRT [161] developed the techniques for learning through contrastive objectives between vision and language modalities. Subsequently, vision encoders such as CLIP [50, 103] and ALIGN [53] scaled these techniques to much larger datasets and model sizes, popularizing vision-language contrastive learning. A series of open-weight contrastive models have been developed to enhance the performance and robustness of CLIP [32, 70, 114, 126, 150, 158]. PE is among this effort.

**Existing Techniques.** Various techniques used in this work have been explored before. BASIC [99] and LAION [114] explored scaling the batch size up to 160K, and shows the benefits of large batch sizes during training. EVA-CLIP [127] uses LAMB optimizer [154] for large batch training of clip models. Rotary positional embedding (RoPE) [124] has been successfully adopted in large language models. In vision transformers [2, 47] adopted 2D rotary positional embeddings. For data engine, a series of works focus on large-scale sourcing and filtering through efficient data curation [32, 38, 114, 150] and explore recaptioning training images using MLLMs or VLMs [31, 63, 94, 149]. We extend these concepts to create a robust training recipe and to extend data engines to video.

**Intermediate Layers Are Better.** Most vision encoders rely on the last layer to extract features. However, when trained on proxy or self-supervised tasks, the last layer is often not the ideal candidate for other tasks [8, 15, 16, 29, 83, 104, 118, 125, 139, 157, 163]. This has been shown for image coloration [160, 163], next token prediction [15, 29, 104], image generation [83, 157], and to a limited extent in CLIP models [125]. In contrast to these works, we first show the same behaviors across multiple classes of models simultaneously. Then we study this behavior for PE specifically in depth, and show it is possible for CLIP training to produce rich spatial and language features in intermediate layers *on par with the best existing models for each*. Finally, we show how to align these features with short finetuning steps to obtain state-of-the-art on a wide variety of tasks. Unlike other alignment [3, 18, 19, 65, 80, 129, 141] and feature combination [44, 107, 116, 157] methods, our main goal is not to instill a large amount of new knowledge into the model, but instead to bring out and refine the latent strong general features that already exist in the original PE model.

# 7 Conclusion

In this work, we have presented Perception Encoders (PE), a family of best-in-class foundation models comprising $PE_{core}$, $PE_{lang}$, and $PE_{spatial}$. We have shown that $PE_{core}$ can outperform the leading models in zero-shot image recognition, while also excelling in zero-shot video recognition. We have demonstrated that $PE_{lang}$ outperforms the best vision encoders for use in multimodal large language models, often by a large margin. We have established that $PE_{spatial}$ outperforms the long-standing state-of-the-art in object detection with a simpler decoder. Throughout all of this, one conclusion is abundantly clear: Perception Encoder unlocks the potential to scale simple contrastive vision-language pretraining to address a wide range of downstream vision tasks.

**Additional Contributors and Acknowledgments.** We would like to thank Abhimanyu Dubey, Adel Ahmadyan, Andrew Westbury, Arkabandhu Chowdhury, Azita Shokrpour, Babak Damavandi, Chay Ryali, Ching-Feng Yeh, Cyprien de Lichy, Didac Suris Coll-Vinent, Dong Wang, Filip Radenovic, George Orlin, Han Zou, Harry Tran, Jitendra Malik, Joelle Pineau, Joseph Greer, Kavya Srinet, Kirmani Ahmed, Laura Gustafson, Lu Zhang, Muhammad Maaz, Natalia Neverova, Nicolas Carion, Oleksandr Maksymets, Ramya Raghavendra, Romy Luo, Ronghang Hu, Sam Doud, Sasha Mitts, Sean Bell, Shane Moon, Shuming Hu, Soerian Lieve, Stephane Kasriel, Valentin Gabeur, Vanessa Stark, Vignesh Ramanathan, Vivian Lee, Xuan Hu, Yang Li, and Ziyang Wang for their contributions and support for the project. And we thank you, the reader, for reading this far.

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

# Appendix

## Table of Contents

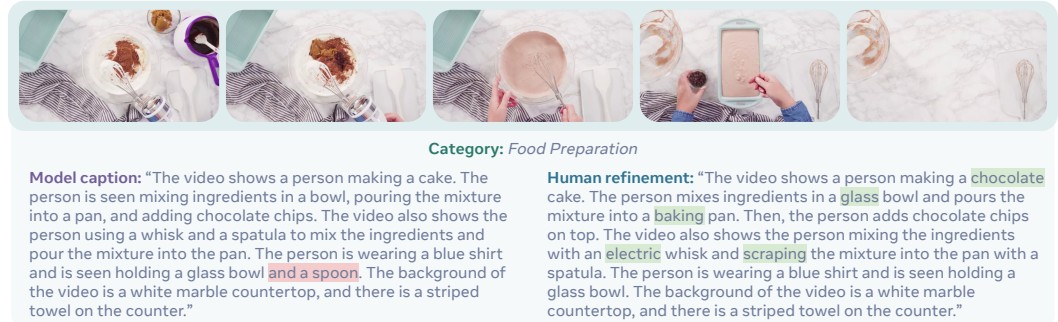

Category: *Food Preparation*

**Model caption:** "The video shows a person making a cake. The person is seen mixing ingredients in a bowl, pouring the mixture into a pan, and adding chocolate chips. The video also shows the person using a whisk and a spatula to mix the ingredients and pour the mixture into the pan. The person is wearing a blue shirt and is seen holding a glass bowl and a spoon. The background of the video is a white marble countertop, and there is a striped towel on the counter."

**Human refinement:** "The video shows a person making a chocolate cake. The person mixes ingredients in a glass bowl and pours the mixture into a baking pan. Then, the person adds chocolate chips on top. The video also shows the person mixing the ingredients with an electric whisk and scraping the mixture into the pan with a spatula. The person is wearing a blue shirt and is seen holding a glass bowl. The background of the video is a white marble countertop, and there is a striped towel on the counter."

Figure 11: **PE Video Dataset Example.** A sample from PVD, our released video-text dataset. Initial captions are generated by our video captioning model and then refined by human annotators. Annotators are instructed to add details and remove model hallucination. In this example, the model hallucination "a spoon" is removed; and more details such as "glass bowl" and the action "scraping" are added. See Fig. 12 for more. Data at `https://ai.meta.com/datasets/pe-video/`

# A  Video Data Engine

## A.1  PE Video Dataset (PVD)

For the benefit of the community, we release a new video dataset: PE Video Dataset (PVD). PVD comprises of 1M high-quality and diverse videos with accompanying tags and descriptions. The videos are motion-centered, covering both first-person and third-person views with a wide coverage of scenes.

We additionally select 120K of these videos with the highest degree of motion to annotate with detailed captions by generating synthetic captions using our video captioner (§2.2) and employing 200 annotators to verify and refine them. We ask the human annotators to improve the synthetic captions by removing any hallucinations, correcting words that describe the video inaccurately, eliminating repetitive or redundant words to make the caption more concise, and adding any missing actions being performed in the video.

We release two versions of annotations for the 120K PVD subset: (1) Human verified captions: extended summaries with an average length of 57.1 words that provide a high-level description of each video. These captions are suitable for CLIP-style training. (2) Long automated captions: detailed and fine-grained descriptions with an average length of 111.7 words that capture spatial and temporal events. These captions are ideal for fine-grained video understanding.

| | |
|---|---|
| Videos | 998,862 |
| Human Captions | 118,862 |
| Total Duration | 4625 hrs |
| Duration (s) | 16.7±9.8 |
| Human Caption Length | 57.1±25.4 |
| Model Caption Length | 111.7±43.2 |

Table 9: **PVD Stats.**

In Fig. 11, we visualize a video example together with their model and human captions from PE Video Dataset (See Fig. 12 for more). The dataset statistics are summarized in Tab. 9. Finally, We use 105K of these refined samples to improve the data engine (§2.2 phase 2) and 15K as a high-quality video retrieval benchmark.

**PVD Benchmark.** We use 15K of the human-refined video-caption pairs as a held-out test set, which we introduce as a new video retrieval benchmark, PVD Benchmark, to evaluate finegrained video-caption alignment. We follow the format of MSR-VTT [151] to construct the benchmark. We select videos from 10 different categories, including hand actions, object interactions, food preparation, work activities, outdoor scenes, animals, water scenes, object handling, close-up shots, and nature scenes, with an overall average caption length of 51.7 words (see Appendix A.2.1 for statistics). We use PVD Benchmark to evaluate SigLIP [158], SigLIP2 [135], InternVL [19], and PE models, and the results can be found in Tab. 25.

## A.2    PE Video Dataset Details

As mentioned above, PVD consists of 1M videos, 120K of which have human-refined video captions and are selected for high motion content. We also select 15K from the 120K videos as a benchmark.

### A.2.1    PVD Benchmark Distribution

| Category | Number of videos | Avg. Caption Length |
|---|---|---|
| *Hand Actions* | 2143 | 54.2 |
| *Object Interactions* | 1864 | 42.6 |
| *Food Preparation* | 1691 | 56.8 |
| *Work Activities* | 1689 | 47.8 |
| *Outdoor Scenes* | 1558 | 50.7 |
| *Animals* | 1423 | 50.9 |
| *Water Scenes* | 1337 | 44.6 |
| *Object Handling* | 1307 | 51.6 |
| *Close-up Shots* | 1122 | 45.1 |
| *Nature Scenes* | 866 | 38.4 |

Table 10: **PVD** Benchmark Statistics. We created a dataset of 15K videos together with human-verified captions. The videos are motion-centered, covering both first-person and third-person views with a wide coverage of scenes.

### A.2.2    Video Data Filtering Pipeline

The goal of video data filtering is to identify videos that contain motions such as object motion, camera motion, interaction between objects, human actions, sequences of actions, and manipulation of objects, while rejecting videos with static scenes, like landscapes, or those that are artificial or highly edited.

To achieve this, we created a video filtering pipeline consisting of the following steps:

**Step 1**: Compute motion features. For each video, we compute a list of features from video frames, including frames per second (fps), number of frames, number of I-frames, motion vector magnitude, and motion vector variance, using off-the-shelf tools like OpenCV [9].

**Step 2**: Extract video frame features. For each video, we uniformly sample three frames and encode them using a DINOv2 model [96] and a SigLIP model [158].

**Step 3**: LLM Features. For each video, we also run a multimodal large language model (LLM) like Llava-Onevision QwenLM 2 0.5B [65] to extract MLLM features. We composed a list of 26 questions and performed MLLM inference on the videos. The questions can be found here in §A.2.3.

**Step 4**: Video Quality Scoring. We combine all the features collected so far and use a random forest model to predict a score between 0 and 5. To train the model, we manually annotated approximately 1,000 videos with scores between 0 and 5. A low score indicates that the video is almost static and can be nearly summarized by a single frame, while a high score indicates that there are multiple temporal events in the video, requiring several frames to accurately caption it. We use these annotated videos as training data to fit a random forest model for video quality score prediction.

**Step 5**: We apply k-means clustering to the videos and rank them within each cluster. By selecting the top-ranked videos from each cluster, we effectively reduce the number of duplicated videos in the final dataset.

### A.2.3    LLM Feature Extraction

We use LLaVA-OneVision [76] model to extract LLM features from the videos. For each video, we prompt with 26 different questions to extract features ranging from, "is the video a landscape video?" to, "are there any moving objects in the video?" The features are then used by a random forest model to determine the video quality score.

LLM Feature extraction question list

Is the camera capturing the scene static? Reply yes or no.
Is the camera capturing the scene moving? Reply yes or no.
Is the video capturing a landscape? Reply yes or no.
Is the video capturing a static scene? Reply yes or no.
Is the scene captured from a distance? Reply yes or no.
Is the video captured with a drone? Reply yes or no.
Is the video computer-generated? Reply yes or no.
Is the video content abstract? Reply yes or no.
Is there something moving through the scene? Reply yes or no.
Is there someone doing something in the video? Reply yes or no.
Are there several things moving in the video? Reply yes or no.
Is there an object that is being manipulated? Reply yes or no.
Are there animals in the video? Reply yes or no.
Is the scene mostly static? Reply yes or no.
Are things occluding each other in this video? Reply yes or no.
Is there something obstructing the view apart from the watermark? Reply yes or no.
Is there a large number of things in the video? Reply yes or no.
Are there more than 5 different objects in the video? Reply yes or no.
Is it hard to keep track of some entities because they are moving so much? Reply yes or no.
Is someone looking at a phone, a tablet or a computer screen? Reply yes or no.
Are they looking at a phone, a tablet or a computer screen during the whole video? Reply yes or no.
Are there several moving persons in this video? Reply yes or no.
Are there several moving animals in this video? Reply yes or no.
Are there several objects in this video? Reply yes or no.
Are there several similar-looking objects in the video? Reply yes or no.
Do they look similar? Reply yes or no.

### A.2.4 Video Caption

**LLM Summarization prompt**

LLM Summarization prompt 72 tokens

Create a concise caption of a video using the provided metadata, video caption, and frame captions.
TASK: Extract key information from the captions and combine it into an alt text format using single phrase or set of phrases that includes all relevant details.
Steps to Follow:
1. Review the metadata (title and description) for general context, you can rely it for entity names but do not rely on it as the primary source of information for your caption.
2. Blend title / description with video caption and frame captions for the main storyline
3. Extract the most relevant and concise information.
4. Combine extracted information into a alt text format using short phrase or set of phrases with approximately 120 tokens, considering special characters like comma as part of the token count.
5. Prioritize including all key information over sentence structure or grammar.
6. Minimize the use of special characters and focus of key information.
What to Avoid:
- Avoid adding or inferring information not present in the original metadata and captions.
- Avoid using complex sentence structures or prioritizing sentence flow.
Create a concise caption of the video based on the metadata, video caption, and frame captions.

### A.2.5  Human Caption Refinement

We employ human annotators to perform caption refinement for the LLM-generated captions in PVD, as described in Appendix A.1. Each annotator is paid a fair wage in compliance with all local laws in the annotators' jurisdictions. The refinement task was developed in accordance with an internal review to ensure ethical consideration for the participants. For instance, the annotators are instructed to reject the job completely if the videos contain any explicit content. Otherwise, we provided the following materials for the annotators as instructions for the task:

**Goal.** Given a video and a caption, directly refine it to make the caption:

1. No Repeating:
   (a) Remove any repeating, redundant information
   (b) Note it is ok to have fine-grained or atomic information if the caption already contains it, if the information is still unique
2. Accurate:
   (a) Every word in the caption is describing a fact in the video
   (b) If anything doesn't exist in the video at all, remove it
   (c) If anything is incorrect comparing to what the video shows, correct it
3. Action Focus:
   (a) Add any missing major action information into the caption
      i. As mentioned in 1. above, if an atomic action exists already, it is ok to keep it. No need to remove it. We only care about adding the missing major actions back.

In summary, the submitted caption should have **no repeating** information, every single word in the caption is **accurate** reflecting a fact in the video, and all **major actions** shown in the video have been **covered**.

**Refinement Criteria.** Use the following guidelines for correcting errors:

1. **Error:** If some words describe something doesn't show clearly in the video
   - Remove it from the caption
2. **Error:** If some words describe something in the video but incorrectly
   - Correct it from the caption to describe the fact in the video
3. **Error:** Repeating or redundant words
   - Merge words from the caption to make it concise and accurate or just remove it if no need of merge
4. **Error:** Action related words
   - If atomic actions exist, No need to remove it. Only remove words for Errors 1 and 3
   - If major actions miss, Add them back into the caption, in a concise and natural way. E.g.
     (a) If the missing actions can just be integrated as just a part of the original sentence, then just integrate it
     (b) If a new sentence is more natural to add the missing action back, then just add a new sentence.

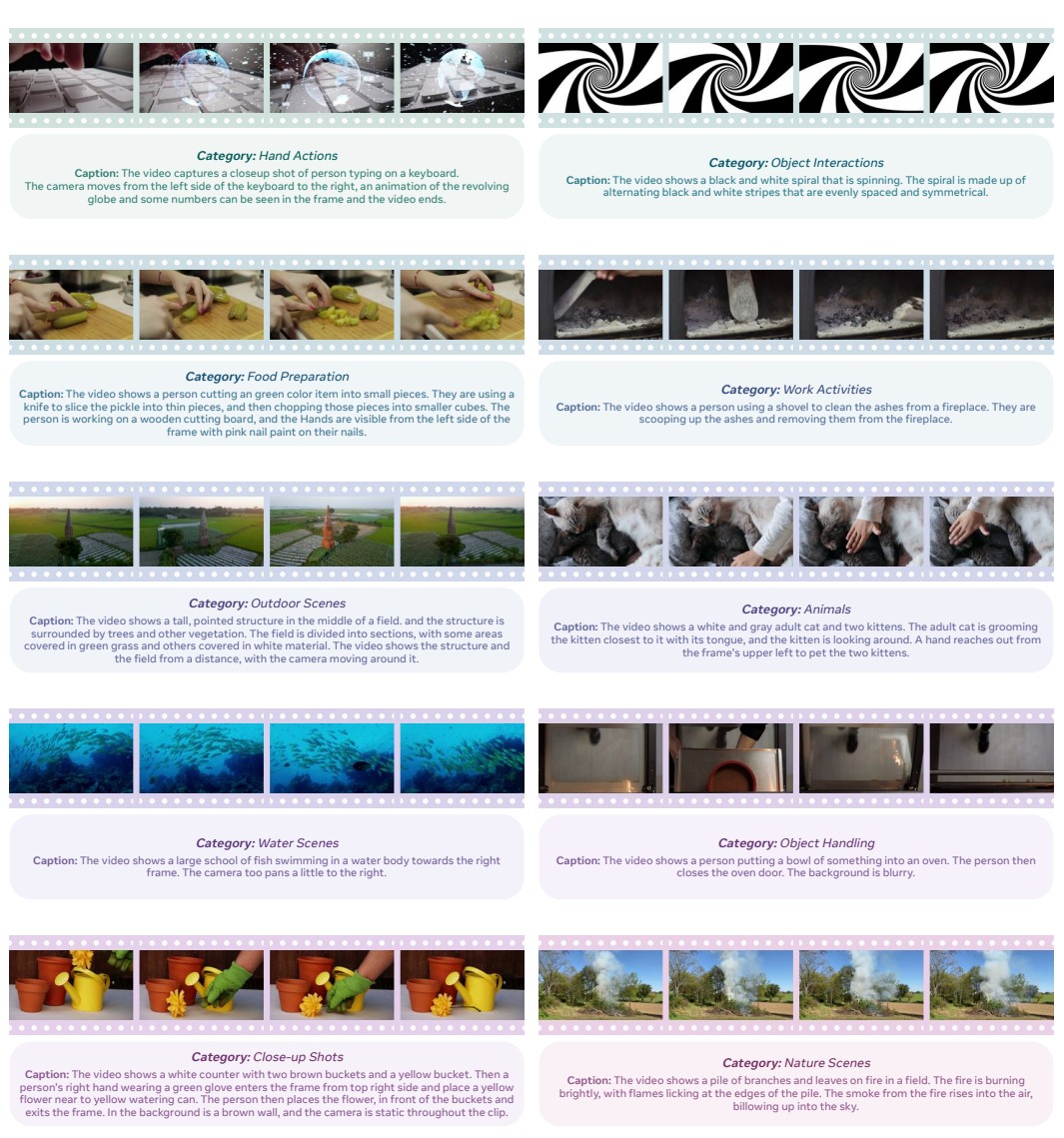

Figure 12: **More PE Video Dataset Examples.** For each of the ten categories, we randomly pick one video and show its video caption. The captions were generated by our video data pipeline and then refined by human annotators.

# B Implementation Details

## B.1 PE: Core

We provide additional implementation details for building PE$_{\text{core}}$. Our implementation is based on OpenCLIP.[1]

### B.1.1 Architecture and Training Setups

**Model Architecture.** Following CLIP, PE$_{\text{core}}$ comprises a Transformer-based [138] vision and a text encoder. We employ customized Transformer configurations as detailed in Tab. 11. For pooling, we an attention pooling block in the style of SigLIP [158] *with 8 heads* from the last-layer feature to construct image and video embeddings. Regarding positional embedding, we use 2D RoPE [124] for relative positional embeddings and 2D learnable absolute positional embeddings (abs) the same size as the model's input resolution. We interpolate positional embeddings to enable support for various resolutions beyond the default. The text context length is 72 for G-scale and 32 for B and L-scale models. Originally a bug, we find it optimal to *not disable the class token* when using attention pooling for smaller models. Thus, the B and L models use a class token, then the attention pooling layer probes all features at once (class token included). Finally, we use an input mean and standard deviation of $(0.5, 0.5, 0.5)$ for simplicity.

| Scale | Tower | Params | Width | Depth | MLP | Heads | CLIP Dim | Pooling | Positional Embedding | Resolution & Context Len | Patch Size | Class Token Register |
|-------|-------|--------|-------|-------|-----|-------|----------|---------|---------------------|-------------------------|------------|---------------------|
| B | Vision | 0.09B | 768 | 12 | 3072 | 12 | 1024 | Attn Pool | RoPE+Abs | 224 | 16 | ✓ |
|   | Text | 0.31B | 1024 | 24 | 4096 | 16 |      | EOS Token | Abs | 32 | - | - |
| L | Vision | 0.32B | 1024 | 24 | 4096 | 16 | 1024 | Attn Pool | RoPE+Abs | 336 | 14 | ✓ |
|   | Text | 0.31B | 1024 | 24 | 4096 | 16 |      | EOS Token | Abs | 32 | - | - |
| G | Vision | 1.88B | 1536 | 50 | 8960 | 16 | 1280 | Attn Pool | RoPE+Abs | 448 | 14 | ✗ |
|   | Text | 0.47B | 1280 | 24 | 5120 | 20 |      | EOS Token | Abs | 72 | - | - |

Table 11: **PE** Model Configurations with full details.

**PE Core Training.** As discussed in §2.3, the training of PE$_{\text{core}}$ involves three stages: 1) image pretraining; 2) image and video finetuning; and 3) an additional model distillation for smaller models. These three stages work together to develop a robust and effective PE$_{\text{core}}$ model.

We first provide training recipes for 1) image pretraining in Tab. 12 and 2) video finetuning in Tab. 13.

| config | values |
|--------|--------|
| optimizer | LAMB |
| $\beta_1, \beta_2$ | (0.9, 0.95) |
| weight decay | 0.05 |
| learning rate | 2e-3 |
| batch size | 131,072 |
| warm-up steps | 2K |
| training steps | 443K (B, L) / 656K (G) |
| data quantity | 5.4B |
| samples seen | 58B (B, L) / 86B (G) |
| max logit scale | 100 |
| mask reg ratio | 0.4 |
| mask reg batch | 8192 |
| progressive res | 112-160-224 (B) 
 98-154-224-336 (L) 
 98-154-224-336-448 (G) |
| data aug | aspect jitter `ar(0.75,1.33)` 
 rand crop `s(0.08,1)` 
 color jitter `j(0.32,0,0.32,0)` 
 hflip `p(0.5)` |

Table 12: **Image Pretraining.**

| config | values |
|--------|--------|
| optimizer | LAMB |
| $\beta_1, \beta_2$ | (0.9, 0.95) |
| weight decay | 0.05 |
| learning rate | 1e-6 |
| batch size | 4096 |
| warm-up steps | 2K |
| training steps | 5.4K |
| data quantity | 22M |
| samples seen | 22M |
| max logit scale | 100 |
| number of frames | 8 |
| data aug | aspect jitter `ar(0.75,1.33)` 
 rand crop `s(0.08,1)` 
 color jitter `j(0.32,0,0.32,0)` 
 hflip `p(0.5)` |

Table 13: **Video Finetuning.**

| config | values |
|--------|--------|
| optimizer | LAMB |
| $\beta_1, \beta_2$ | (0.9, 0.95) |
| weight decay | 0.05 |
| learning rate | 1e-6 |
| batch size | 16384 |
| warm-up steps | 2K |
| training steps | 269K |
| data quantity | 5.4B |
| samples seen | 4.4B |
| max logit scale | 100 |
| teacher logit scale | 200 (§C.3) |
| data aug | None |

Table 14: **Distillation.**

After training the largest G-scale model, we train the smaller models with image pretraining, then distill with image distillation in Tab. 14, then finally apply video finetuning at the end.

---

[1] https://github.com/mlfoundations/open_clip, MIT License

**Distillation Method.** To maximize the performance of smaller models (B and L scales in Tab. 2), we employ a distillation finetuning approach [48] using $PE_{core}G$ as the teacher. This process involves a short finetuning schedule where both the student and teacher models encode image and text inputs separately to compute image-to-text and text-to-image similarity distributions, similar to CLIP training [103]. The student's distributions are then optimized to match those of the teacher by minimizing KL-divergence, distilling multimodal relational knowledge from the teacher into the student.

Notably, we find that using a smaller softmax temperature for the teacher's distributions, specifically $0.5\times$ the temperature used for the student's distribution, significantly enhances the effectiveness of knowledge distillation. By leveraging the strong embeddings provided by $PE_{core}G$, our short distillation finetuning schedule significantly boosts the performance of both B and L scale models of $PE_{core}$ (see Appendix C.3).

### B.1.2 Zero-Shot Classification and Retrieval

**Zero-Shot Evaluation on Images and Videos.** We use CLIPBench[2] for zero-shot classification and retrieval benchmarking. The benchmark datasets and splits are obtained from the original dataset websites or HuggingFace. We extend the CLIPBench zero-shot evaluation to include video datasets such as MSR-VTT and Kinetics, and will release our model checkpoints, evaluation code, and scripts for reproducibility.

**Prompt Design.** For zero-shot image-text and video-text retrieval, we rely solely on the original captions without any additional prompts. In contrast, for zero-shot classification, we utilize task-specific prompts graciously provided by the InternVL [19] authors. All additional prompts will be released.

For example, we employ specific prompts for zero-shot image classification on various ImageNet benchmarks (e.g., ImageNet val, ImageNet v2) and video classification on Kinetics datasets (e.g., K400, K600, K700).

```
Zero-Shot Image Classification Prompts - ImageNet

a bad photo of a {c}. a photo of many {c}. a sculpture of a {c}. a photo of the hard
to see {c}. a low resolution photo of the {c}. a rendering of a {c}. graffiti of a {c}.
a bad photo of the {c}. a cropped photo of the {c}. a tattoo of a {c}. the embroidered
{c}. a photo of a hard to see {c}. a bright photo of a {c}. a photo of a clean {c}. a
photo of a dirty {c}. a dark photo of the {c}. a drawing of a {c}. a photo of my {c}.
the plastic {c}. a photo of the cool {c}. a close-up photo of a {c}. a black and white
photo of the {c}. a painting of the {c}. a painting of a {c}. a pixelated photo of the
{c}. a sculpture of the {c}. a bright photo of the {c}. a cropped photo of a {c}. a
plastic {c}. a photo of the dirty {c}. a jpeg corrupted photo of a {c}. a blurry photo
of the {c}. a photo of the {c}. a good photo of the {c}. a rendering of the {c}. a
{c} in a video game. a photo of one {c}. a doodle of a {c}. a close-up photo of the
{c}. a photo of a {c}. the origami {c}. the {c} in a video game. a sketch of a {c}.
a doodle of the {c}. a origami {c}. a low resolution photo of a {c}. the toy {c}. a
rendition of the {c}. a photo of the clean {c}. a photo of a large {c}. a rendition
of a {c}. a photo of a nice {c}. a photo of a weird {c}. a blurry photo of a {c}. a
cartoon {c}. art of a {c}. a sketch of the {c}. a embroidered {c}. a pixelated photo
of a {c}. itap of the {c}. a jpeg corrupted photo of the {c}. a good photo of a {c}.
a plushie {c}. a photo of the nice {c}. a photo of the small {c}. a photo of the weird
{c}. the cartoon {c}. art of the {c}. a drawing of the {c}. a photo of the large {c}.
a black and white photo of a {c}. the plushie {c}. a dark photo of a {c}. itap of a
{c}. graffiti of the {c}. a toy {c}. itap of my {c}. a photo of a cool {c}. a photo of
a small {c}. a tattoo of the {c}.
```

---

[2] https://github.com/LAION-AI/CLIP_benchmark, MIT License

```
Zero-Shot Video Classification Prompts - Kinetics
```

```
a photo of {c}. a photo of a person {c}. a photo of a person using {c}. a photo of
a person doing {c}. a photo of a person during {c}. a photo of a person performing
{c}. a photo of a person practicing {c}. a video of {c}. a video of a person {c}. a
video of a person using {c}. a video of a person doing {c}. a video of a person during
{c}. a video of a person performing {c}. a video of a person practicing {c}. a example
of {c}. a example of a person {c}. a example of a person using {c}. a example of a
person doing {c}. a example of a person during {c}. a example of a person performing
{c}. a example of a person practicing {c}. a demonstration of {c}. a demonstration of
a person {c}. a demonstration of a person using {c}. a demonstration of a person doing
{c}. a demonstration of a person during {c}. a demonstration of a person performing
{c}. a demonstration of a person practicing {c}.
```

**Evaluation Method.** Several works use different input transformations for different datasets when evaluating zero-shot performance (e.g., [32, 127, 135, 158]). To be as fair as possible, we follow [127] in evaluating with two transformations—center crop and non aspect ratio preserving resize ("squash")—and report the max between the two for all models and all datasets we evaluate. Additionally, ObjectNet has a red border around every image to facilitate deduplication, which we remove for evaluation. Finally, we follow [19] in using *retrieval reweighting* (DSL), applying the softmax score distribution to the similarities used for retrieval:

$$\text{scores} = \text{scores} * \text{softmax}(\text{scores}, \text{dim}=0) \tag{1}$$

This slightly improves retrieval for most models, so we do it for all models we evaluate for fairness. Notably, we were able to reproduce the reported numbers for most papers with these techniques, but for cases where we could not, we default to the reported number.

## B.2 PE: Language Alignment

We provide details of the MLLM experimental setup in §4. We describe *data*, *model*, and *training* separately.

**Data.** Our MLLM training contains *warmup* data and *supervised finetuning (SFT)* data. Our warmup data is a 1M subset image-text pairs of our $\text{PE}_{\text{core}}$ pretraining dataset. For SFT data, we use a diverse data mix consisting of 2.6M unique samples. This dataset is composed of 1.7M[3] visual QAs samples from the Cauldron [64], 0.5M grounded QA pairs from Visual Genome [59], Flickr-Entities [100] and Densely Captioned Images [136], 0.1M image-captioning pairs from COCO [74] and 0.3M text-only samples. This comprehensive data mix allows us to thoroughly assess our model's capabilities in various MLLM tasks.

**Model.** As described in §D.1, we use a simple vision-language model architecture where a vision encoder and a pretrained decoder-only LLM are connected by a vision projector. For all tables, we use either Llama3.1-instruct 8B or QwenLM 2.5-instruct 7B as a language model, and 2-layer MLP as a vision projector. For fair comparison, we use the native resolution for image input. During inference, we evaluate the models on video tasks in *zeroshot* manner: We concatenate all video frames into a sequence and feed to language model, without seeing video samples during SFT. For all video tasks, we use 8 frames with the same native resolution of height and width. For $\text{PE}_{\text{core}}$ and $\text{PE}_{\text{lang}}$, this makes $448 \times 448 \times 8$ input and $32 \times 32 \times 8$ vision tokens.

**Training.** MLLM training consists of *warmup* and *supervised finetuning (SFT)* stages. In both stages, we freeze vision encoder and train vision projector and LLM. During warmup stage, we use a global batch size of 128 with a learning rate of $1 \times 10^{-4}$. We gradually increase the learning rate from $1 \times 10^{-6}$ to $1 \times 10^{-4}$ over 120 steps, and follow a cosine learning rate decay schedule to train a total of 8,000 steps. During SFT stage, we use a global batch size 256 with a learning rate of $1 \times 10^{-5}$. Similar to the warmup, we gradually increase the learning rate from $1 \times 10^{-7}$ to $1 \times 10^{-5}$ over 300 steps, and follow a cosine learning rate decay schedule to train a total of 12.5K steps. We truncate text-sequences longer than 2,048 tokens on top the visual tokens. This makes the maximum sequence length to be (num. vision tokens) $+ 2,048$. With $448 \times 448$ input resolution and patch size of 14, we set the maximum sequence length to $1,024 + 2,048 = 3,072$. To represent bounding boxes on output side for image grounding tasks, we simply use text tokens to represent each bounding box:

---

[3]We excluded multi-images samples.

each coordinate is normalized between 000 and 999, in "[x, y, x, y]" box format for top-left and bottom-right corners (*e.g.*, [012, 122, 633, 782]).

For all baselines, we search for the **best** intermediate layer features to adapt to LLM. We search over $\{-1, -2, -4, -6, -8, -10, -12, -14, -16, -18, -20, -40\}$ layers (counting from last) and report the best result in average over OCR/Chart/Document Q&A, Visual Q&A, Image Captioning and Video Understanding.

## B.3   PE: Spatial Alignment

### B.3.1   Training Details

**Loss Functions.** For self-aligning to frozen $PE_{core}G$ layer 41 features ($L_{core}$), we minimize the negative cosine similarity:

$$L_{core} = -\frac{1}{n_{tok}} \sum \left( \frac{(S_{50})(T_{41})^T}{||S_{50}|| \cdot ||T_{41}||} \right) \tag{2}$$

where $S_{50}$ denotes the last layer features of the student, $T_{41}$ denotes frozen layer 41 features from $PE_{core}G$, and $n_{tok}$ represents the number of tokens. Note that we chose 41 fairly arbitrarily (it is layer 40 when written with indexing from 0). Judging by Fig. 4, any layer around 40 should work (and 39 may be slightly better).

For the encouraging locality loss ($L_{loc}$), we compute the pairwise cosine similarity between a model's own tokens and itself. This forms a "spatial correspondence map" for what tokens should be considered similar. We then compute the same for the student, and minimize the difference between the two with MSE loss:

$$L_{loc} = \frac{1}{n_{tok}^2} \sum \left( \frac{(S_{50})(S_{50})^T}{||S_{50}||^2} - \frac{(T_{SAM})(T_{SAM})^T}{||T_{SAM}||^2} \right)^2 \tag{3}$$

where $T_{SAM}$ denotes the "SAM Mask Logits" constructed in §E.1.2. We also find using a temperature ($t$) on the SAM teacher's pairwise cosine similarity term ($x$) useful: $e^{t(x-1)}$. The full loss is $L_{spatial} = L_{core} + L_{loc}$.

**Hyperparameters.** In Tab. 15 we show the training hyperparameters for spatial alignment, finetuned on top of the initial $PE_{core}G$ checkpoint. Then in Tab. 16 and Tab. 17, we show the settings for the two teachers and losses. Note that when running the teachers, we run them on the exact same image as the student (same data aug and all). Additionally, because the SAM 2.1 teacher operates at a resolution of 1024, we upsample the image, generate the mask logits, and then downsample the result. Both teachers are frozen.

| config | values |
|---|---|
| optimizer | LAMB |
| $\beta_1, \beta_2$ | (0.9, 0.95) |
| weight decay | 0.05 |
| learning rate | 5e-4 |
| batch size | 12,288 |
| warm-up steps | 0 |
| training steps | 24K |
| data quantity | 5.4B ($PE_{core}$ PT Data) |
| samples seen | 300M |
| | |
| resolution | 448 |
| mask ratio | 0.75 |
| mask size | 2×2 tokens |
| | |
| droppath | 0.4 |
| layerscale | 0.1 |
| | |
| data aug | aspect jitter ar(0.75,1.33) color jitter j(0.32,0,0.32,0) hflip p(0.5) |

Table 15: **Spatial Alignment.**

| config | values |
|---|---|
| model | SAM 2.1-L |
| layer | mask logits |
| resolution | 1024 (interp→448) |
| | |
| loss | Eq. 3 |
| loss weight | 1 |
| temperature | 20 |
| | |
| sample points | 32×32 (1024) |
| pred iou threshold | 0 |
| stability score threshold | 0 |
| mask threshold | 0 |

Table 16: **SAM 2.1 Teacher.**

| config | values |
|---|---|
| model | $PE_{core}G$ |
| layer | 41 |
| resolution | 448 |
| | |
| loss | Eq. 2 |
| loss weight | 1 |

Table 17: **$PE_{core}G$ Teacher.**

### B.3.2 Visualization Method

To visualize the features in Fig. 10 and Fig. 20, our goal is to map a 1536-dimensional space down to 3 dimensions to view how the model encodes each token in relation to each other. One naive approach would be to apply PCA with 3 dimensions across all token in the image. However, we find this alone can be misleading.

Specifically, if the model has rich semantics, it should be the case that most of those 1536 features have some useful information in them. Some of that information could be spatially contiguous, some of it not. We want PCA to only select the *spatially contiguous* information, since we are trying to evaluate the spatial quality of the features. However, naively applying PCA will not necessarily do that, especially for models with information aggregated in "global tokens" (§E.1.1). Despite these tokens carrying important information, they are not spatially contiguous. Thus, if PCA dedicates a large portion of its 3 dimensions to global tokens, the features will *look* like their spatial quality is bad, despite the features containing good spatial information.

So, how do we select for only the *spatially contiguous* information to visualize? The answer is simple: by definition, the spatially contiguous information will be. . . spatially contiguous. To keep the spatially contiguous information while lowering the impact of the global tokens, we can simply apply a low pass filter to the features (specifically, a gaussian blur with kernel size 3 and a $\sigma$ of 1). To retain the detail of the original features, we can average the two together. Thus, to visualize features, we use the 3D PCA of the of the following. $x$ denotes the model's output features, and $g(x)$ denotes gaussian blur.

$$0.5x + 0.5g(x, k = 3, \sigma = 1) \tag{4}$$

We show the impact of this in Fig. 13. Blurring the features make them appear more detailed! In reality, that information was always there, just PCA did not show it. Thus, great care must be taken when visualizing high dimensional feature spaces. If they were easy to map to 3 dimensions—you would not need 1536 of them!

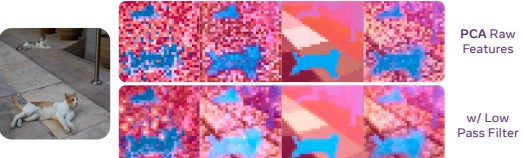

Figure 13: **Feature Visualization Ablation.** With raw features (top row), PCA misses spatially contiguous parts of the feature space and instead focuses on global tokens (which carry information but are not spatially coherent). By applying a simple low pass filter (bottom row), we can reveal spatial information that PCA originally missed (see column 2: with raw features, the background looks like a mess, with the low pass filter the tiles become visible).

Then, to map the PCA dimensions to RBG pixel values, we map each PCA component to a corresponding channel in LCh color space, then convert those LCh colors to RGB to get the final image. Note that we use LCh instead of RGB directly for aesthetic reasons, and also because LCh is a cylindrical color space—where smooth changes to the values look like smooth changes in colors to humans—and thus is easier to discern.

### B.3.3 Frozen Feature Dense Prediction

We discuss the detailed settings of the results for dense prediction with frozen features in Tab. 6. Each model is evaluated with its native resolution up to 448 or 448 (whichever is optimal).

**Zero-Shot Tracking.** We evaluate our pretrained models on label propagation task using the protocols in [51, 104] on DAVIS dataset [101]. This evaluation does not require any finetuning or probing, therefore preserves the spatial features in the model. Following Toto [104], we use the features from the last n = 7 frames to find the nearest neighbor patch in the current frame, and then propagate the masks from the previous frames to the current frame. Note that this evaluation method does not require any training.

**Semantic Segmentation.** For semantic segmentation, we evaluate our pretrained models on ADE20K [164] semantic segmentation task. We use a batch norm and a linear layer to map intermediate spatial features to segmentation masks following [96]. The models are evaluated and then features are resized to $518 \times 518$. We only use features from single layer. The probing layers are finetuned with AdamW [81] with a learning rate of 0.001.

**Depth Estimation.** For depth estimation on NYUv2 [120], we follow [73, 96]. We use a DPT-head [106] on top of our frozen pretrained model and use only single layer features. We scale the size of the DPT-head for each models based on the hidden size for each architecture. Because NYU is a small dataset and the models we evaluate are large, we observe the results for most models are noisy and prone to overfitting. Thus, for fair comparison we train *all models* for 20 epochs and for *all models* take the lowest validation loss over all epochs.

**Frozen Detection.** For the frozen feature detection results presented in §3, we evaluated using Mask R-CNN [42] as a probe. We used a resolution of 1024 for Fig. 4 and 768 for the remainining experiments in §3. Because the backbones were frozen, we did not add any global attention and instead simply tiled the input image with a window size of 32 for the 1024px experiments and 24 for the 768px experiments. All models were interpolated to patch 16. Finally, the backbones were frozen and only the FPN and R-CNN heads trained for 15 epochs on COCO with a stepwise decay LR without drop path.

### B.3.4 End-to-End Finetuning Detection and Segmentation

We provide a detailed discussion of settings of end-to-end finetuning on detection and segmentation presented in Tab. 7 using Detectron2 [147].[4] The hyperparameters can be found in Tab. 18. We find that the default 100-epoch protocol in ViTDet [71, 146] causes overfitting problems in COCO experiments especially for billion-level parameter vision encoders, so we tune the training epochs, learning rate, drop path and learning rate decay accordingly.

The LVIS experiment setting is the same as COCO except all L-size models use learning rate of 2e-4 and all g-size and G-size models use 75 epochs.

| config | values | model | lr | epochs | drop path | lr decay | layers | global window index | window size |
|---|---|---|---|---|---|---|---|---|---|
| optimizer | AdamW | MetaCLIP-G | 5e-5 | 75 | 0.5 | 0.9 | 48 | (11, 23, 35, 47) | 14 |
| optimizer momentum | (0.9, 0.999) | SigLIP2-so | 1e-4 | 100 | 0.4 | 0.8 | 27 | (2, 10, 18, 26) | 14 |
| weight decay | 0.1 | SigLIP2-g | 5e-5 | 75 | 0.5 | 0.9 | 40 | (9, 19, 29, 39) | 14 |
| learning rate schedule | Step-wise decay | DINOv2-L | 1e-4 | 100 | 0.4 | 0.8 | 24 | (5, 11, 17, 23) | 32 |
| batch size | 64 | DINOv2-g | 5e-5 | 36 | 0.5 | 0.9 | 40 | (9, 19, 29, 39) | 32 |
| image size | $1024 \times 1024$ | **PE$_{core}$G** | 5e-5 | 75 | 0.5 | 0.9 | 50 | (12, 24, 36, 49) | 32 |
| augmentation | LSJ [0.1, 2.0] | **PE$_{spatial}$G** | 5e-5 | 36 | 0.5 | 0.9 | 50 | (12, 24, 36, 49) | 32 |
| postional embedding | abswin [7] | | | | | | | | |
| patch size | 16 | | | | | | | | |

Table 18: **Settings for End-to-End Finetuning Detection and Segmentation.**

### B.3.5 System-Level Comparison on Detection

We describe our implementation for system-level comparison to the state-of-the-arts on COCO object detection in Tab 8. Our implementation is based on the DETA repository.[5] We replace the vision encoder with our PE$_{spatial}$ and maintain the same hyperparameters as in the end-to-end finetuning settings, while keeping the detector unchanged. The training process consists of three stages:

| Test-Time Aug | AP$_{box}$ |
|---|---|
| No TTA | 65.2 |
| + More Queries | 65.3 |
| + SoftNMS [6] | 65.8 |
| + Flip Aug | 65.8 |
| + Multiscale Aug | **66.0** |

Table 19: **Test-Time Aug** for system-level comparison on COCO in Tab. 8.

1. **Initial Training**: Train on Objects365 for 12 epochs with an image resolution of $1024 \times 1024$, a total batch size of 256, and a learning rate of 2e-4, which is divided by 10 at the 10th epoch.

---

[4]https://github.com/facebookresearch/detectron2, Apache 2.0
[5]https://github.com/jozhang97/DETA, Apache 2.0

2. **Increasing Resolution**: Continue training on Objects365 for 6 epochs with a resolution of $1536 \times 1536$, a total batch size of 128, and a learning rate of 5e-5, which is divided by 10 at the 5th epoch.

3. **Finetuning**: Finetune on COCO dataset for 12 epochs with an image resolution of $1728 \times 1728$, a total batch size of 64, and a learning rate of 5e-5, which is divided by 10 at the 8th epoch.

4. **Further Increasing Resolution**: Further finetune on COCO dataset for 3 epochs with a resolution of $1824 \times 1824$, a total batch size of 64. To save GPU memory, we use SGD optimizer instead of Adam, with a learning rate of 5e-3, which is divided by 10 at the 2th epoch.

We apply a series of test-time augmentations to further improve the performance, see Tab. 19.

# C Additional PEcore Results

## C.1 Robust Image Pretraining

In Tab. 20, we present the raw data for the robustness metrics in Fig. 2. Across the board, each change improved almost all metrics (with the exception of progressive resolution slightly hurting the average and mask regularization slightly hurting ImageNet Adversarial). The fact that there were no tradeoffs to these changes, indicate that their improvements to the features are general. This could be why most of these changes improved performance for downstream tasks as well.

Note that in §2.1, we only discuss changes that we know to work. There are several changes that we have tried that do not work (i.e., do not improve performance or lower performance). For instance: average pooling instead of using a class token, increasing the text tower size, using hue or contrast jitter, and maintaining the same resolution throughout training but dropping tokens instead of progressive resolution (FLIP-style).

We also find increasing batch size and increasing training iterations for an L scale model to have equivalent effects. This is in contrast to the batch size scaling observed by [158], but it is possible that this difference is down to a hyperparameter issue.

| | Step | *Avg Class.* | ImageNet *val [25]* | ImageNet *v2 [109]* | ObjectNet *IN Classes [4]* | ImageNet *Adversarial [46]* | ImageNet *Renditions [45]* | ImageNet *Sketch [140]* |
|---|---|---|---|---|---|---|---|---|
| | | | | | *Zero-Shot Classification* | | | |
| 1 | Baseline | 75.3 | 78.9 | 71.9 | 73.7 | 68.3 | 91.1 | 67.8 |
| 2 | Progressive Resolution | 75.1 | 78.9 | 71.8 | 72.4 | 69.9 | 90.5 | 67.0 |
| 3 | High Batch Size | 76.2 | 79.5 | 72.8 | 74.1 | 71.8 | 91.0 | 68.1 |
| 4 | LAMB and High LR | 76.9 | 79.9 | 73.3 | 74.3 | 73.5 | 91.5 | 68.6 |
| 5 | High Resolution (336) | 78.3 | 80.4 | 73.8 | 75.6 | 79.2 | 92.0 | 68.8 |
| 6 | 2D RoPE | 79.2 | 80.7 | 74.1 | 77.4 | 80.9 | 92.7 | 69.4 |
| 7 | Attention Pooling | 80.1 | 81.0 | 74.8 | 78.4 | 82.9 | 93.4 | 69.9 |
| 8 | Data Augmentation | 80.8 | 81.1 | 75.2 | 80.8 | 83.1 | 93.5 | 71.2 |
| 9 | Mask Regularization | 80.9 | 81.3 | 75.3 | 80.9 | 82.8 | 93.8 | 71.2 |

Table 20: **Robust Image Pretraining Full Results.** Raw results for the robustness metrics metrics in Fig. 2. Almost every change improves every metric, but some metrics are improved more than others (e.g., ObjectNet and ImageNet-A).

**Scaling Behavior.** In Fig. 14, we show the performance of our recipe (Fig. 2.9) *vs.* the original CLIP recipe (Fig. 2.1) across S/14, B/14, and L/14 models. For each benchmark, our recipe scales around the same rate or better than the original CLIP recipe. On some difficult datasets like ObjectNet [4] and ImageNet Adversarial [46], our recipe shows distinctly better scaling. This indicates that the improvements in performance were not at the cost of scalability, meaning we can further benefit from scaling the model size.

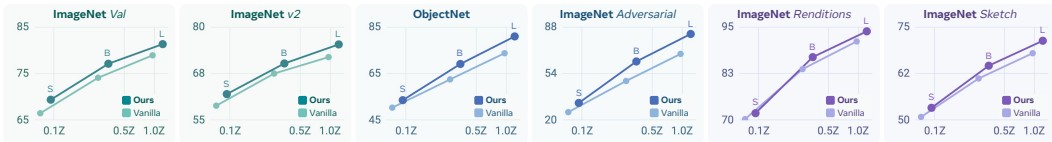

Figure 14: **Scaling Behavior (Model Size).** Results before and after our recipe changes (Fig. 2) for S/14, B/14, and L/14 models. Our recipe improves scaling for difficult metrics like ObjectNet [4] and ImageNet Adeversarial [46].

In Fig. 15, we additionally show the performance of our recipe *vs.* the original CLIP recipe across L/14 models trained with 120K steps (one-third schedule), 240K steps (two-thirds schedule), and 360K steps (full ablation schedule). All models are their own training runs with full learning rate annealing and the progressive resolution schedule adjusted proportionally. We see nearly linear trends for our recipe on most datasets. This suggests we can train longer for more performance, even at L scale and with 24B samples seen already.

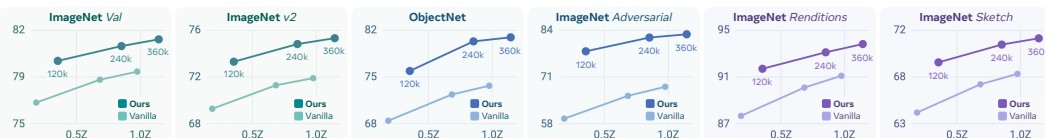

Figure 15: **Scaling Behavior (Training Steps).** Results before and after our recipe changes for an L/14 model trained with 120K, 240K, and 360K steps, adjusting the learning rate and progressive resolution schedules accordingly. Despite our recipe being much stronger than the original, there is still room for further improvement by training longer.

## C.2 Additional Video Ablations

**Human Refined Data for Captioning.** In Tab. 21, we ablate the captioning performance of our video captioner component in our video data engine with or without human-refined video captioning data. For all benchmarks tested, the human-refined data significantly improves captioning performance.

| Captioner | AuroraCap [12] | | VCG Diverse [85] | | VCG [84] |
|---|---|---|---|---|---|
| | Score | Acc | Score | Acc | Score |
| PLM | 2.2 | 51.9 | 3.1 | 65.1 | 34.3 |
| PLM + Human-Refined Data | **3.4** | **71.1** | **3.6** | **79.4** | **35.2** |

Table 21: **Video Captioning.** Adding human-refined data greatly boosts captioning performance.

**Video Scaling Behavior.** In Fig. 16, we investigate the impact of scaling recaptioned video data on a later checkpoint of the same image-only model as in Fig. 1. Notably, scaling synthetic video data demonstrates consistent improvement in both image and video benchmarks. Full results of this scaling experiment can be found in the Appendix 13.

In the top row, scaling synthetic video data consistently improves performance on image benchmarks, with monotonic improvements of +1.1% in ObjectNet and +1.6% in ImageNet Adversarial. ImageNet val and ImageNet v2 have smaller gains, with accuracy increases of 0.3% to 0.5%, plateauing at ∼7M samples. We also observe a significant boost to zero-shot retrieval (here, COCO [74]) of +3.8% to +4.1% top-1 recall.

The video tasks listed in the bottom row demonstrate a consistent story. We observe a significant jump in performance between none and 3M videos across all video classification tasks, indicating that there is a domain gap for image-only models that hinders their ability to perform well on video out of the box. Further scaling synthetic video data leads to substantial performance gains in both video classification and retrieval. Video classification accuracy improves consistently by +5.6% to +11.7% without plateauing, while video retrieval shows significant improvements of +7.7 to +15.3 top-1 recall.

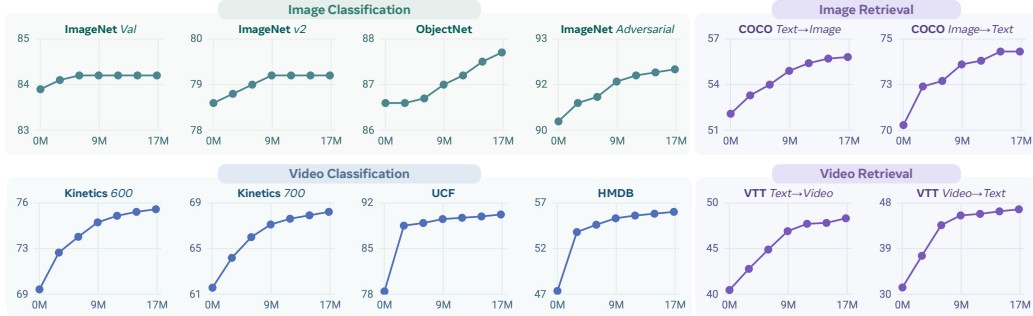

Figure 16: **Video Data Scaling.** Finetuning on videos recaptioned by the PE video data engine from 0M (baseline image-only model) to 17M samples consistently improves both image and video performance, both classification and retrieval.

These experiments highlight the quality of our video data engine and its ability to significantly improve encoder performance, even with only a relatively modest 17M videos compared to the billions of images seen during pretraining. Our video data engine is a vital component in build a strong, unified image-video encoder.

**Video Scaling Behavior Detailed Results.** The detailed video data scaling results are presented in Tab. 22. Our experiments demonstrate that increasing the number of synthetic video data generated by the proposed video data engine enhances the performance of classification and retrieval on both image and video benchmarks. On image benchmarks, while improvements on ImageNet val and v2 plateaued earlier compared to ObjectNet and ImageNet Adversarial, MS-COCO retrieval performance continued to show gains. On video benchmarks, scaling synthetic video data consistently yields better performance for both classification and retrieval tasks. We expect that further scaling up the video data with our video data engine will continue to drive performance improvements.

| Video Data Size | Average Image | Image Zero-Shot | | | | | | Average Video | Video Zero-Shot | | | | | | |
|---|---|---|---|---|---|---|---|---|---|---|---|---|---|---|---|
| | | ImageNet val [25] | ImageNet v2 [109] | ObjectNet IN Classes [4] | ImageNet Adversarial [46] | MS-COCO txt→img [74] | MS-COCO img→txt [74] | | Kinetics 400 [54] | Kinetics 600 [54] | Kinetics 700 [54] | UCF 101 [123] | HMDB 51 [61] | MSR-VTT txt→vid [151] | MSR-VTT vid→txt [151] |
| 0M | 77.0 | 83.9 | 78.6 | 86.6 | 90.3 | 52.1 | 70.3 | 57.0 | 70.3 | 69.4 | 61.6 | 78.5 | 47.4 | 40.5 | 31.4 |
| 3M | 77.7 | 84.1 | 78.8 | 86.6 | 90.9 | 53.3 | 74.2 | 61.6 | 72.4 | 72.2 | 64.2 | 88.5 | 53.8 | 42.8 | 37.6 |
| 6M | 78.0 | 84.2 | 79.0 | 86.7 | 91.1 | 54.0 | 72.7 | 63.6 | 73.5 | 73.4 | 66.0 | 88.9 | 54.6 | 44.9 | 43.6 |
| 8M | 78.4 | 84.2 | 79.2 | 87.0 | 91.6 | 54.9 | 73.6 | 64.8 | 74.5 | 74.5 | 67.7 | 89.5 | 55.3 | 46.9 | 45.5 |
| 11M | 78.6 | 84.2 | 79.2 | 87.2 | 91.8 | 55.4 | 73.8 | 65.2 | 75.1 | 75.0 | 67.6 | 89.7 | 55.6 | 47.7 | 45.8 |
| 14M | 78.8 | 84.2 | 79.2 | 87.5 | 91.9 | 55.7 | 74.3 | 65.5 | 75.4 | 75.3 | 67.9 | 89.9 | 55.8 | 47.8 | 46.3 |
| 17M | 78.9 | 84.2 | 79.2 | 87.7 | 92.0 | 55.8 | 74.3 | 65.8 | 75.7 | 75.5 | 68.2 | 90.2 | 56.0 | 48.3 | 46.7 |

Table 22: **Scaling Video Data.** Increasing the number of synthetic video data generated by our proposed video data engine consistently enhances the performance of image and video classification and retrieval tasks.

## C.3 Smaller Models

**Ablation: Distillation Temperature.** To optimize the performance of smaller models (B and L-scales in Tab. 2), we utilize a distillation finetuning approach with $PE_{core}G$ as the teacher model. During this process, both student and teacher models encode image and text inputs to compute image-to-text and text-to-image similarity distributions, similar to CLIP training [103]. The student's distributions are then optimized to match those of the teacher by minimizing KL-divergence loss on both image-to-text and text-to-image similarity distributions.

We find that using a fixed and smaller temperature (i.e., higher logit scale), which controls the range of logits in the softmax, significantly enhances the effectiveness of distillation. This results in a sharper distribution for the teacher's distributions. In contrast, the student's temperature remains learnable, consistent with our pretraining procedure and CLIP training.

In Tab. 23, we present an ablation study examining the impact of temperature on the teacher's distribution. For this analysis, we utilize a pretrained *vanilla* CLIP model (ViT-B/14, resolution 224), which serves as a baseline for comparison (see §2.1 for details). The models are finetuned using

| Model | Teacher's Temp | Model Scale | Zero-Shot Classification | | | | | | |
|---|---|---|---|---|---|---|---|---|---|
| | | | Avg Class. | ImageNet val [25] | ImageNet v2 [109] | ObjectNet IN Classes [4] | ImageNet Adversarial [46] | ImageNet Renditions [45] | ImageNet Sketch [140] |
| vanilla pretrained model | - | B | 66.2 | 74.2 | 67.4 | 62.5 | 50.2 | 83.0 | 59.8 |
| distillation | ×2 | B | 65.2 | 71.8 | 65.5 | 61.4 | 50.2 | 83.6 | 58.6 |
| | ×1 | B | 68.0 | 74.9 | 68.1 | 64.7 | 54.1 | 85.3 | 61.1 |
| | ×0.7 | B | 68.2 | 75.1 | 68.2 | 65.3 | 54.4 | 85.1 | 61.3 |
| | ×0.5 | B | **68.3** | 75.2 | 68.2 | 65.3 | 54.2 | 85.2 | 61.4 |

Table 23: **Ablation Study on Teacher's Distribution Temperature.** We evaluate the effect of varying temperatures on the teacher's distribution, using a pretrained vanilla CLIP model (ViT-B/14, resolution 224) as a baseline (details in §2.1). The models are finetuned via distillation with a short schedule of 50K steps.

distillation with a concise schedule of 50K steps. Notably, our results show that employing a smaller temperature for the teacher's distributions yields improved performance on zero-shot ImageNet benchmarks.

**Building strong smaller models.** In Tab. 24, we demonstrate our step-by-step training strategy for building strong smaller models at the L scale, as discussed in §2.3. Specifically, we outline our approach to image pretraining, image distillation, and video finetuning, and distillation. Leveraging the robust foundation established by our pretraining techniques (§2.1), we show that distilling from $PE_{core}G$, our strongest unified perception encoder, yields improvements on both image and video benchmarks. Furthermore, a short-scheduled video finetuning provides an additional boost in performance on both benchmarks.

| Model | Stage | Average Image | ImageNet val [25] | ImageNet v2 [109] | ObjectNet IN Classes [4] | ImageNet Adversarial [46] | MS-COCO txt→img [74] | MS-COCO img→txt [74] | Average Video | Kinetics 400 [54] | Kinetics 600 [54] | Kinetics 700 [54] | UCF 101 [123] | HMDB 51 [61] | MSR-VTT txt→vid [151] | MSR-VTT vid→txt [151] |
|---|---|---|---|---|---|---|---|---|---|---|---|---|---|---|---|---|
| | | | *Image Zero-Shot* | | | | | | | *Video Zero-Shot* | | | | | | |
| SigLIP2-L/16 | - | 76.0 | 83.1 | 77.4 | 84.4 | 84.3 | 55.3 | 71.4 | 56.2 | 65.3 | 62.5 | 56.8 | 86.7 | 49.3 | 41.5 | 31.4 |
| $PE_{core}L$ | image pretraining | 75.1 | 82.9 | 76.8 | 81.8 | 85.6 | 53.0 | 70.4 | 59.0 | 68.0 | 67.7 | 58.5 | 85.5 | 57.7 | 42.0 | 33.4 |
| $PE_{core}L$ | +image distill from $PE_{core}G$ | 77.6 | **83.6** | **78.1** | 84.4 | 88.9 | 56.0 | 74.7 | 64.5 | 73.0 | 72.6 | 64.8 | 86.5 | 58.0 | 47.9 | 48.4 |
| $PE_{core}L$ | +video finetuning | **78.0** | 83.5 | 77.9 | **84.7** | **89.0** | **57.1** | **75.9** | **65.3** | **73.4** | **72.7** | **65.3** | **87.1** | **58.5** | **50.3** | **50.1** |

Table 24: **Building Strong Smaller Models.** This table illustrates the step-by-step process of developing the $PE_{core}L$ 336px model, as outlined in §2.3. Starting with the pretrained $PE_{core}L$, both image distillation, along with video finetuning, enhance performance across image and video benchmarks, resulting in a unified L-scale model.

## C.4 Additional Results

**Additional Zero-Shot Benchmarks.** We further evaluate $PE_{core}$ on an additional set of zero-shot classification and retrieval benchmarks we construct in Tab. 25 to address key gaps in common benchmarks. For comparison, we also evaluate SigLIP2 [135] and InternVL-C [19] on these benchmarks.

First, we note that the version of ObjectNet [4] that is standard to benchmark robustness (e.g., in Tab. 3) is *not* the full set. ObjectNet consists of 313 classes of objects in challenging and uncommon orientations, locations, and viewpoints. However, the standard version used for benchmarking is a 113 class subset of classes that overlap with ImageNet-1k [25]. Naturally, benchmarking in this way rewards performing well on ImageNet classes over generality. To remove this bias, we construct the full ObjectNet set with all classes and compare to the reduced ObjectNet set in Tab. 25. Surprisingly, we find that while $PE_{core}G$ performs +7.6% over InternVL-C and only +0.2% over SigLIP2-g-opt on the reduced ObjectNet set, it performs +11.8% over InternVL-C and +0.9% over SigLIP2-g-opt on the full set of classes, highlighting PE's generality.

Next, we include iNaturalist [137] as a *zero-shot* benchmark because of its level of specificity with 2,101 fine-grained long-tail classes. $PE_{core}G$ outperforms the next best SigLIP2-g-opt model by *+9.6%*, emphasizing PE's long tail knowledge. We then evaluate PE's cultural diversity on Dollar

| Model | Encoder Params | Resolution | Data | ObjectNet IN Overlap (113) [4] | ObjectNet All Classes (313) [4] | iNaturalist 2017 [137] | Dollar St 58 [38,110] | TextCaps img→txt [119] | TextCaps Flip img→txt [119] | PVD Bench text→vid | PVD Bench vid→txt |
|---|---|---|---|---|---|---|---|---|---|---|---|
| | | | | *Zero-Shot Classification* | | | | *Zero-Shot Retrieval* | | | |
| SigLIP2-B/16 [135] | 0.1B | 224 | 10B | **73.6** | **59.1** | 16.9 | **55.9** | 72.0 | 69.8 | 53.9 | 60.1 |
| **$PE_{core}B$** | 0.1B | 224 | 5.4B | 71.9 | 58.3 | **25.9** | 52.1 | **72.3** | **71.9** | **59.8** | **61.1** |
| SigLIP2-L/16 [135] | 0.3B | 384 | 10B | 84.4 | 73.2 | 26.7 | 57.6 | 78.0 | 76.2 | 61.9 | **67.1** |
| **$PE_{core}L$** | 0.3B | 336 | 5.4B | **84.7** | **74.3** | **35.3** | **59.6** | **78.5** | **78.3** | **64.7** | 65.2 |
| InternVL-C [19] | 5.5B | 224 | 5B | 80.6 | 67.2 | 19.4 | 58.2 | 72.3 | 67.8 | 63.4 | 65.1 |
| SigLIP2-g-opt [135] | 1.1B | 384 | 10B | 88.0 | 78.1 | 31.5 | 59.3 | **78.8** | 76.9 | 62.5 | 67.1 |
| **$PE_{core}G$** | 1.9B | 448 | 5.4B | **88.2** | **79.0** | **41.1** | **62.3** | **78.8** | **78.7** | **77.0** | **76.6** |

Table 25: **Additional Zero-Shot Results.** We present several additional zero-shot benchmarks from existing datasets and our own PVD (§A.1) to address evaluation gaps left by standard benchmarks.

Street [110], which consists of images of under-represented populations. We use the version provided by [38] and re-evaluate all models to ensure a fair comparison. Here too we find $PE_{core}G$ to outperform existing methods, with +3.0% over SigLIP2-g-opt. Further, we test OCR performance by setting up TextCaps [119] as a retrieval dataset. Notably, $PE_{core}$ performs on par or better than SigLIP, which is known for good OCR performance. This is potentially surprising, as the horizontal flip augmentation we used during robust pretraining (§2.1) is typically thought to hurt OCR performance. However, instead it seems to have given $PE_{core}$ the ability to read backwards: we test the same TextCaps retrieval but with all images horizontally flipped. Other models suffer from this, but $PE_{core}G$'s performance only drops by 0.1%. Finally, we evaluate $PE_{core}G$ on the PVD benchmark (§A.1), a challenging video retrieval task on 15K diverse and human-refined videos. Here, $PE_{core}G$ significantly outperforms InternVL [19] by +13.6% on text→video and +9.5% to SigLIP2 [135] on video→text.

**Frozen Encoder Probing Results.** To compare against models that are not capable of zero-shot classification, we additionally evaluate $PE_{core}$ using k nearest neighbors (following [96]), linear probing (following [19]), and attention probing (following [36]) on top of the ImageNet-1k [25] train set. We present these results in Tab. 26 and compare to other encoders using their reported numbers. In every case, $PE_{core}G$ outperforms all existing open encoders, including those with significantly more parameters.

| Model | Encoder Params | Resolution | Data | *Encoder Probing* | | |
| | | | | ImageNet [25] *KNN* | ImageNet [25] *Linear* | ImageNet [25] *Attention* |
|---|---|---|---|---|---|---|
| DINOv2-g [96] | 1.1B | 224 | 145M | 83.5 | 86.5 | 87.2[†] |
| RADIOv2.5-g [44] | 1.1B | 518 | - | | 85.3 | - |
| AIMv2 3B [36] | 2.7B | 448 | 7.2B | - | - | 89.5 |
| InternVL-C [19] | 5.5B | 224 | 5B | - | 88.2 | - |
| EVA 18B [127] | 17.5B | 224 | 2B | - | 88.9 | - |
| **$PE_{core}G$** | 1.9B | 448 | 5.4B | **86.8** | **89.5** | **89.8** |

Table 26: **Encoder Probing Results.** We evaluate $PE_{core}G$'s frozen features using the typical probing methods to compare to models without zero-shot support. [†]from [36].

## C.5 Additional Layerwise Scaling Analysis

In the main paper, we explored the scalability of our pretriaining recipe v.s. the original CLIP recipe. However, we only analyzed it there for a single spatial task. To see whether the trend is consistent, we repeat this scaling analysis on a wide variety of downstream language modeling tasks using the same frozen evaluation setup as Fig. 4 and report the results in Fig. 17. Surprisingly, the simple change in pretraining recipe improves scaling for most language tasks as well—including output-side grounding (RefCOCO). Note that in this benchmarking setup, the LLM never sees videos during training so the Video Q&A per-layer results are noisy. Yet, the best layer trend is still the same.

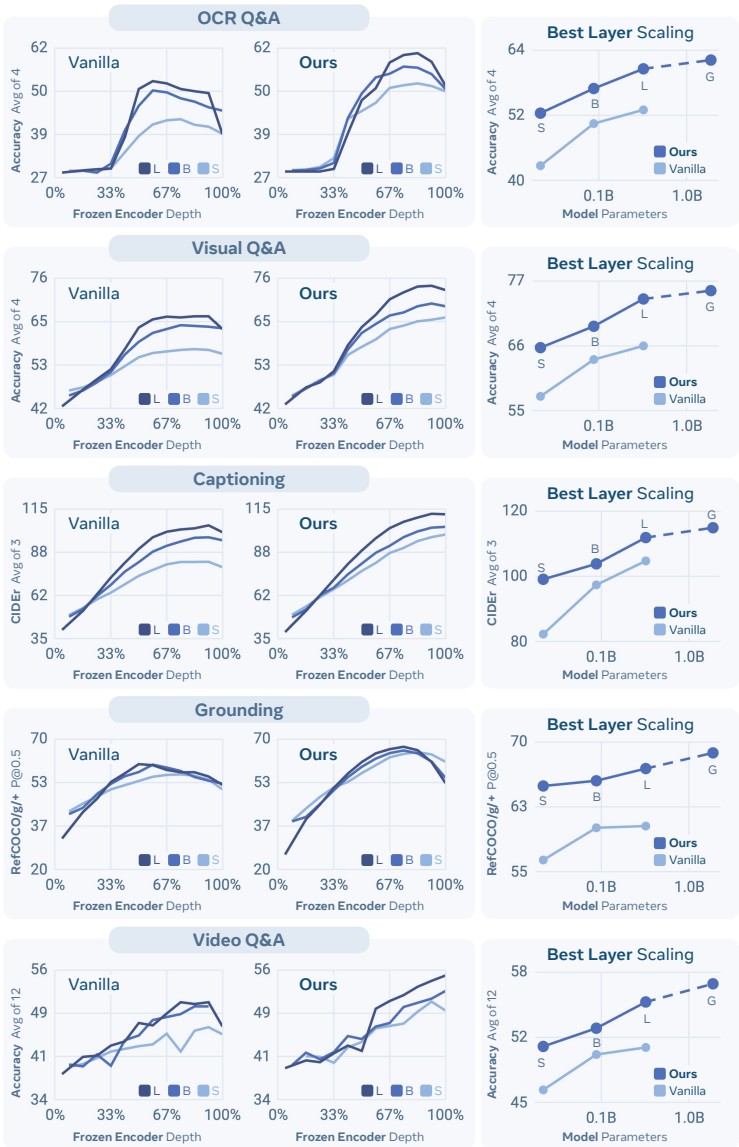

Figure 17: **Further Scalability Analysis.** We repeat the analysis from Fig. 7 on a wide range of downstream tasks by adapting to a language model. Each category is an average of several downstream tasks (see §4).

# D   Additional $PE_{lang}$ Results

## D.1   Alignment Method Derivation

Here we discuss in detail the derivation of the alignment approach discussed in §4. We design the alignment approach not just keeping in midn the best performance, but also ensuring that the resulting aligned model is *general*. That is, we want the aligned model to be transferrable to as many settings as possible regardless of resolution, decoder, or training setup.

**MLLM Evaluation Tasks.** In this section, our main testbed is to adapt vision encoders to MLLMs and test on various MLLM tasks. We evaluate the downstream performance of each MLLM across five task categories: (1) *OCR, Chart, Document Q&A* on ChartQA [162], DocVQA [89], InfoVQA [90] and AI2D [56]; (2) *Visual Q&A* on TextVQA [122], OK-VQA [115], POPE [72], and VQAv2 [39]; (3) *Captioning* on Flicker [155], COCO [74], and No Cap [1]; (4) *Video Understanding* on VideoMME [37], STAR [145], TGIF-QA [52], EgoSchema [87], MVBench [67], and PerceptionTest [102]; and finally (5) *Grounding* on RefCOCO [55].

To arrive at the optimal alignment recipe, we first conduct ablation studies using a 20M subset of the data. In Tab. 27, we ablate the LLM sizes, training parameters, vision projector types, output layers to project, and encoder regularization. We evaluate across OCR Q&A, Captioning, Visual Q&A, and Video Q&A and find the best configuration.

**LLM Setup.** We explore different *scales* (1B or 3B parameters) and *freezing* weights of the LLM. We observe that going from 1B to 3B parameters increases average score by 1.6 points (76.5→78.1). Unfreezing the LLM boosts this number to 78.4.

**Vision Projector.** Using a *2-layer MLP* vision projector instead of a *linear layer* improves the average score from 77.2 to 78.1, while only adding few parameters (13.5M → 27M).

**PE Output Layer.** As shown in §3, $PE_{core}G$ has intermediate layers that perform significantly better than the last layer when used as features for certain tasks. However, it is not clear if that same behavior applies when finetuning. We test applying the projector to layers 41, 47, and 50 (the last layer), and find that layer 47 works best. Incidentally, this is also the optimal layer for frozen VQ&A in Fig. 4.

| LLM scale | LLM unfrozen | Regularization? | Projector | Layer | Avg. | OCR Q&A *Average of 4* | Captioning *Average of 3* | Visual Q&A *Average of 4* | Video Q&A *Average of 6* |
|---|---|---|---|---|---|---|---|---|---|
| *LLM Setup* | | | | | | | | | |
| 1B | | | MLP | 47 | 76.5 | 60.7 | 115.1 | 76.0 | 54.0 |
| 3B | | | MLP | 47 | 78.1 | 65.9 | 115.7 | 76.6 | 54.1 |
| 3B | ✓ | | MLP | 47 | 78.4 | 65.8 | 117.6 | 76.3 | 53.7 |
| *Vision Projector* | | | | | | | | | |
| 3B | | | Linear | 47 | 77.2 | 64.5 | 114.1 | 76.5 | 53.7 |
| 3B | | | MLP | 47 | 78.1 | 65.9 | 115.7 | 76.6 | 54.1 |
| *PE Output Layer* | | | | | | | | | |
| 3B | | | MLP | 50 | 75.9 | 56.6 | 116.7 | 76.5 | 53.7 |
| 3B | | | MLP | 47 | 78.1 | 65.9 | 115.7 | 76.6 | 54.1 |
| 3B | | | MLP | 41 | 76.9 | 65.5 | 112.8 | 75.4 | 53.9 |
| *PE Regularization* | | | | | | | | | |
| 3B | | ✓ | MLP | 47 | 79.9 | 69.0 | 117.5 | 77.4 | 55.6 |
| 3B | ✓ | ✓ | MLP | 47 | **80.1** | 68.7 | 118.3 | 77.0 | 56.3 |

Table 27: **Language Alignment.** We find the best configuration to language align $PE_{core}G$ using autoregressive language training.

**PE Regularization.** We apply LayerScale [132] and DropPath [49] to the vision encoder during the alignment, for stabilizing training. This improves the 78.1 average score to 79.9 (+1.8 points). Unfreezing the LLM boosts this number further to 80.1. We choose this configuration (last row) as our final alignment setup.

To construct $PE_{lang}$, we scale this recipe up the 70M samples covering natural images, documents/charts/diagrams, and videos, perform alignment as described, and extract the resulting vision encoder. Compared to the 20M sample ablation setting in Tab. 27, the final $PE_{lang}$ trained on 70M total samples gives another +2.1 points to 82.2 on the average across OCR Q&A, Captioning, Visual Q&A, and Video Q&A.

## D.2   Layer Analysis Details

In Tab. 28, we present the raw numbers for the layer analysis plots in Fig. 8. Note that layer analysis was not performed exhaustively on every layer. Additionally, $PE_{lang}$ removes the last four layers of the model.

| Layer | OCR VQA | | Captioning | | Natural VQA | | Grounding | |
|---|---|---|---|---|---|---|---|---|
| | PE$_{core}$ | PE$_{lang}$ | PE$_{core}$ | PE$_{lang}$ | PE$_{core}$ | PE$_{lang}$ | PE$_{core}$ | PE$_{lang}$ |
| 50 | 48.6 | | 114.0 | | 74.1 | | 39.1 | |
| 49 | 53.8 | | 114.5 | | 74.9 | | 54.2 | |
| 48 | 56.3 | | 114.8 | | 74.8 | | 55.6 | |
| 47 | 57.3 | 72.4 | 113.9 | 120.1 | 75.3 | 78.1 | 57.8 | 71.2 |
| 46 | 58.3 | 72.2 | 114.1 | 120.0 | 75.1 | 78.2 | 58.4 | 70.8 |
| 45 | 59.6 | 72.0 | 114.6 | 119.7 | 75.1 | 77.7 | 61.2 | 70.7 |
| 43 | 59.9 | 71.5 | 113.8 | 117.7 | 74.8 | 77.4 | 63.1 | 70.5 |
| 42 | 60.3 | 71.3 | 113.1 | 116.6 | 73.8 | 77.2 | 64.1 | 70.8 |
| 41 | 60.8 | 70.7 | 112.5 | 115.4 | 73.3 | 76.7 | 66.6 | 71.2 |
| 40 | 61.4 | 69.8 | 112.0 | 115.0 | 74.0 | 76.3 | 66.5 | 71.0 |
| 39 | 61.8 | 70.0 | 111.0 | 113.5 | 74.0 | 75.6 | 67.1 | 71.1 |
| 38 | 62.1 | 69.3 | 110.2 | 112.6 | 73.9 | 75.3 | 68.7 | 70.3 |
| 36 | 61.3 | 67.9 | 108.5 | 109.4 | 73.0 | 74.0 | 67.8 | 70.4 |
| 34 | 58.7 | 65.4 | 102.8 | 104.7 | 70.0 | 72.2 | 66.5 | 69.6 |
| 33 | 57.3 | 64.1 | 100.1 | 102.7 | 69.0 | 71.3 | 65.9 | 68.9 |
| 32 | 54.2 | 63.2 | 96.9 | 100.1 | 67.8 | 70.6 | 65.5 | 68.7 |
| 31 | 50.9 | 60.7 | 93.2 | 96.7 | 65.5 | 68.7 | 63.0 | 67.3 |
| 21 | 29.6 | 30.3 | 59.3 | 71.3 | 49.1 | 52.0 | 41.5 | 51.6 |
| 11 | 28.8 | 28.9 | 47.4 | 59.0 | 47.0 | 49.2 | 30.9 | 43.3 |
| 2 | 28.2 | 28.7 | 38.6 | 42.8 | 43.3 | 43.8 | 22.8 | 27.2 |

Table 28: **Raw Language Layer Analysis Results.** The raw values for the plots in Fig. 8.

## D.3 Unfrozen Encoder Results

In our standard MLLM evaluation, we always freeze the vision encoder when tuning the LLM for downstream MLLM tasks. This is to ensure that we test the quality of each vision encoder without any bias from our finetuning setup. However, this introduces a lingering question of whether unlocking the encoder during LLM finetuning would eliminate any lead PE$_{lang}$ has over the other models.

Thus, in this section, we repeat the same MLLM evaluations as the main paper but with the encoder unfrozen. Each experiment uses 1024 tokens per image. In Tab. 29, we show the unfrozen encoder results compared to AIMv2 3B [36] and SiglIP2 g-opt [135]. It seems all models, including PE$_{lang}$G, benefit from unlocking the encoder. However, PE$_{lang}$G still outperforms the other models overall, often by a significant margin.

We perform similar evaluation in Tab. 30, this time comparing across PE$_{lang}$ model scales and to the original PE$_{core}$. And here we see that both PE$_{lang}$ models significantly outperform the PE$_{core}$ ones in this unfrozen setup, especially for the larger G size. Thus, it seems that a language alignment step is still necessary even when the encoder is unfrozen during MLLM construction.

| Model | OCR / Chart / Doc. Q&A | | | | | Visual Q&A | | | | | Captioning | | | | Avg. Ground. RefCOCO/g/+ [55] | Video | | | | | | |
|---|---|---|---|---|---|---|---|---|---|---|---|---|---|---|---|---|---|---|---|---|---|---|
| | Avg. OCR QA | ChartQA Acc. [162] | DocVQA Acc. [89] | Info. QA Acc. [90] | AI2D Acc. [56] | Avg. VQA | TextVQA Acc. [122] | OK-VQA Acc. [115] | POPE Acc. [72] | VQAv2 Acc. [39] | Avg. Cap. | Flickr CIDEr [155] | COCO CIDEr [74] | No Cap CIDEr [1] | | Avg. Video | VideoMME Acc. [37] | STAR Acc. [145] | TGIF-QA Acc. [52] | EgoSchema Acc. [87] | MVBench Acc. [67] | PerceptionTest Acc. [102] |
| AIMv2 3B [36] | 65.0 | 77.4 | 68.8 | 39.4 | 74.4 | 76.9 | 73.1 | 64.4 | 88.4 | 81.7 | 119.2 | 97.1 | 139.7 | 120.7 | 69.7 | 54.3 | 46.8 | 53.2 | 64.5 | 58.4 | 48.9 | 53.9 |
| SigLIP2-g-opt [135] | 65.0 | 77.3 | 68.6 | 39.6 | 74.6 | **78.8** | 74.9 | 67.7 | 89.5 | **83.2** | 120.0 | **97.9** | 140.7 | 121.5 | **73.4** | 54.4 | 43.8 | 52.7 | 66.8 | 59.0 | **51.2** | 52.9 |
| **PE$_{lang}$G** | **72.8** | **81.6** | **84.7** | **48.1** | **76.8** | 78.4 | 74.1 | 66.9 | **89.6** | 82.9 | **120.8** | 97.4 | **141.7** | **123.4** | **73.4** | **57.9** | **48.0** | **58.9** | **70.4** | **63.0** | 50.9 | **56.2** |

Table 29: **MLLM Results with Encoder Unfrozen (PE$_{lang}$ vs. Others).** Same setting as Tab. 5 using 1024 tokens per image, but with the vision encoder unfrozen during LLM finetuning.

| Model | OCR / Chart / Doc. Q&A | | | | | Visual Q&A | | | | | Captioning | | | | Avg. Ground. RefCOCO/g/+ [55] | Video | | | | | | |
|---|---|---|---|---|---|---|---|---|---|---|---|---|---|---|---|---|---|---|---|---|---|---|
| | Avg. OCR QA | ChartQA Acc. [162] | DocVQA Acc. [89] | Info. QA Acc. [90] | AI2D Acc. [56] | Avg. VQA | TextVQA Acc. [122] | OK-VQA Acc. [115] | POPE Acc. [72] | VQAv2 Acc. [39] | Avg. Cap. | Flickr CIDEr [155] | COCO CIDEr [74] | No Cap CIDEr [1] | | Avg. Video | VideoMME Acc. [37] | STAR Acc. [145] | TGIF-QA Acc. [52] | EgoSchema Acc. [87] | MVBench Acc. [67] | PerceptionTest Acc. [102] |
| **PE$_{core}$L** | 64.4 | 76.9 | 70.6 | 39.1 | 70.8 | 75.8 | 69.7 | **64.5** | 88.5 | 80.3 | **115.9** | **93.7** | 136.8 | 117.1 | 70.9 | 53.3 | 45.3 | 52.6 | 65.4 | 53.4 | **50.3** | 52.8 |
| **PE$_{lang}$L** | 71.2 | 81.2 | 82.5 | 45.6 | 75.3 | 76.6 | 71.7 | 64.4 | 89.2 | 80.9 | 114.5 | 85.1 | 138.7 | 119.7 | 73.0 | 55.5 | 47.1 | 55.1 | 68.4 | 58.4 | 49.9 | 54.3 |
| **PE$_{core}$G** | 62.8 | 73.6 | 67.2 | 38.5 | 71.7 | 76.0 | 68.9 | 65.6 | 88.4 | 80.9 | 117.2 | 94.0 | 137.9 | 119.7 | 69.2 | 55.0 | 44.2 | 56.0 | 66.3 | 60.6 | 50.5 | 52.5 |
| **PE$_{lang}$G** | 72.8 | 81.6 | 84.7 | 48.1 | 76.8 | 78.4 | 74.1 | 66.9 | 89.6 | 82.9 | 120.8 | 97.4 | 141.7 | 123.4 | 73.4 | 57.9 | 48.0 | 58.9 | 70.4 | 63.0 | 50.9 | 56.2 |

Table 30: **MLLM Results with Encoder Unfrozen (Core vs. Lang).** Same setting as Tab. 5 using 1024 tokens per image, but with the vision encoder unfrozen during LLM finetuning.

## D.4 Additional Results

Here we provide additional comparisons of $PE_{core}$ and $PE_{lang}$ with other vision encoders that are popular choices in MLLM literature: MetaCLIP [150], SigLIP2 [135], CLIP [103], AIMv2 [36], DINOv2 [96], and InternViT2.5 [18]. Overall, these encoders span several different pretraining losses (e.g., contrastive, captioning, self-supervised, and mixed supervision), encoder sizes (from 300M to 6B parameters), and resolutions (from 224 to 512). *For all vision encoders, we find the best intermediate layers to train MLLM for fair comparison* (more details in Appendix B.2).

**Main Results.** In Tab. 5, we showed benchmarks results for native resolution input across existing encoders, $PE_{core}$ and $PE_{lang}$. Here we provide additional comments about those results. Notably, AIMv2 [36], InternViT2.5 [18], SigLIP2 [135] and $PE_{lang}$ are trained jointly with a language decoder using next token prediction objective, and thus they perform better overall compared to the base contrastive and self-supervised models across all the metrics. However, $PE_{lang}$ uses a fraction of the training FLOPs for language alignment tuning, while significantly outperforming all vision encoders by large margin (an average of +3.5 points for G and +2.0 points for L).

In Tab. 31, we compare $PE_{core}$ and $PE_{lang}$ with *dynamic resolution* setting [75, 80]. More specifically, we use up to 4 tiles, following after a *thumbnail*, which is a whole image resized into $448 \times 448$. With the maximum number of tiles of 4, the model can cover $\{1 \times 1, 1 \times 2, 1 \times 3, 1 \times 4, 2 \times 1, 2 \times 2, 3 \times 1, 4 \times 1\}$ tile ratios. Similar to the Tab. 5, we show that $PE_{lang}$ largely outperforms the baseline vision encoders by large margins across all categories of MLLM tasks. Note that $PE_{lang}$ has been alignment-tuned with native resolution input, as opposed to *e.g.*, InternViT 2.5, which has been midtrained with dynamic tiling, which shows $PE_{lang}$'s strong generality for different input formats.

**Transferability.** As $PE_{lang}$ is aligned with Llama 3.2-instruct 3B, we conduct a separate set of experiments to check if our model performs well with a different base LLM. In Tab. 32 we repeat the native resolution comparison with QwenLM 2.5 7B [153]. Interestingly, $PE_{lang}$ not only outperforms all vision encoders in this setting, but it also outperforms InternViT2.5 [18], which is specifically aligned to QwenLM 2 [152] throughout midtraining. In fact, $PE_{lang}$G with QwenLM even improves its performance with Llama in some cases like with OCR Q&A and video benchmarks, emphasizing the generality of our language alignment.

**Grounding Breakdown.** Next, in Tab. 33, 34, 35, we show full RefCOCO/+/g [55] results across all setups. Overall, $PE_{lang}$ L or G show the best performance across all RefCOCO splits, except with Qwen2.5 LM. This is because (1) InternViT 2.5 6B is midtrained with Qwen2 LM, and (2) during pre/mid-training the training data of RefCOCO/+/g are seen.

**Table 31:** **4+1 Tile Llama 8B MLLM Results.**

| Model | Encoder Params | Resolution Patch Size | OCR / Chart / Doc. Q&A | | | | | Visual Q&A | | | | | Captioning | | | | Avg. Ground. RefCOCO/g/+ [55] | Avg. Video | Video | | | | | |
| | | | Avg. OCR QA | ChartQA [162] | DocVQA [89] | Info. QA [90] | AI2D [56] | Avg. VQA | TextVQA [122] | OK-VQA [115] | POPE [72] | VQAv2 [39] | Avg. Cap. | Flicker [155] | COCO [74] | No Cap [1] | | | VideoMME [37] | STAR [145] | TGIF-QA [52] | EgoSchema [87] | MVBench [67] | PerceptionTest [102] |
|---|---|---|---|---|---|---|---|---|---|---|---|---|---|---|---|---|---|---|---|---|---|---|---|---|
| *256 Tokens per Tile* | | | | | | | | | | | | | | | | | | | | | | | | |
| MetaCLIP-L [150] | 0.3B | 224/14 | 61.8 | 71.1 | 62.5 | 40.2 | 73.3 | 74.6 | 65.3 | 64.9 | 88.5 | 79.8 | 113.4 | 90.4 | 133.5 | 116.2 | 67.1 | 48.0 | 44.8 | 47.1 | 62.7 | 39.0 | 46.0 | 48.3 |
| MetaCLIP-G [150] | 1.8B | 224/14 | 60.3 | 68.1 | 61.3 | 39.1 | 72.8 | 74.9 | 65.4 | 65.9 | 88.2 | 80.1 | 114.2 | 91.8 | 134.4 | 116.5 | 66.0 | 49.0 | 46.5 | 46.5 | 62.5 | 45.0 | 44.7 | 48.9 |
| PE$_\text{lang}$ G† | 1.7B* | 224/14 | 70.2 | 79.8 | 79.1 | 47.5 | 74.6 | 76.0 | 70.6 | 64.3 | 88.3 | 80.6 | 116.3 | 92.0 | 136.4 | 120.5 | 69.5 | 56.6 | 49.0 | 55.9 | 69.9 | 61.2 | 50.0 | 53.6 |
| *576 Tokens per Tile* | | | | | | | | | | | | | | | | | | | | | | | | |
| CLIP [103] | 0.3B | 336/14 | 69.6 | 76.8 | 78.2 | 50.3 | 72.9 | 76.3 | 71.8 | 64.9 | 88.0 | 80.4 | 114.0 | 90.9 | 134.4 | 116.6 | 68.5 | 50.8 | 46.6 | 52.2 | 65.0 | 44.6 | 46.3 | 49.9 |
| AIMv2-L [36] | 0.3B | 336/14 | 66.7 | 74.1 | 74.9 | 45.2 | 72.4 | 77.4 | 73.5 | 65.6 | 89.0 | 81.7 | 116.4 | 92.5 | 137.1 | 119.5 | 66.6 | 54.1 | 43.4 | 54.3 | 70.6 | 56.0 | 47.3 | 52.7 |
| SigLIP2-so [135] | 0.4B | 384/16 | 55.5 | 61.4 | 54.9 | 33.3 | 72.3 | 76.5 | 70.1 | 66.0 | 88.6 | 81.2 | 118.0 | 95.8 | 138.3 | 119.8 | 66.5 | 54.3 | 44.9 | 52.8 | 66.8 | 58.6 | 49.6 | 53.3 |
| SigLIP2-g-opt [135] | 1.1B | 384/16 | 56.2 | 63.1 | 55.3 | 34.0 | 72.4 | 77.0 | 70.3 | 66.7 | 89.6 | 81.6 | 117.7 | 94.9 | 137.8 | 120.3 | 66.5 | 53.9 | 46.2 | 53.9 | 66.6 | 53.8 | 48.5 | 54.7 |
| PE$_\text{lang}$ G† | 1.7B* | 336/14 | 77.5 | 82.1 | 88.5 | 61.8 | 77.4 | 79.7 | 80.2 | 66.4 | 89.8 | 82.5 | 120.3 | 97.4 | 140.2 | 123.2 | 71.9 | 59.8 | 49.4 | 62.7 | 74.1 | 64.0 | 53.1 | 55.6 |
| *1024 Tokens per Tile* | | | | | | | | | | | | | | | | | | | | | | | | |
| SigLIP2-so [135] | 0.4B | 512/16 | 56.9 | 66.0 | 56.5 | 34.3 | 70.9 | 76.4 | 69.9 | 66.2 | 88.4 | 81.2 | 117.8 | 94.7 | 137.8 | 120.9 | 67.8 | 46.2 | 47.0 | 44.9 | 66.7 | 39.2 | 34.5 | 45.1 |
| PE$_\text{core}$ L | 0.3B | 448/14 | 67.1 | 72.4 | 78.3 | 46.4 | 71.2 | 76.4 | 74.0 | 63.7 | 88.8 | 79.0 | 113.9 | 91.5 | 134.5 | 115.7 | 62.9 | 51.4 | 47.0 | 51.2 | 62.7 | 49.6 | 47.8 | 50.1 |
| PE$_\text{lang}$ L | 0.3B | 448/14 | 78.3 | 82.8 | 89.3 | 65.2 | 75.9 | 78.5 | 78.8 | 64.4 | 89.6 | 81.3 | 117.8 | 94.7 | 138.1 | 120.7 | 71.6 | 56.5 | 47.0 | 57.2 | 68.0 | 59.8 | 52.3 | 54.7 |
| AIMv2 3B [36] | 2.7B | 448/14 | 67.5 | 73.0 | 78.2 | 46.5 | 72.2 | 78.8 | 79.2 | 66.2 | 88.3 | 81.7 | 119.0 | 95.8 | 139.7 | 121.5 | 65.1 | 54.0 | 49.6 | 55.4 | 67.3 | 49.6 | 49.9 | 52.5 |
| InternViT2.5 6B [18] | 5.5B | 448/14 | 67.4 | 74.6 | 74.3 | 47.6 | 72.9 | 75.9 | 71.3 | 64.8 | 87.7 | 79.7 | 110.4 | 85.3 | 132.5 | 113.5 | 56.8 | 52.0 | 46.0 | 49.6 | 65.0 | 50.6 | 49.6 | 51.3 |
| PE$_\text{core}$ G | 1.9B | 448/14 | 68.0 | 73.4 | 81.2 | 47.6 | 69.7 | 76.4 | 74.3 | 62.5 | 89.1 | 79.6 | 113.0 | 91.6 | 134.5 | 112.9 | 67.6 | 53.2 | 46.0 | 54.3 | 67.0 | 51.2 | 48.7 | 52.0 |
| PE$_\text{lang}$ G | 1.7B* | 448/14 | 78.6 | 81.8 | 89.8 | 67.8 | 75.0 | 80.3 | 82.3 | 66.7 | 89.6 | 82.8 | 119.6 | 95.2 | 140.3 | 123.4 | 71.8 | 59.0 | 49.6 | 61.8 | 73.9 | 60.0 | 52.6 | 56.3 |

Table 31: **4+1 Tile Llama 8B MLLM Results.** Llama 3.1-instruct 8B [80] is used as a language model. *PE$_\text{lang}$ has 1.7B parameters since we discard the last 3 layers during language alignment. All MLLMs are trained with dynamic tiling for different image sizes and aspect ratio. We use up to 4 image tiles of $448 \times 448$ (or the corresponding resolution for each encoder). The image tiles follow after a *thumbnail* input, similar to prior work [75]. †Evaluation on an model that was interpolated without additional training (i.e., *zero-shot* resolution).

| Model | Encoder Params | Resolution Patch Size | OCR / Chart / Doc. Q&A | | | | | Visual Q&A | | | | | Captioning | | | | Avg. Ground. RefCOCO/g/+ [55] | Avg. Video | Video | | | | | |
| | | | Avg. OCR QA | ChartQA [162] | DocVQA [89] | Info. QA [90] | AI2D [56] | Avg. VQA | TextVQA [122] | OK-VQA [115] | POPE [72] | VQAv2 [39] | Avg. Cap. | Flicker [155] | COCO [74] | No Cap [1] | | | VideoMME [37] | STAR [145] | TGIF-QA [52] | EgoSchema [87] | MVBench [67] | PerceptionTest [102] |
|---|---|---|---|---|---|---|---|---|---|---|---|---|---|---|---|---|---|---|---|---|---|---|---|---|
| *576 Tokens per Image* | | | | | | | | | | | | | | | | | | | | | | | | |
| SigLIP2-so [135] | 0.4B | 384/16 | 60.5 | 72.0 | 59.1 | 36.7 | 74.3 | 76.2 | 69.0 | 65.4 | 89.2 | 81.1 | 116.3 | 91.6 | 137.3 | 120.0 | 70.0 | 57.0 | 51.3 | 55.8 | 66.0 | 61.0 | 51.9 | 55.7 |
| SigLIP2-g-opt [135] | 1.1B | 384/16 | 60.8 | 71.0 | 60.4 | 36.7 | 75.2 | 76.8 | 70.3 | 65.6 | 89.5 | 81.8 | 118.8 | 96.4 | 139.0 | 121.1 | 69.9 | 58.3 | 52.0 | 57.6 | 68.1 | 62.0 | 52.8 | 57.4 |
| PE$_\text{lang}$ G† | 1.7B* | 336/14 | 66.8 | 77.5 | 72.4 | 41.1 | 76.4 | 76.0 | 67.9 | 65.4 | 89.1 | 81.5 | 118.8 | 94.6 | 139.5 | 122.3 | 70.1 | 60.2 | 54.6 | 61.7 | 69.8 | 63.6 | 54.3 | 57.2 |
| *1024 Tokens per Image* | | | | | | | | | | | | | | | | | | | | | | | | |
| InternViT2.5 [18] | 0.3B | 448/14 | 60.3 | 75.4 | 61.1 | 36.2 | 68.4 | 74.2 | 65.6 | 63.7 | 87.8 | 79.5 | 112.1 | 88.5 | 133.5 | 114.1 | 68.1 | 55.8 | 50.3 | 54.7 | 66.6 | 59.0 | 50.6 | 53.8 |
| SigLIP2-so [135] | 0.4B | 512/16 | 66.3 | 77.2 | 71.9 | 42.4 | 73.9 | 77.9 | 74.2 | 65.6 | 89.9 | 81.8 | 117.1 | 93.0 | 138.0 | 120.3 | 70.5 | 55.9 | 50.3 | 57.3 | 67.2 | 62.6 | 50.3 | 47.4 |
| PE$_\text{core}$ L | 0.3B | 448/14 | 63.5 | 73.9 | 67.4 | 40.5 | 72.2 | 75.7 | 69.2 | 64.0 | 89.4 | 80.2 | 113.3 | 88.7 | 135.2 | 115.9 | 66.5 | 57.3 | 49.6 | 57.8 | 67.7 | 60.8 | 52.3 | 55.5 |
| PE$_\text{lang}$ L | 0.3B | 448/14 | 70.2 | 80.6 | 80.7 | 46.0 | 73.5 | 76.8 | 72.8 | 64.1 | 89.4 | 81.0 | 116.4 | 93.4 | 137.6 | 118.1 | 70.4 | 58.3 | 51.6 | 59.8 | 67.4 | 62.2 | 53.4 | 55.4 |
| DINOv2 [96] | 1.1B | 448/14 | 31.3 | 21.7 | 14.7 | 24.6 | 64.3 | 61.0 | 18.9 | 59.5 | 88.9 | 76.9 | 110.1 | 87.3 | 132.1 | 110.8 | 69.3 | 54.3 | 46.9 | 56.5 | 63.4 | 56.8 | 49.7 | 52.2 |
| AIMv2 3B [36] | 2.7B | 448/14 | 66.0 | 76.7 | 70.5 | 41.4 | 75.2 | 77.9 | 74.2 | 66.2 | 89.3 | 81.9 | 119.2 | 96.4 | 139.2 | 122.0 | 67.6 | 56.3 | 45.9 | 58.0 | 67.8 | 60.8 | 51.4 | 53.9 |
| InternViT2.5 [18] | 5.5B | 448/14 | 64.2 | 78.2 | 65.3 | 39.6 | 73.6 | 76.4 | 70.1 | 64.5 | 89.3 | 81.7 | 117.6 | 95.9 | 138.4 | 118.6 | 72.8 | 56.1 | 50.3 | 59.1 | 67.3 | 56.6 | 51.1 | 52.2 |
| PE$_\text{core}$ G | 1.9B | 448/14 | 64.8 | 75.9 | 68.8 | 41.6 | 72.9 | 75.2 | 67.9 | 62.4 | 89.7 | 80.7 | 113.1 | 91.7 | 135.2 | 112.3 | 70.5 | 57.0 | 48.7 | 58.3 | 66.9 | 60.8 | 52.9 | 54.5 |
| PE$_\text{lang}$ G | 1.7B* | 448/14 | 72.9 | 81.6 | 83.7 | 49.5 | 76.7 | 77.9 | 74.9 | 64.5 | 90.3 | 81.9 | 118.9 | 94.6 | 139.8 | 122.3 | 72.1 | 60.4 | 54.1 | 62.5 | 68.3 | 66.6 | 54.2 | 56.8 |

Table 32: **MLLM Results with QwenLM 2.5 7B.** Same setting as Tab. 5, but with QwenLM2.5 7B [153] as the language model. Although PE$_\text{lang}$ is aligned to Llama3.2 3B, the language alignment transfers well to a different language model.

| Model | Encoder Params | Resolution Patch Size | Avg. Ground. | Grounding | | | | | | | |
|---|---|---|---|---|---|---|---|---|---|---|---|
| | | | | RefCOCO val[55] | RefCOCO testA[55] | RefCOCO testB[55] | RefCOCO+ val[55] | RefCOCO+ testA[55] | RefCOCO+ testB[55] | RefCOCOg val[55] | RefCOCOg test[55] |
| *576 Tokens per Image* | | | | | | | | | | | |
| CLIP [103] | 0.3B | 336/14 | 65.0 | 66.7 | 61.4 | 71.6 | 57.6 | 62.5 | 54.5 | 73.2 | 72.8 |
| AIMv2-L [36] | 0.3B | 336/14 | 63.3 | 65.4 | 61.6 | 69.6 | 55.0 | 60.0 | 52.0 | 71.1 | 71.5 |
| AIMv2-L Dist. [36] | 0.3B | 336/14 | 62.6 | 64.8 | 61.0 | 69.4 | 54.4 | 59.0 | 51.3 | 70.8 | 70.0 |
| SigLIP2-so [135] | 0.4B | 384/16 | 67.4 | 68.8 | 66.5 | 71.0 | 60.3 | 61.8 | 58.5 | 76.2 | 76.0 |
| SigLIP2-g-opt [135] | 1.1B | 384/16 | 66.5 | 67.9 | 66.1 | 70.1 | 58.8 | 61.7 | 57.1 | 75.5 | 75.0 |
| **PE$_{lang}$ G**† | 1.7B* | 336/14 | 68.9 | 69.8 | 67.5 | 73.2 | 61.5 | 64.0 | 60.8 | 77.3 | 77.7 |
| *1024 Tokens per Image* | | | | | | | | | | | |
| InternViT2.5 L [18] | 0.3B | 448/14 | 66.9 | 69.3 | 66.7 | 72.6 | 58.3 | 63.1 | 57.2 | 74.2 | 74.0 |
| SigLIP2-so [135] | 0.4B | 512/16 | 69.6 | 71.4 | 69.2 | 74.4 | 61.3 | 64.8 | 60.3 | 77.9 | 77.2 |
| **PE$_{core}$ L** | 0.3B | 448/14 | 59.7 | 61.7 | 55.3 | 66.9 | 53.1 | 58.8 | 48.0 | 68.5 | 67.5 |
| **PE$_{lang}$ L** | 0.3B | 448/14 | 70.5 | 71.8 | **70.2** | 73.0 | 63.7 | 66.1 | 62.7 | 78.8 | 78.9 |
| DINOv2 [96] | 1.1B | 448/14 | 64.9 | 67.2 | 62.5 | 70.5 | 57.0 | 61.0 | 54.5 | 73.1 | 73.1 |
| AIMv2 3B [36] | 2.7B | 448/14 | 36.1 | 37.6 | 34.1 | 40.7 | 32.7 | 36.2 | 32.0 | 36.9 | 38.6 |
| InternViT2.5 6B [18] | 5.5B | 448/14 | 68.0 | 70.2 | 67.6 | 72.2 | 60.6 | 64.0 | 58.7 | 75.3 | 75.2 |
| **PE$_{core}$ G** | 1.9B | 448/14 | 66.6 | 68.3 | 64.4 | 72.3 | 58.7 | 62.7 | 56.0 | 75.1 | 75.0 |
| **PE$_{lang}$ G** | 1.7B* | 448/14 | **71.3** | **71.9** | 69.9 | **75.1** | **64.2** | **67.3** | **63.0** | **79.4** | **79.2** |

Table 33: **Llama MLLM-Based Zeroshot RefCOCO.** With Llama 3.1-instruct 8B [80] as the LLM.

| Model | Encoder Params | Resolution Patch Size | Avg. Ground. | Grounding | | | | | | | |
|---|---|---|---|---|---|---|---|---|---|---|---|
| | | | | RefCOCO val[55] | RefCOCO testA[55] | RefCOCO testB[55] | RefCOCO+ val[55] | RefCOCO+ testA[55] | RefCOCO+ testB[55] | RefCOCOg val[55] | RefCOCOg test[55] |
| *576 Tokens per Image* | | | | | | | | | | | |
| SigLIP2-so [135] | 0.4B | 384/16 | 70.0 | 73.6 | 73.0 | 74.3 | 60.9 | 62.7 | 59.9 | 78.4 | 77.2 |
| SigLIP2-g-opt [135] | 1.1B | 384/16 | 69.9 | 73.3 | 72.4 | 73.6 | 60.5 | 62.3 | 60.7 | 78.4 | 78.2 |
| **PE$_{lang}$ G**† | 1.7B* | 336/14 | 70.1 | 73.4 | 72.0 | 75.3 | 62.0 | 64.2 | 61.2 | 78.4 | 77.7 |
| *1024 Tokens per Image* | | | | | | | | | | | |
| InternViT2.5 L [18] | 0.3B | 448/14 | 68.1 | 72.4 | 69.1 | 74.1 | 59.3 | 62.4 | 56.6 | 75.2 | 75.5 |
| SigLIP2-so [135] | 0.4B | 512/16 | 70.5 | 74.1 | 73.7 | 74.4 | 61.7 | 62.9 | 61.0 | 78.6 | 77.9 |
| **PE$_{core}$L** | 0.3B | 448/14 | 66.5 | 70.4 | 67.8 | 71.5 | 57.7 | 61.1 | 56.2 | 75.8 | 75.3 |
| **PE$_{lang}$L** | 0.3B | 448/14 | 70.4 | 74.4 | 72.6 | 74.6 | 62.2 | 64.0 | 62.0 | 79.0 | 78.7 |
| DINOv2 [96] | 1.1B | 448/14 | 69.3 | 73.4 | 71.1 | 73.9 | 60.0 | 63.9 | 59.0 | 76.4 | 76.7 |
| AIMv2 3B [36] | 2.7B | 448/14 | 67.6 | 71.4 | 67.7 | 72.3 | 59.2 | 61.2 | 56.3 | 76.4 | 76.4 |
| InternViT2.5 6B‡ [18] | 5.5B | 448/14 | **72.8** | **77.7** | **76.5** | **77.1** | 63.6 | **66.0** | 62.2 | **80.0** | 79.5 |
| **PE$_{core}$G** | 1.9B | 448/14 | 70.5 | 74.0 | 71.8 | 75.8 | 61.5 | 64.8 | 60.1 | 78.5 | 77.3 |
| **PE$_{lang}$G** | 1.7B* | 448/14 | 72.1 | 75.4 | 72.9 | 76.3 | **64.2** | 65.9 | **62.9** | 79.7 | **79.7** |

Table 34: **Qwen MLLM-Based Zeroshot RefCOCO.** With QwenLM 2.5 7B [153] as the LLM. All MLLMs report zeroshot results on RefCOCO/+/g datasets. ‡Trained with RefCOCO/+/g beforehand.

| Model | Encoder Params | Resolution Patch Size | Avg. Ground. | Grounding | | | | | | | |
|---|---|---|---|---|---|---|---|---|---|---|---|
| | | | | RefCOCO val[55] | RefCOCO testA[55] | RefCOCO testB[55] | RefCOCO+ val[55] | RefCOCO+ testA[55] | RefCOCO+ testB[55] | RefCOCOg val[55] | RefCOCOg test[55] |
| *256 Tokens per Tile* | | | | | | | | | | | |
| MetaCLIP-L [150] | 0.3B | 224/14 | 67.1 | 69.3 | 65.0 | 73.2 | 60.5 | 64.9 | 56.5 | 74.3 | 73.4 |
| MetaCLIP-G [150] | 1.8B | 224/14 | 66.0 | 67.9 | 63.2 | 71.9 | 59.2 | 62.9 | 55.8 | 73.8 | 73.1 |
| **PE$_{lang}$ G**† | 1.7B* | 224/14 | 70.3 | 71.6 | 69.6 | 73.7 | 63.3 | 66.2 | 62.6 | 78.6 | 78.2 |
| *576 Tokens per Tile* | | | | | | | | | | | |
| CLIP [103] | 0.3B | 336/14 | 68.5 | 70.7 | 66.6 | 74.1 | 61.1 | 65.9 | 58.1 | 76.0 | 75.1 |
| AIMv2-L [36] | 0.3B | 336/14 | 66.6 | 68.4 | 65.5 | 71.4 | 59.3 | 63.4 | 56.5 | 74.2 | 74.2 |
| SigLIP2-so [135] | 0.4B | 384/16 | 66.5 | 67.9 | 66.1 | 70.1 | 58.8 | 61.7 | 57.1 | 75.5 | 75.0 |
| SigLIP2-g-opt [135] | 1.1B | 384/16 | 66.5 | 68.2 | 65.6 | 70.1 | 59.0 | 62.3 | 58.0 | 74.8 | 74.0 |
| **PE$_{lang}$ G**† | 1.7B* | 336/14 | 71.9 | 73.6 | 71.5 | 74.9 | 64.8 | 67.3 | 63.9 | 80.4 | 80.6 |
| *1024 Tokens per Tile* | | | | | | | | | | | |
| SigLIP2-so [135] | 0.4B | 512/16 | 67.8 | 69.2 | 67.8 | 71.2 | 59.9 | 62.5 | 59.0 | 76.9 | 76.0 |
| **PE$_{core}$L** | 0.3B | 448/14 | 62.9 | 65.3 | 59.9 | 69.2 | 56.6 | 62.2 | 52.0 | 70.1 | 70.0 |
| **PE$_{lang}$L** | 0.3B | 448/14 | 71.6 | **73.0** | **70.8** | 74.3 | **65.2** | **67.2** | 62.9 | 79.7 | 79.7 |
| AIMv2 3B [36] | 2.7B | 448/14 | 65.1 | 66.9 | 62.9 | 71.1 | 58.1 | 62.4 | 55.6 | 71.8 | 72.2 |
| InternViT2.5 6B‡ [18] | 5.5B | 448/14 | 56.8 | 61.0 | 56.4 | 65.8 | 51.0 | 57.0 | 46.1 | 58.0 | 58.9 |
| **PE$_{core}$ G** | 1.9B | 448/14 | 67.6 | 69.2 | 65.8 | 72.4 | 59.9 | 64.1 | 58.3 | 75.1 | 75.6 |
| **PE$_{lang}$ G** | 1.7B* | 448/14 | **71.8** | 72.6 | 70.7 | **74.6** | 64.8 | 66.6 | **64.6** | 80.4 | 80.3 |

Table 35: **4+1 Tile Llama 8B MLLM-Based Zeroshot RefCOCO.** With Llama 3.1-instruct 8B [80] as the LLM. We use up to 4 image tiles of the encoder's native resolution, with a *thumbnail* image in front, similar to prior work [75]. ‡Trained with RefCOCO/+/g beforehand.

# E  Additional PE$_{spatial}$ Results

## E.1  Alignment Method Derivation

Here we detail the analysis performed to arrive at the spatial alignment method discussed in §5.

### E.1.1  Core Feature Analysis

We begin by analyzing the spatial properties of the features for PE$_{core}$G in the range of layers where it performed optimally for zero-shot tracking in §3. In Fig. 18, we plot (1) the pairwise feature cosine similarity between the pink token and all others, (2) the head average attention map for that token, and (3) the full attention matrix ($HW \times HW$).

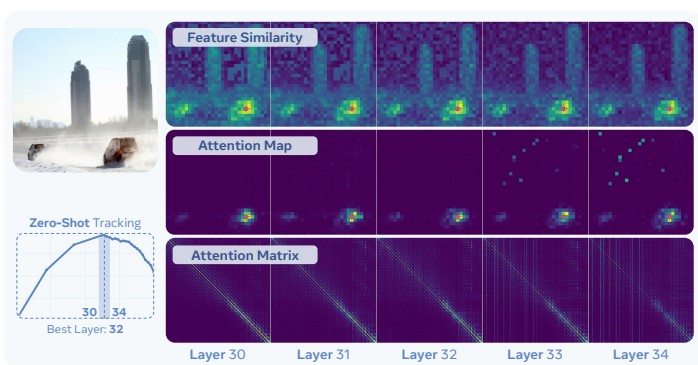

Figure 18: **PE$_{core}$G Feature Analysis.** To understand the dichotomy between optimal PE$_{core}$ features for spatial tasks observed in Fig. 4, we analyze the spatial properties of the features between layers 30 and 34.

**An 18 Layer Decoder.** Remarkably, the cause for the tracking performance peak at layer 32 is abundantly clear from observing the visualizations. Up until layer 32, the attention maps remain local. However, that changes abruptly at layer 33, at which point several tokens in the background of the image become "global" tokens. As shown by the vertical lines in the full attention matrix, starting from layer 33 every token attends to them. Thus, every layer 33 and up become part of a *decoder* for global information.

This is not a new phenomenon. Recent work [23] shows this happening in all modern vision transformers above L scale. But notably these "global tokens" are not necessarily harmful. Given the optimal layer for most tasks in Fig. 4 lies within the global token region, the information they aggregate is useful downstream. However, tracking in §3 is zero-shot and relies purely on spatial correspondences, meaning it cannot make use of the global tokens. This explains why tracking peaks right before their introduction, while tasks that rely on semantic understanding or have larger decoders that can benefit from them do well with the later layers.

### E.1.2  Spatial Alignment Method

Given the analysis in §E.1.1, we have two objectives in creating a spatial alignment method: (1) we must preserve the optimal *semantic information* of the model (including the global tokens) that peaks around layer 40, and (2) we must do so while emphasizing *local alignment* in service of spatial tasks with shallow decoders. The first can be easily achieved by aligning with the model's own features (e.g., with MaskFeat [144]), but the second is more challenging. To accomplish this, we employ the Segment Anything Model (SAM) 2.1 [108] in a novel way to enforce spatial correspondence information in PE.

**Retaining Semantics.** To retain the strong semantic features from PE$_{core}$, we finetune the model with itself as a teacher. Specifically, we train the model to maximize the cosine similarity between its *last layer* and the frozen layer 41 features of its initialization (a layer around the peak for many tasks in Fig. 4). On its own this would be a tautology, so we apply heavy regularization to the student: DropPath [49] and LayerScale [132] similar to language alignment, as well as performing MaskFeat [144] with 75% masking. We keep the teacher fixed in contrast to other state-of-the-art spatial models, which all employ an EMA teacher [96, 135]. This could potentially help, but we opt for simplicity.

**Encouraging Locality.** While we could "retain" locality by self-distilling from layer 32 features, that may be less effective as we are already distilling another layer of the model. Instead, we turn to a model that is explicitly tuned for locality: SAM [57, 108]. Notably, several works [107, 113, 116]

have shown SAM to *not* be an effective teacher when distilling from multiple sources (though recently [44] has shown it can help with some tricks). However, upon observation of the raw features of SAM 2.1-L (Fig. 19), the main problem may be the same one we are currently trying to solve: *SAM has global tokens as well*! In this case, they appear as dark spots in a grid-like arrangement across all examples in Fig. 19 raw features.

Using the features of a model that itself has global tokens to mitigate the effect of global tokens is dubious at best. But, we do not have to use SAM's *features* to learn locality. At its core, SAM is a model that transforms points into spatially contiguous masks of select object. If what we want is smooth, locally consistent features, we can use the *mask predictions* themselves. Specifically, we query SAM 2.1-L with 1024 points arranged in a $32 \times 32$ grid. For each point, SAM returns a $H \times W$ mask logit the size of the image, which it normally would threshold and NMS. However, we instead concatenate those logits into a $H \times W \times 1024$ tensor and use *that* as the feature

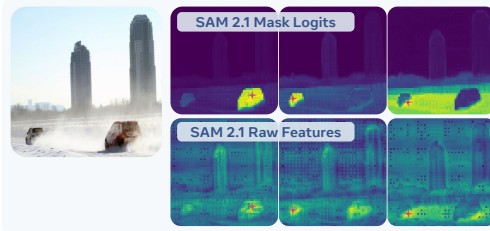

Figure 19: **SAM 2.1 Feature Similarity.** The cosine similarity between the pink marked token and all others for SAM 2.1-L [108] features *vs.* our proposed mask logit features.

map for alignment. This explicitly produces locally well-aligned features compared to the underlying feature space and has no spatial artifacts caused by global tokens, as shown in Fig. 19.

Then to align, we distill the spatial correspondences between tokens by computing their pairwise cosine similarity for both the student and the teacher (creating a $HW \times HW$ matrix for each) and aligning them with MSE loss. Unlike SAM's underlying feature space (which [44] shows may be brittle to interpolation), the mask logit features are robust to interpolation, so we simply interpolate them down and train at the $PE_{core}$ model's original 448px resolution. Finally, like for self-distillation we add the same masking and regularization. For both teachers, we apply loss to all tokens and add no extra parameters other than LayerScale.

## E.2 Layer Analysis Details

In Tab. 36, we present the raw numbers for the layer analysis plots in Fig. 9. Note that layer analysis was not performed exhaustively on every layer.

| | Detection | | | | Depth | | | | Tracking | | | | Segmentation | | | |
|---|---|---|---|---|---|---|---|---|---|---|---|---|---|---|---|---|
| Layer | $PE_{core}$ | PE Teacher | SAM Teacher | $PE_{spatial}$ | $PE_{core}$ | PE Teacher | SAM Teacher | $PE_{spatial}$ | $PE_{core}$ | PE Teacher | SAM Teacher | $PE_{spatial}$ | $PE_{core}$ | PE Teacher | SAM Teacher | $PE_{spatial}$ |
| 50 | 35.0 | 44.4 | 33.4 | 44.5 | 0.31 | 0.26 | 0.47 | 0.28 | 42.8 | 57.4 | 70.3 | 61.5 | 38.6 | 46.1 | 21.4 | 48.9 |
| 49 | 37.3 | 44.8 | 34.3 | 44.8 | 0.29 | 0.26 | 0.49 | 0.28 | 44.8 | 57.4 | 70.2 | 61.4 | 39.8 | 46.2 | 23.6 | 49.3 |
| 48 | 38.3 | 44.8 | 35.7 | 45.2 | 0.28 | 0.27 | 0.44 | 0.28 | 45.5 | 57.5 | 70.7 | 61.5 | 40.4 | 46.3 | 25.3 | 49.0 |
| 47 | 39.3 | 44.7 | 37.0 | 45.2 | 0.28 | 0.26 | 0.43 | 0.27 | 46.8 | 57.7 | 71.2 | 61.3 | 40.9 | 46.6 | 28.6 | 49.0 |
| 46 | 39.8 | 45.0 | 38.4 | 45.6 | 0.27 | 0.26 | 0.42 | 0.26 | 49.1 | 57.8 | 71.3 | 61.1 | 41.4 | 46.3 | 31.9 | 49.1 |
| 45 | 40.8 | 45.0 | 39.4 | 45.5 | 0.26 | 0.27 | 0.40 | 0.27 | 50.7 | 57.9 | 71.5 | 61.1 | 41.5 | 45.8 | 34.3 | 48.9 |
| 44 | 41.4 | 45.3 | 40.5 | 45.9 | 0.26 | 0.26 | 0.38 | 0.27 | 51.7 | 58.1 | 71.1 | 60.7 | 41.5 | 45.6 | 36.8 | 48.7 |
| 43 | 41.8 | 45.4 | 41.2 | 45.9 | 0.26 | 0.26 | 0.34 | 0.26 | 52.4 | 58.1 | 70.4 | 60.5 | 41.3 | 45.5 | 38.7 | 48.0 |
| 42 | 42.1 | 45.4 | 41.9 | 46.1 | 0.26 | 0.27 | 0.35 | 0.26 | 53.1 | 58.2 | 69.8 | 59.8 | 41.4 | 45.1 | 40.4 | 47.6 |
| 41 | 42.6 | 45.4 | 42.6 | 46.0 | 0.26 | 0.29 | 0.36 | 0.27 | 54.2 | 58.2 | 69.2 | 59.6 | 41.1 | 44.6 | 41.3 | 46.8 |
| 40 | 42.8 | 45.4 | 43.1 | 46.1 | 0.26 | 0.27 | 0.35 | 0.29 | 54.5 | 57.8 | 68.5 | 59.5 | 41.1 | 44.4 | 42.0 | 46.6 |
| 39 | 42.6 | 45.1 | 43.4 | 45.9 | 0.25 | 0.25 | 0.35 | 0.26 | 54.9 | 57.4 | 68.0 | 59.3 | 41.1 | 43.7 | 42.5 | 46.1 |
| 38 | 43.2 | 44.9 | 43.8 | 45.9 | 0.25 | 0.28 | 0.35 | 0.26 | 54.9 | 56.8 | 67.6 | 58.9 | 40.2 | 42.8 | 43.3 | 45.5 |
| 37 | 43.0 | 44.4 | 43.7 | 45.3 | 0.28 | 0.29 | 0.34 | 0.28 | 55.3 | 56.4 | 67.2 | 58.7 | 40.1 | 41.4 | 42.8 | 44.9 |
| 36 | 42.9 | 44.0 | 43.7 | 44.5 | 0.29 | 0.30 | 0.35 | 0.30 | 55.8 | 56.0 | 66.7 | 58.4 | 39.3 | 40.1 | 42.6 | 43.3 |
| 35 | 42.4 | 43.1 | 43.3 | 44.3 | 0.29 | 0.31 | 0.33 | 0.29 | 55.6 | 55.8 | 66.3 | 58.2 | 38.4 | 38.9 | 42.1 | 42.1 |
| 34 | 42.0 | 42.5 | 42.7 | 43.1 | 0.29 | 0.31 | 0.36 | 0.29 | 55.7 | 55.6 | 65.8 | 57.9 | 38.3 | 37.5 | 41.3 | 41.2 |
| 33 | 41.1 | 41.2 | 42.0 | 42.3 | 0.30 | 0.32 | 0.35 | 0.32 | 56.0 | 55.6 | 65.4 | 58.0 | 36.8 | 36.3 | 40.4 | 40.1 |
| 32 | 40.5 | 40.4 | 41.1 | 41.4 | 0.31 | 0.34 | 0.35 | 0.33 | 56.8 | 55.5 | 64.9 | 57.7 | 36.4 | 34.6 | 39.6 | 38.6 |
| 31 | 38.8 | 39.1 | 40.0 | 39.9 | 0.34 | 0.39 | 0.37 | 0.37 | 56.4 | 55.2 | 64.3 | 57.4 | 34.7 | 33.2 | 37.7 | 36.9 |
| 21 | 24.7 | 26.9 | 27.3 | 27.3 | 0.51 | 0.52 | 0.49 | 0.52 | 52.1 | 52.8 | 55.2 | 53.8 | 16.0 | 16.8 | 19.4 | 19.2 |
| 11 | 19.6 | 20.5 | 20.7 | 20.5 | 0.56 | 0.60 | 0.56 | 0.60 | 43.7 | 40.6 | 41.7 | 41.6 | 8.6 | 7.9 | 8.9 | 8.7 |
| 1 | 12.7 | 12.1 | 12.2 | 11.9 | 0.66 | 0.70 | 0.69 | 0.70 | 28.2 | 16.4 | 17.5 | 16.2 | 3.1 | 3.0 | 3.1 | 3.1 |

Table 36: **Raw Spatial Layer Analysis Results.** The raw values for the plots in Fig. 9.

### E.3 Smaller Models

We additionally distill smaller models from the original $\text{PE}_{\text{spatial}}\text{G}$ checkpoint by applying the same strategy as in spatial alignment (Sec. 5): train jointly with a semantic teacher loss and a spatial teacher loss. To align $\text{PE}_{\text{spatial}}\text{G}$, the semantic teach was an intermediate layer of the original $\text{PE}_{\text{core}}\text{G}$ model and the spatial teacher was SAM 2.1 [108].

For smaller spatial models, we repeat this alignment by finetuning the corresponding $\text{PE}_{\text{core}}$ checkpoint with the two loss functions on a portion of the pretraining data. However, unlike $\text{PE}_{\text{spatial}}\text{G}$, we do not align the smaller models to themselves and SAM 2.1. Instead, we set *both* teachers to $\text{PE}_{\text{spatial}}\text{G}$. The semantic pairwise similarity teacher loss is applied directly on the last layer of each model, and the direct distillation semantic teacher loss is applied after a linear layer (to match the feature dimension).

Results for the distilled models are given below in Tab. 37. Each model is trained with 1024 tokens as input (448px for patch 14, 512px for patch 16).

| Encoder | Res/Patch | Tracking DAVIS (↑) [101] | | | Segmentation ADE20k (↑) [164] | | |
| | | Best | Last | Idx | Best | Last | Idx |
|---|---|---|---|---|---|---|---|
| $\text{PE}_{\text{spatial}}\text{G}$ | 448/14 | 61.5 | 61.5 | 50/50 | 49.3 | 48.9 | 49/50 |
| $\text{PE}_{\text{spatial}}\text{L}$ | 448/14 | 60.6 | 60.1 | 23/24 | 48.1 | 48.1 | 24/24 |
| $\text{PE}_{\text{spatial}}\text{B}$ | 512/16 | 58.9 | 58.4 | 11/12 | 44.4 | 44.4 | 12/12 |
| $\text{PE}_{\text{spatial}}\text{S}$ | 512/16 | 57.5 | 57.5 | 12/12 | 37.5 | 37.5 | 12/12 |
| $\text{PE}_{\text{spatial}}\text{T}$ | 512/16 | 55.0 | 54.6 | 11/12 | 27.6 | 27.6 | 12/12 |

Table 37: **Distilled Spatial Models**. Smaller spatial models compared to the original $\text{PE}_{\text{spatial}}\text{G}$ teacher checkpoint. Evaluation with the same settings as Tab. 6.

## E.4 Additional Qualitative Results

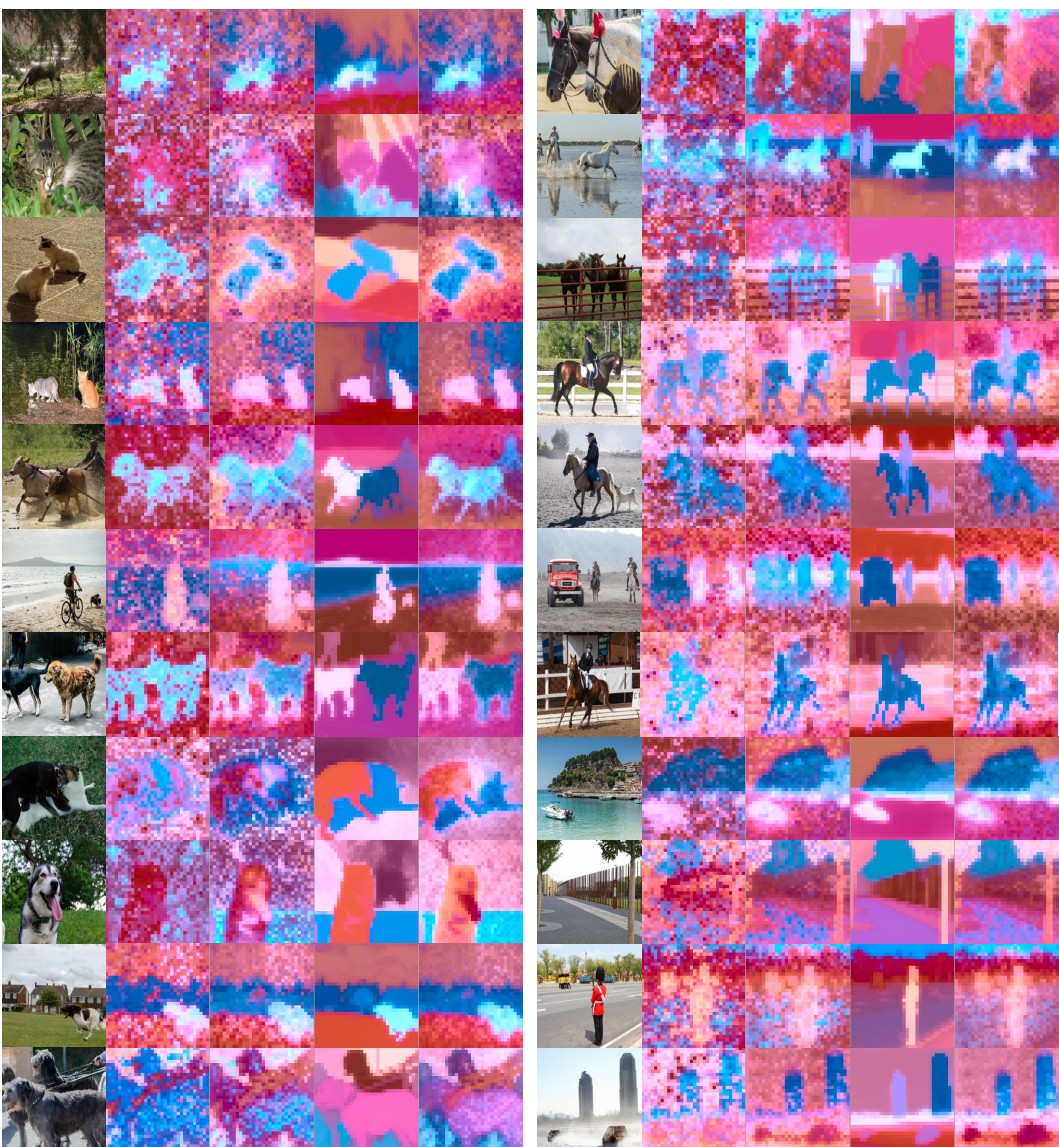

Figure 20: **More Visualizations** of the feature space following Fig. 10. After the image itself, column 1 is $PE_{core}G$ last layer features, column 2 is $PE_{core}G$ aligned to its own layer 41, column 3 is $PE_{core}G$ aligned to SAM 2.1-L [108] mask logits, and column 4 is $PE_{core}G$ aligned to both, denoted $PE_{spatial}G$. See §B.3.2 for visualization method. Example images are from SA-1B [57].

# F  Broader Impacts

This work covers several artifacts and techniques that may or may not have broader societal impact. In this section, we enumerate those components and discuss what positive and negative implications they may have for each.

**PE$_{core}$.** The core PE model is a CLIP model that matches the given image or video and a string of text. This can be used in data curation, image or video search and retrieval, as well as several downstream use cases where vision and language alignment are necessary. PE$_{core}$ improves upon prior work in this area substantially in both robustness (see Tab. 3) and fairness (see Tab. 25), as well as extending to video (Tab. 4). This has the potential to improve standalone use cases such as search, but also it has the potential to improve downstream machine learning system, for instance by providing better data curation. However, PE$_{core}$ is not perfect and still makes errors (see benchmarks above). This has the potential for negative impact if PE$_{core}$ is used without regard to the possibility of mistakes.

**PE$_{lang}$.** Language alignment extend PE to downstream applications using Multimodal Large Language Models (MLLMs). While, we do not release any complete MLLM artifacts ourselves in this work, the PE$_{lang}$ we intend to release is an important component of such a system. While the usage of PE$_{lang}$ in these systems has the potential to increase performance (especially on tasks requiring OCR, see Tab. 5), all MLLM systems have potential to hallucinate and generate errors. A system developed with PE$_{lang}$ would be no exception to this issue.

**PE$_{spatial}$.** Similarly, spatial alignment extends PE to downstream applications such as tracking, segmentation, and detection. PE$_{spatial}$ improves performance in these areas *vs.* prior models, which has the potential to improve security systems, image editing systems, and other traditional computer vision systems. However, that naturally also comes with the risk that these systems can be used for unintended purposes. To mitigate this, we release only the PE$_{spatial}$ feature encoder and not any of the downstream application heads.

**PE Video Dataset (PVD).** Along with the models, we also release a novel annotated video-caption dataset consisting of high quality samples selected for high motion content. The captions are generated by an MLLM and then refined by human annotators. Because of the quality of this dataset, this has the potential to improve downstream applications such as video generation, video-language alignment (which is what this work uses it for), and video benchmarking (with our PVD Benchmark). However, it is important to note these captions were initially generated by a model. Even with human refinement, there may be mistakes or hallucinations left in the captions, which could impact downstream use. Similarly, downstream uses of the dataset such as training video generators may have harmful implications. To address this, we will control the access to the dataset as well as include appropriate license terms.

