# OpenReview forum: "Perception Encoder: The best visual embeddings are not at the output of the network"
_NeurIPS.cc/2025/Conference — NeurIPS 2025 oral_

### Official Review · Reviewer_AWet · 2025-06-20

**Clarity:** 4
**Significance:** 3
**Originality:** 2
**Rating:** 5
**Confidence:** 5

**Summary:**

This paper presents Perception Encoder (PE), a family of vision encoders that achieve state-of-the-art performance across a wide range of image, video, and multimodal tasks using a unified contrastive pretraining strategy. Unlike previous work that often combines multiple pretraining objectives (e.g., contrastive, captioning, self-supervised spatial losses), the authors show that large-scale contrastive vision-language pretraining alone, when executed with a well-engineered data pipeline and training recipe, is sufficient to yield strong general-purpose features. These features emerge in intermediate layers of the model, rather than at the output. To make these features usable, the authors introduce two alignment strategies: language alignment (PElang) and spatial alignment (PEspatial), which bring the strong intermediate representations to the end of the network for effective use in downstream MLLM and spatial tasks respectively. The paper also contributes a novel video-text dataset and promises open-source release.

**Questions:**

1. Please see weakness section.

2. In the PElang alignment method, the last 3 layers of PEcore are discarded before fine-tuning. If the alignment stage can bring useful features for mllm to the last layer, why still dropping layers?

2. The language alignment is performned on llama3.2 3B, and the final performance is reported by training PElang with  Llama 3.1-
instruct 8B -- the same family of models. Will the language alignment still work for different class of models? For example, aligning PEcore with qwen-1B and then perform final training with llama-8B.

**Ethical Concerns:**

["NO or VERY MINOR ethics concerns only"]

**Final Justification:**

The authors response has clearly addressed my concerns. I will keep the rating for this paper as accept and increase my confidence rating.

**Limitations:**

Yes.

**Quality:**

3

**Strengths And Weaknesses:**

**Strengths**

1. The empirical study is thorough, with extensive experiments across classification, retrieval, dense prediction, and MLLM benchmarks. The results convincingly establish new SOTA in multiple categories. Ablation studies clearly isolate the impact of each component (contrastive recipe modifications, video-data engine, alignment stages).

2. It demonstrates that a single, simple contrastive pretraining can yield intermediate features competitive with specialized captioning and self-supervised models.

3. It shows  that intermediate contrastive features can be “lifted” to the network output via short finetuning, unlocking broad downstream utility. The paper also introduces a two-teacher spatial alignment combining self-distillation and SAM-derived locality.

**Weakness**

1. The comparison in Table 5 evaluates PElang against other frozen vision encoders, but does not unfreeze them during MLLM fine-tuning. This creates a potential bias in favor of PElang. The authors should ideally also report performance for these baselines when the image encoder is unfrozen during fine-tuning.

2. The paper does not include a result for directly plugging PEcore (without PElang alignment) into an MLLM and unfreezing it. It remains unclear whether an explicit language alignment stage is strictly necessary or whether one-stage fine-tuning with unfrozen image encoder would suffice.

---

> ### Author Rebuttal · Authors · 2025-07-30
>
> > [Q3] The language alignment is performned on llama3.2 3B, and the final performance is reported by training PElang with Llama 3.1- instruct 8B -- the same family of models. Will the language alignment still work for different class of models? For example, aligning PEcore with qwen-1B and then perform final training with llama-8B.
>
> This is a good question! It’s very important for the community that PE lang’s performance is transferable to other LLMs, so we tested PE lang (aligned to Llama 3.2 3B) using _QwenLM 2.5 7B_ as the LLM in Appendix C.7 / Table 29 with the encoder frozen. Despite being aligned to a different family of models, PE lang G still significantly outperforms other models on almost every benchmark when benchmarked using QwenLM.
>
>
>
>
>
> ---
> > [W1] The comparison in Table 5 evaluates PElang against other frozen vision encoders, but does not unfreeze them during MLLM fine-tuning. This creates a potential bias in favor of PElang. The authors should ideally also report performance for these baselines when the image encoder is unfrozen during fine-tuning.
>
> This is a great point, thank you for the suggestion. It could be unfair to other models that weren’t language tuned to compare to PE lang, which is a language tuned model. Several of the models we compared to are already language-aligned with an LLM (AIMv2, SigLIP2 pretrained with captioning, InternViT 2.5 extracted from an MLLM), but your proposed experiment makes this comparison a lot more concrete.
>
> We ran this experiment (same as Table 5, but with the encoder unfrozen) on AIMv2 3B, SigLIP2 g-opt, and PE lang G, all with 1024 tokens per image:
>
> |  | Chart QA | Doc VQA | Info. QA | AI 2D | Text VQA | OK VQA | POPE | VQA v2 | Flickr | COCO | No Cap | Ref COCO | Ref COCO+ | Ref COCOg | Video MME | STAR | TGIF QA | Ego Schema | MV Bench | Perception Test |
> |---|:---:|:---:|:---:|:---:|:---:|:---:|:---:|:---:|:---:|:---:|:---:|:---:|:---:|:---:|:---:|:---:|:---:|:---:|:---:|:---:|
> | AIMv2 3B | 77.4 | 68.8 | 39.4 | 74.4 | 73.1 | 64.4 | 88.4 | 81.7 | 97.1 | 139.7 | 120.7 | 69.3 | 62.3 | 77.5 | 46.8 | 53.2 | 64.5 | 58.4 | 48.9 | 53.9 |
> | SigLIP2 gopt | 77.3 | 68.6 | 39.6 | 74.6 | **74.9** | **67.7** | 89.5 | **83.2** | **97.9** | 140.7 | 121.5 | **74.8** | 65.0 | 80.3 | 43.8 | 52.7 | 66.8 | 59.0 | **51.2** | 52.9 |
> | PE lang G | **81.6** | **84.7** | **48.1** | **76.8** | 74.1 | 66.9 | **89.6** | 82.9 | 97.4 | **141.7** | **123.4** | 73.4 | **65.9** | **80.8** | **48.0** | **58.9** | **70.4** | **63.0** | 50.9 | **56.2** |
>
>
> It seems all models, including PE lang, benefit from unlocking the encoder. However, PE lang G still outperforms the other models overall, often by a significant margin. Again, thank you for the suggestion—we will include this table in the Appendix of the final draft!
>
>
>
>
>
>
> ---
> > [W2] The paper does not include a result for directly plugging PEcore (without PElang alignment) into an MLLM and unfreezing it. It remains unclear whether an explicit language alignment stage is strictly necessary or whether one-stage fine-tuning with unfrozen image encoder would suffice.
>
> Another good point, thank you for mentioning this. We repeated the experiment above (Table 5 with encoder unfrozen, 1024 tokens) with PE core v.s. PE lang for both L and G, reported below:
>
>
> |  | Chart QA | Doc VQA | Info. QA | AI 2D | Text VQA | OK VQA | POPE | VQA v2 | Flickr | COCO | No Cap | Ref COCO | Ref COCO+ | Ref COCOg | Video MME | STAR | TGIF QA | Ego Schema | MV Bench | Perception Test |
> |---|:---:|:---:|:---:|:---:|:---:|:---:|:---:|:---:|:---:|:---:|:---:|:---:|:---:|:---:|:---:|:---:|:---:|:---:|:---:|:---:|
> | PE core L | 76.9 | 70.6 | 39.1 | 70.8 | 69.7 | **64.5** | 88.5 | 80.3 | **93.7** | 136.8 | 117.1 | 71.4 | 63.5 | 77.7 | 45.3 | 52.6 | 65.4 | 53.4 | **50.3** | 52.8 |
> | PE lang L | **81.2** | **82.5** | **45.6** | **75.3** | **71.7** | 64.4 | **89.2** | **80.9** | 85.1 | **138.7** | **119.7** | **72.8** | **65.7** | **80.5** | **47.1** | **55.1** | **68.4** | **58.4** | 49.9 | **54.3** |
> |   |  |  |  |  |  |  |  |  |  |  |  |  |  |  |  |  |  |  |  |  |
> | PE core G | 73.6 | 67.2 | 38.5 | 71.7 | 68.9 | 65.6 | 88.4 | 80.9 | 94.0 | 137.9 | 119.7 | 70.7 | 61.4 | 75.5 | 44.2 | 56.0 | 66.3 | 60.6 | 50.5 | 52.5 |
> | PE lang G | **81.6** | **84.7** | **48.1** | **76.8** | **74.1** | **66.9** | **89.6** | **82.9** | **97.4** | **141.7** | **123.4** | **73.4** | **65.9** | **80.8** | **48.0** | **58.9** | **70.4** | **63.0** | **50.9** | **56.2** |
>
>
> Both PE lang models significantly outperform the PE core ones in this unfrozen setup, especially for the larger G size. Thus, it seems that a language alignment step is still necessary even when the encoder is unfrozen during MLLM construction. This is important to show, so thank you again for your suggestion! We’ll add this table to the Appendix as well.
>
>
>
>
>
>
> ---
> > [Q2] In the PElang alignment method, the last 3 layers of PEcore are discarded before fine-tuning. If the alignment stage can bring useful features for mllm to the last layer, why still dropping layers?
>
> Good catch! This was an artifact of our early experimentation and the order we performed our ablations. In Appendix C.6 / Table 27, we detail how we designed the language alignment recipe. And there, we found aligning to layer 47 was better overall than layer 50. However, later in development we found regularization (drop path and layerscale, last two rows of Table 27) to be particularly useful for robust alignment and transferability. If we would repeat all the ablation experiments with regularization included first, it’s likely that layer 50 alignment would outperform or at least be on par to layer 47.
>
>
>
>
> ---
> We would like to thank the reviewer for their thoughtful comments and great suggestions. We particularly appreciate your comments that our _“empirical study is thorough”_, _“the results convincingly establish new SOTA in multiple categories”_, and that our _“ablation studies clearly isolate the impact of each component”_. We are also grateful for your suggested experiments, which we believe have greatly strengthened the language alignment section of the paper. Please feel free to let us know if anything needs further clarification and we will try to answer.

---

> ### Comment · Reviewer_AWet · 2025-08-05
>
> The authors response has clearly addressed my concerns. I will keep the rating for this paper as accept and increase my confidence rating. Congrats on such great paper with solid experiments.

---

### Official Review · Reviewer_jvg2 · 2025-06-26

**Clarity:** 4
**Significance:** 4
**Originality:** 4
**Rating:** 6
**Confidence:** 4

**Summary:**

This paper first introduces a new data processing pipeline for obtaining high-quality image/video captions. Then, based on these data, the PE encoder is proposed. It adopts the simple contrastive pretraining objective, but achieves SOTA performance. Developing on this encoder, the authors argue that for different downstream tasks, the vision features should be extracted from different layers of the ViT, which is not surprising. Massive experiments are conducted to validate the proposed method. The authors also plan to release the model and dataset.

**Questions:**

For pretraining the PE encoder, the authors now just implement a simple contrastive objective. I am curious if we can implement it with more advanced ones like SigLIP2?

**Ethical Concerns:**

["NO or VERY MINOR ethics concerns only"]

**Final Justification:**

Thanks to the authors for the detailed reply. I would love to raise my score. I believe the conclusion in this paper is solid, and the findings are interesting, which will benefit the whole research community a lot.

**Limitations:**

yes

**Quality:**

4

**Strengths And Weaknesses:**

Strengths:
1. The paper is well-formatted and easy to follow.
2. Massive experiments are conducted, which makes the conclusion convincing.
3. The open-source of this work would bring significant benefits to the research community.
4. The method is simple but effective, and personally, I enjoy this kind of straightforward work.
5. The data processing pipeline is useful.

Minor Weaknesses: I don't see any major weaknesses, but would like to raise a minor concern.

I feel like the finding that the best visual embeddings are from different intermediate layers is not surprising, and it is almost well-known in all kinds of previous works. For VLMs, InternVL uses -2/-4th visual feature from the ViT(the authors also mentioned this in Line 268). For common tasks like detection or segmentation, it is almost a standard protocol to use visual features from multiple layers of ViT. In this work, the authors seem to just unify them in one paper and just implement a widely-used contrastive loss to pretrain the ViT, which slightly diminishes the novelty of this paper.

---

> ### Author Rebuttal · Authors · 2025-07-30
>
> > [W1]  I don't see any major weaknesses, but would like to raise a minor concern.
> >
> > I feel like the finding that the best visual embeddings are from different intermediate layers is not surprising, and it is almost well-known in all kinds of previous works. For VLMs, InternVL uses -2/-4th visual feature from the ViT(the authors also mentioned this in Line 268). For common tasks like detection or segmentation, it is almost a standard protocol to use visual features from multiple layers of ViT. In this work, the authors seem to just unify them in one paper and just implement a widely-used contrastive loss to pretrain the ViT, which slightly diminishes the novelty of this paper.
>
> Thank you, this is a great point and important to clarify. There are two aspects of the main “best layer” claim:
>
> 1. Models can have features that are relevant to a given task but not output them.
> 2. _Our model_ (contrastively pretrained) shows very strong general features, on par with the best models in each domain.
>
> The first part (1), as you point out, is not new. In fact, we dedicate a section of the related work (L372-383) showing other works that share the same finding. What we add to the conversation is controlled, large-scale layer-wise ablations of large models across a diverse set of tasks. In this respect, we show surprising results that all models we test (DINOv2-g, AIMv2 3B, PE core G) have optimal layers for some tasks as deep as the very middle of the network (tracking for AIMv2, OCR for DINOv2).
>
> However, the main crux of our work that we try to describe in Section 3 is (2): that, when trained in the right way, a fairly vanilla CLIP model has intermediate features on par with the best pretrained models whose losses specialize for that task (AIMv2 for MLLM, DINOv2 for spatial). We explicitly ablate this phenomenon in Appendix C.5, where we show these general features were a direct result of our robust pretraining recipe in Section 2.1 and that this general performance _scales_. To our knowledge, this is the first time this has ever been shown in a CLIP model.
>
> Thank you again for catching this! We will update the introduction in the final draft to make this point clear.
>
>
>
>
> ---
> > [Q1] For pretraining the PE encoder, the authors now just implement a simple contrastive objective. I am curious if we can implement it with more advanced ones like SigLIP2?
>
> Thank you for the suggestion. In this work, we wanted to keep the pretraining as simple as possible. We did consider using multiple losses to achieve more broad alignment, but given our layer-wise analysis we found that producing one model with aligned features for classification, retrieval, MLLM, and dense tasks is challenging. Thus, we opted to create three separate models to maximize utility to the community and to study the effects of alignment for these domains individually.
>
> Interestingly, concurrently with this work, SigLIP2 chose the opposite direction by attempting to support all MLLM and dense prediction tasks with a single model using multiple losses during pretraining. However, as we show in Table 6, this only seemed to improve alignment by a few layers v.s. SigLIP1 if at all (at least for spatial tasks). Thus, it seems like SigLIP2’s loss mix is not the right answer for joint pretraining. We hope that our analysis and layer-wise evaluation techniques can help accelerate future work in this area.
>
>
>
>
> ---
> Finally, we would like to thank their reviewer for their comprehensive review. We especially appreciate your comments that, _“the paper is well-formatted and easy to follow”_, _“massive experiments are conducted, which makes the conclusion convincing”_, and _“the open-source of this work would bring significant benefits to the research community”_. If you still have any thoughts or would like further elaboration, feel free to let us know.

---

### Official Review · Reviewer_KER3 · 2025-07-02

**Clarity:** 4
**Significance:** 4
**Originality:** 4
**Rating:** 6
**Confidence:** 4

**Summary:**

The paper „Perception Encoder: The best visual embeddings are not at the output of the network“ proposes a multi-step training strategy to create a new vision foundation model, effective for various tasks. First, the authors train a base image model with a vision-language CLIP-like contrastive objective, using carefully ablated components. Next, they employ a video data engine to source captioned videos, with which their train their all-purpose Core model. Furthermore, they discover that different layers within the network are most effective for different tasks, and with that in mind, perform language and spatial alignment of their core model, to train specialized spatial and language models.

**Questions:**

Overall, the paper is already top notch, without minor room for improvement. However, I’m still asking myself:
- Why do you think the video data engine moves the performance mainly for COCO-like dataset and not ImageNet object-centric datasets? Has the performance of the model already plateaued for ImageNet at that point?
- It would be nice to have to separate alignment trainings nicely displayed in a supplementary table (see weakness)

I really like the paper, and rate it with „Strong Accept“. I would appreciate if the authors would address my points in the rebuttal.

**Ethical Concerns:**

["NO or VERY MINOR ethics concerns only"]

**Final Justification:**

The authors have addressed my remaining weaknesses. I therefore keep my rating of Strong Accept.

**Limitations:**

Yes

**Quality:**

4

**Strengths And Weaknesses:**

Strengths:
- The proposed method for the base image model is very systematically designed and each component’s effect is well investigated
- Training a unified video and image VFM is a step in the right direction, since both forms of data can benefit from each other
- Employing a data engine to move to video data provides a very targeted way to source data that is effective at improving the model’s performance
- The design components of the video data engine are well ablated and thoroughly investigated
- The results presented in Tables 3 (image) and 4 show consistent across many different datasets and tasks.
- The most interesting finding is presented in Sec. 3, where the authors discover that different layers of the network are best suited for the various different tasks, such as language and spatial tasks. This serves as a nice motivation for the following alignment trainings.
- The spatial alignment to layer 41 AND SAM logits is an interesting design choice, and the visualizations show that it preserves a good mix of details and semantics in the last layer.
- Overall, the results of the alignment models show a strong performance and underline the effectiveness of the respective alignment trainings.

Weaknesses:
- The video data engine ablation shows that it moves the needle only a little bit for the object-centric image datasets like ImageNet, but the gains are much greater for the more complex image datasets like COCO and, obviously, all video datasets. It would be interesting to learn about why especially COCO got such a boost, but not ImageNet. Perhaps the video scenes are more similar to COCO scenes?
- While I appreciate Fig. 6 with the graphs showing the effects of aligning just to layer 41 or just to SAM logits, I think it would nice to have these numbers in a Table somewhere in the supplementary.
- In the Table 3 evaluation on image tasks, without specialized alignment of the model, the performance many times just marginally exceeds that of SigLIP2, if at all. Overall in this table, these two models are relatively close together.

---

> ### Author Rebuttal · Authors · 2025-07-30
>
> > [W1] The video data engine ablation shows that it moves the needle only a little bit for the object-centric image datasets like ImageNet, but the gains are much greater for the more complex image datasets like COCO and, obviously, all video datasets. It would be interesting to learn about why especially COCO got such a boost, but not ImageNet. Perhaps the video scenes are more similar to COCO scenes?
> >
> > [Q2] Why do you think the video data engine moves the performance mainly for COCO-like dataset and not ImageNet object-centric datasets? Has the performance of the model already plateaued for ImageNet at that point?
>
> Thank you for the question; this is important to explain. We believe the video data helps image performance in two ways:
> 1. The synthetic captions we generate (see Fig. 9 for examples) are often higher quality than the alt-text we use during pretraining. Several works [1,2,3] have shown this to be effective in improving _retrieval_ performance, which includes datasets such as COCO and Flickr (see Table 3).
> 2. Synthetic captions typically _do not_ improve classification performance (ImageNet, ObjectNet, etc.), as the structured descriptions of synthetic captions do not capture the long-tail information necessary for fine-grained classification. However, the fact this new data consists of _videos_ means that the model now sees a diverse set of objects from many different angles and in many novel situations. This is particularly noticeable in the improvement of classification robustness metrics, particularly ObjectNet and ImageNet-A (see Table 3).
>
> Taken together, the video data improves the robustness of classification and the effectiveness of retrieval.
>
>
> [1] “What If We Recaption Billions of Web Images with LLaMA-3?”
> [2] “Revisit Large-Scale Image-Caption Data in Pretraining Multimodal Foundation Models”
> [3] “Altogether: Image Captioning via Re-aligning Alt-text”
>
>
>
>
>
> ---
> > [W3] In the Table 3 evaluation on image tasks, without specialized alignment of the model, the performance many times just marginally exceeds that of SigLIP2, if at all. Overall in this table, these two models are relatively close together.
>
> This is true! Though, we’d like to note that (1) SigLIP2 is concurrent work—we include it in the results because we are confident in our model and techniques; (2) PE core jointly performs image and video CLIP tasks. SigLIP2 doesn’t support this and thus does much worse on videos in Table 4; (3) models trained on WebLI (Coca, SigLIP, SigLIP2) have dominated the CLIP benchmarks for the past 3 years. This is the first time they’ve been outperformed without WebLI.
>
>
>
>
> ---
> > [W2] While I appreciate Fig. 6 with the graphs showing the effects of aligning just to layer 41 or just to SAM logits, I think it would nice to have these numbers in a Table somewhere in the supplementary.
> >
> > [Q2] It would be nice to have to separate alignment trainings nicely displayed in a supplementary table
>
> Thank you for the suggestion! We will add the raw data for Figures 5 and 6 in the appendix. In the meantime, here are the raw numbers. Note that the layer-wise analysis wasn’t done exhaustively for every layer.
>
> Figure 5:
>
> | Layer |  | OCR Core | OCR Lang |  | Cap Core | Cap Lang |  | VQA Core | VQA Lang |  | Grnd Core | Grnd Lang |
> |:---:|:---:|:---:|:---:|:---:|:---:|:---:|:---:|:---:|:---:|:---:|:---:|:---:|
> | 50 |  | 48.6 |  |  | 114.0 |  |  | 74.1 |  |  | 39.1 |  |
> | 49 |  | 53.8 |  |  | 114.5 |  |  | 74.9 |  |  | 54.2 |  |
> | 48 |  | 56.3 |  |  | 114.8 |  |  | 74.8 |  |  | 55.6 |  |
> | 47 |  | 57.3 | 72.4 |  | 113.9 | 120.1 |  | 75.3 | 78.1 |  | 57.8 | 71.2 |
> | 46 |  | 58.3 | 72.2 |  | 114.1 | 120.0 |  | 75.1 | 78.2 |  | 58.4 | 70.8 |
> | 45 |  | 59.6 | 72.0 |  | 114.6 | 119.7 |  | 75.1 | 77.7 |  | 61.2 | 70.7 |
> | 43 |  | 59.9 | 71.5 |  | 113.8 | 117.7 |  | 74.8 | 77.4 |  | 63.1 | 70.5 |
> | 42 |  | 60.3 | 71.3 |  | 113.1 | 116.6 |  | 73.8 | 77.2 |  | 64.1 | 70.8 |
> | 41 |  | 60.8 | 70.7 |  | 112.5 | 115.4 |  | 73.3 | 76.7 |  | 66.6 | 71.2 |
> | 40 |  | 61.4 | 69.8 |  | 112.0 | 115.0 |  | 74.0 | 76.3 |  | 66.5 | 71.0 |
> | 39 |  | 61.8 | 70.0 |  | 111.0 | 113.5 |  | 74.0 | 75.6 |  | 67.1 | 71.1 |
> | 38 |  | 62.1 | 69.3 |  | 110.2 | 112.6 |  | 73.9 | 75.3 |  | 68.7 | 70.3 |
> | 36 |  | 61.3 | 67.9 |  | 108.5 | 109.4 |  | 73.0 | 74.0 |  | 67.8 | 70.4 |
> | 34 |  | 58.7 | 65.4 |  | 102.8 | 104.7 |  | 70.0 | 72.2 |  | 66.5 | 69.6 |
> | 33 |  | 57.3 | 64.1 |  | 100.1 | 102.7 |  | 69.0 | 71.3 |  | 65.9 | 68.9 |
> | 32 |  | 54.2 | 63.2 |  | 96.9 | 100.1 |  | 67.8 | 70.6 |  | 65.5 | 68.7 |
> | 31 |  | 50.9 | 60.7 |  | 93.2 | 96.7 |  | 65.5 | 68.7 |  | 63.0 | 67.3 |
> | 21 |  | 29.6 | 30.3 |  | 59.3 | 71.3 |  | 49.1 | 52.0 |  | 41.5 | 51.6 |
> | 11 |  | 28.8 | 28.9 |  | 47.4 | 59.0 |  | 47.0 | 49.2 |  | 30.9 | 43.3 |
> | 2 |  | 28.2 | 28.7 |  | 38.6 | 42.8 |  | 43.3 | 43.8 |  | 22.8 | 27.2 |
>
>
> Figure 6:
>
> | Layer |  | Det Core | Det PE | Det SAM | Det Spatial |   |   | Depth Core | Depth PE | Depth SAM | Depth Spatial |   |   | Track Core | Track PE | Track SAM | Track Spatial |   |   | Seg Core | Seg PE | Seg SAM | Seg Spatial |
> |:---:|:---:|:---:|:---:|:---:|:---:|:---:|:---:|:---:|:---:|:---:|:---:|:---:|:---:|:---:|:---:|:---:|:---:|:---:|:---:|:---:|:---:|:---:|:---:|
> | 50 |  | 35.0 | 44.4 | 33.4 | 44.5 |  |  | 0.31 | 0.26 | 0.47 | 0.28 |  |  | 42.8 | 57.4 | 70.3 | 61.5 |  |  | 38.6 | 46.1 | 21.4 | 48.9 |
> | 49 |  | 37.3 | 44.8 | 34.3 | 44.8 |  |  | 0.29 | 0.26 | 0.49 | 0.28 |  |  | 44.8 | 57.4 | 70.2 | 61.4 |  |  | 39.8 | 46.2 | 23.6 | 49.3 |
> | 48 |  | 38.3 | 44.8 | 35.7 | 45.2 |  |  | 0.28 | 0.27 | 0.44 | 0.28 |  |  | 45.5 | 57.5 | 70.7 | 61.5 |  |  | 40.4 | 46.3 | 25.3 | 49.0 |
> | 47 |  | 39.3 | 44.7 | 37.0 | 45.2 |  |  | 0.28 | 0.26 | 0.43 | 0.27 |  |  | 46.8 | 57.7 | 71.2 | 61.3 |  |  | 40.9 | 46.6 | 28.6 | 49.0 |
> | 46 |  | 39.8 | 45.0 | 38.4 | 45.6 |  |  | 0.27 | 0.26 | 0.42 | 0.26 |  |  | 49.1 | 57.8 | 71.3 | 61.1 |  |  | 41.4 | 46.3 | 31.9 | 49.1 |
> | 45 |  | 40.8 | 45.0 | 39.4 | 45.5 |  |  | 0.26 | 0.27 | 0.40 | 0.27 |  |  | 50.7 | 57.9 | 71.5 | 61.1 |  |  | 41.5 | 45.8 | 34.3 | 48.9 |
> | 44 |  | 41.4 | 45.3 | 40.5 | 45.9 |  |  | 0.26 | 0.26 | 0.38 | 0.27 |  |  | 51.7 | 58.1 | 71.1 | 60.7 |  |  | 41.5 | 45.6 | 36.8 | 48.7 |
> | 43 |  | 41.8 | 45.4 | 41.2 | 45.9 |  |  | 0.26 | 0.26 | 0.34 | 0.26 |  |  | 52.4 | 58.1 | 70.4 | 60.5 |  |  | 41.3 | 45.5 | 38.7 | 48.0 |
> | 42 |  | 42.1 | 45.4 | 41.9 | 46.1 |  |  | 0.26 | 0.27 | 0.35 | 0.26 |  |  | 53.1 | 58.2 | 69.8 | 59.8 |  |  | 41.4 | 45.1 | 40.4 | 47.6 |
> | 41 |  | 42.6 | 45.4 | 42.6 | 46.0 |  |  | 0.26 | 0.29 | 0.36 | 0.27 |  |  | 54.2 | 58.2 | 69.2 | 59.6 |  |  | 41.1 | 44.6 | 41.3 | 46.8 |
> | 40 |  | 42.8 | 45.4 | 43.1 | 46.1 |  |  | 0.26 | 0.27 | 0.35 | 0.29 |  |  | 54.5 | 57.8 | 68.5 | 59.5 |  |  | 41.1 | 44.4 | 42.0 | 46.6 |
> | 39 |  | 42.6 | 45.1 | 43.4 | 45.9 |  |  | 0.25 | 0.25 | 0.35 | 0.26 |  |  | 54.9 | 57.4 | 68.0 | 59.3 |  |  | 41.1 | 43.7 | 42.5 | 46.1 |
> | 38 |  | 43.2 | 44.9 | 43.8 | 45.9 |  |  | 0.25 | 0.28 | 0.35 | 0.26 |  |  | 54.9 | 56.8 | 67.6 | 58.9 |  |  | 40.2 | 42.8 | 43.3 | 45.5 |
> | 37 |  | 43.0 | 44.4 | 43.7 | 45.3 |  |  | 0.28 | 0.29 | 0.34 | 0.28 |  |  | 55.3 | 56.4 | 67.2 | 58.7 |  |  | 40.1 | 41.4 | 42.8 | 44.9 |
> | 36 |  | 42.9 | 44.0 | 43.7 | 44.5 |  |  | 0.29 | 0.30 | 0.35 | 0.30 |  |  | 55.8 | 56.0 | 66.7 | 58.4 |  |  | 39.3 | 40.1 | 42.6 | 43.3 |
> | 35 |  | 42.4 | 43.1 | 43.3 | 44.3 |  |  | 0.29 | 0.31 | 0.33 | 0.29 |  |  | 55.6 | 55.8 | 66.3 | 58.2 |  |  | 38.4 | 38.9 | 42.1 | 42.1 |
> | 34 |  | 42.0 | 42.5 | 42.7 | 43.1 |  |  | 0.29 | 0.31 | 0.36 | 0.29 |  |  | 55.7 | 55.6 | 65.8 | 57.9 |  |  | 38.3 | 37.5 | 41.3 | 41.2 |
> | 33 |  | 41.1 | 41.2 | 42.0 | 42.3 |  |  | 0.30 | 0.32 | 0.35 | 0.32 |  |  | 56.0 | 55.6 | 65.4 | 58.0 |  |  | 36.8 | 36.3 | 40.4 | 40.1 |
> | 32 |  | 40.5 | 40.4 | 41.1 | 41.4 |  |  | 0.31 | 0.34 | 0.35 | 0.33 |  |  | 56.8 | 55.5 | 64.9 | 57.7 |  |  | 36.4 | 34.6 | 39.6 | 38.6 |
> | 31 |  | 38.8 | 39.1 | 40.0 | 39.9 |  |  | 0.34 | 0.39 | 0.37 | 0.37 |  |  | 56.4 | 55.2 | 64.3 | 57.4 |  |  | 34.7 | 33.2 | 37.7 | 36.9 |
> | 21 |  | 24.7 | 26.9 | 27.3 | 27.3 |  |  | 0.51 | 0.52 | 0.49 | 0.52 |  |  | 52.1 | 52.8 | 55.2 | 53.8 |  |  | 16.0 | 16.8 | 19.4 | 19.2 |
> | 11 |  | 19.6 | 20.5 | 20.7 | 20.5 |  |  | 0.56 | 0.60 | 0.56 | 0.60 |  |  | 43.7 | 40.6 | 41.7 | 41.6 |  |  | 8.6 | 7.9 | 8.9 | 8.7 |
> | 1 |  | 12.7 | 12.1 | 12.2 | 11.9 |  |  | 0.66 | 0.70 | 0.69 | 0.70 |  |  | 28.2 | 16.4 | 17.5 | 16.2 |  |  | 3.1 | 3.0 | 3.1 | 3.1 |
>
>
>
>
> ---
> Finally, we’d like to extend our thanks to the reviewer for their glowing review and for describing our work as _“already top notch”_. We hope we could adequately answer your questions with this response. Feel free to let us know if anything still needs further elaboration.

---

> > ### Comment · Reviewer_KER3 · 2025-08-01
> > **Response to the rebuttal**
> >
> > Thank you for addressing my remaining comments, and congrats on a great paper!

---

### Official Review · Reviewer_KE9i · 2025-07-02

**Clarity:** 4
**Significance:** 4
**Originality:** 4
**Rating:** 5
**Confidence:** 4

**Summary:**

The paper introduces Perception Encoder, a family of vision encoders trained solely with contrastive vision-language learning, demonstrating that this single pretraining strategy can match or surpass other pretraining objectives (e.g., captioning) across a wide range of image and video tasks.

The authors’ core research finding is that the best features for tasks like VQA, detection, and depth estimation typically emerge in different intermediate layers. Therefore, they propose language alignment and spatial alignment methods, for multimodal language modeling and dense prediction downstream use cases respectively, to lift the best features from the middle layers toward the end.

Perception Encoder provides useful resources for general-purpose vision understanding, as the authors plan to release the models, code, and dataset.

**Questions:**

1. Line 277~282 mentions that the model training mix did not contain grounding data and it means that the significantly lifted grounding performance is entirely due to the strong intermediate grounding features now being aligned to the end of the network. However, the alignment method itself doesn’t intuitively or theoretically guarantee such an outcome. So what mechanism might explain this emergent grounding capability?
2. Line 257~259 concludes that “a well-tuned large-scale contrastive model can learn general embeddings in the process of fitting its objective, but it fails to output them”. However, based on the results in Figure 4, this limitation seems to also apply to models like AIMv2 and DINOv2, which are not purely contrastive. Is there experimental evidence that this phenomenon is specific to contrastive models? Or does this insight actually more broadly apply to other well-tuned pretrained models regardless of their training objective?
3. Why not design a Perception Encoder checkpoint that integrates both language alignment and spatial alignment? This could enable a single checkpoint to support both downstream multimodal language tasks and dense spatial/spatio-temporal prediction tasks. Was this direction considered, and what are the trade-offs or challenges in doing so?

**Ethical Concerns:**

["NO or VERY MINOR ethics concerns only"]

**Final Justification:**

I agree with reviewer pctU that the work has limited novelty from a scientific research perspective, as it does not introduce substantial new ideas or insights. Several limitations and flaws noted by all reviewers also reduce its scientific contribution. However, I also feel that developing such an encoder is a nontrivial effort (despite being largely engineering) and it is highly meaningful especially with the code, data, and model checkpoints open sourced. Considering these points, I agree with the other reviewers and recommend acceptance.

**Limitations:**

Yes.

**Paper Formatting Concerns:**

None.

**Quality:**

3

**Strengths And Weaknesses:**

Strengths:
1. Comprehensive, well-motivated, and large-scale experiments were conducted.
2. The analysis is dense and insightful.
3. The work makes a meaningful contribution by releasing powerful vision encoders, along with code and dataset, which could benefit a wide range of future research.

Weaknesses:
1. For downstream language modeling (e.g., MLLM development) and dense spatial tasks, separate perception encoder checkpoints are ultimately required.
2. There is no theoretical guarantee that the proposed language alignment and spatial alignment methods will ensure the most informative features are propagated from the middle to the final layer. In fact, Figure 5 shows that performance plateaus toward the final layers, and may decline beyond the 50-layer mark (however, we cannot confirm this because the authors experimented with a model that contains only 50 layers); Similarly, Figure 6 indicates that for some tasks, such as detection, the best-performing layer is not the final one.

---

> ### Author Rebuttal · Authors · 2025-07-30
>
> > [Q1] Line 277~282 mentions that the model training mix did not contain grounding data and it means that the significantly lifted grounding performance is entirely due to the strong intermediate grounding features now being aligned to the end of the network. However, the alignment method itself doesn’t intuitively or theoretically guarantee such an outcome. So what mechanism might explain this emergent grounding capability?
>
> Thank you for the great question! While PE core’s grounding performance peaks at layer 38 and then trends downward, PE lang’s performance plateaus instead (with the last layer performing similarly to layer 38). That is why we suspect the alignment “brings out” those features, even though there is no data in the training mix for it. As for the mechanism, we have additional analysis in Appendix C.8.1 that shows the last ~18 layers of the PE core G model form a sort of “decoder” to produce the CLIP token. This is close to the optimal layer for grounding (38), so it’s possible that the reason PE core drops in grounding performance is because this “CLIP decoder” is not suited for grounding. Since MLLM alignment naturally replaces the “CLIP decoder” with an “MLLM decoder”, the relevant features may now be passed through. Of course, this is just a hypothesis based on the current evidence. More work would need to be done to show this concretely.
>
>
>
>
>
> ---
> > [W2] There is no theoretical guarantee that the proposed language alignment and spatial alignment methods will ensure the most informative features are propagated from the middle to the final layer. In fact, Figure 5 shows that performance plateaus toward the final layers, and may decline beyond the 50-layer mark (however, we cannot confirm this because the authors experimented with a model that contains only 50 layers); Similarly, Figure 6 indicates that for some tasks, such as detection, the best-performing layer is not the final one.
>
> Absolutely, this is true for spatial alignment since it’s a proxy task. Like you said, there’s no guarantee that training on this proxy task will align the performance of every downstream spatial task to the last layer, given each uses different and sometimes specialized decoders. However, language alignment is more concrete. Since all MLLM tasks share the same decoder (and LLM), it’s more likely that a single layer can be the best for all tasks. This would be a given if the decoder used at test time was the same as the decoder we aligned with, since the loss is directly minimized for that specific layer with that specific decoder. Given that we align with Llama 3.2 3B and test with Llama 3.1 8B, this is not guaranteed. But empirically, it seems to be the case. And we even see evidence that this alignment can transfer to entirely different LLM families, for instance to QwenLM 2.5 7B (see Appendix C.7 / Table 29).
>
>
>
>
>
>
> ---
> > [Q2] Line 257~259 concludes that “a well-tuned large-scale contrastive model can learn general embeddings in the process of fitting its objective, but it fails to output them”. However, based on the results in Figure 4, this limitation seems to also apply to models like AIMv2 and DINOv2, which are not purely contrastive. Is there experimental evidence that this phenomenon is specific to contrastive models? Or does this insight actually more broadly apply to other well-tuned pretrained models regardless of their training objective?
>
> Thank you, this is a great point and is important to clarify! We make two separate claims about that in our work:
> 1. Models can have features that are relevant to a given task but not output them.
> 2. _Our model_ (contrastively pretrained) shows very strong general features, on par with the best models in each domain.
>
> For (1), you are absolutely correct: AIMv2 shows exactly the same phenomenon for tracking and DINOv2 for OCR and VQA. This is something that has already been widely shown (though, not to the extreme extent of the middle layers being the best): e.g., InternViT uses the 4th to last layer, DINOv2 itself evaluates with intermediate layers. However, the crux of our work is (2), that, when trained in the right way, a fairly vanilla CLIP model has intermediate features on par with the best pretrained models whose losses specialize for that task (AIMv2 for MLLM, DINOv2 for spatial). We more explicitly ablate this phenomenon in Appendix C.5, where we show these general features were a direct result of our robust pretraining recipe in Section 2.1 and that this general performance _scales_. To our knowledge, this is the first time this has ever been shown in a CLIP model.
>
>
>
>
> ---
> > [W1] For downstream language modeling (e.g., MLLM development) and dense spatial tasks, separate perception encoder checkpoints are ultimately required.
> >
> > [Q3] Why not design a Perception Encoder checkpoint that integrates both language alignment and spatial alignment? This could enable a single checkpoint to support both downstream multimodal language tasks and dense spatial/spatio-temporal prediction tasks. Was this direction considered, and what are the trade-offs or challenges in doing so?
>
> Yes, this is something we considered! Unfortunately, getting the performance of the model aligned to the end for all tasks and all decoders is very challenging (as you pointed out earlier with the spatial tasks not being fully aligned). We chose the separate checkpoints direction simply because we found this to be the best course of action in the near term, given our layer and alignment analysis. In fact, concurrently with this work, SigLIP2 attempted to support all the MLLM and dense prediction tasks with a single model by mixing together losses during pretraining. However, as we show in Table 6, this only seemed to improve alignment by a few layers v.s. SigLIP1 if at all. It seems like this is an area that may take the research community multiple iterations to get right. We hope that our analysis and layer-wise evaluation techniques can help accelerate that process.
>
>
>
>
>
> ---
> Finally, we would like to thank the reviewer for their thorough comments and questions. We also thank the reviewer for their kind words that we conduct _“comprehensive, well-motivated, and large-scale experiments”_, have _”dense and insightful”_ analysis, and make _“a meaningful contribution by releasing powerful vision encoders, along with code and dataset, which could benefit a wide range of future research”_. Please let us know if anything is unclear or needs further elaboration, and we will try to answer.

---

### Official Review · Reviewer_pctU · 2025-07-05

**Clarity:** 4
**Significance:** 2
**Originality:** 1
**Rating:** 3
**Confidence:** 4

**Summary:**

The paper presents Perception Encoder (PE), a family of large-scale vision encoders, trained with vision-language contrastive objective. It first introduces an image-only pretraining recipe combining progressive resolution, large-batch training, LAMB optimizer, and RoPE encoding, significantly improving over OpenCLIP. The proposed PEcore, a single encoder, could achieve state-of-the-art performance on image and video benchmarks. An additional probing analysis reveals that the best features lie in intermediate layers of encoder. PElang and PEspatial also could achieve competitive results on VQA/OCR tasks and dense prediction tasks.

**Questions:**

This paper does not have new technical contribution, so I do not have specific technical questions at this time. However, the authors are recommended to address the key weaknesses identified above. In particular:
1. How does the proposed approach fundamentally differ from existing open-source models such as OpenCLIP and ViCLIP? A clear comparison highlighting any unique design choices or training insights would help clarify the paper’s novelty.
2. Are there specific empirical findings, insights, or general principles uncovered through large-scale experiments that advance the field beyond what existing models provide?
3. The observation that intermediate layers yield the best embeddings is not new. Similar findings have been reported in recent works like "Scaling 4D Representations". Without deeper analysis, this cannot be considered a main contribution.

**Ethical Concerns:**

["NO or VERY MINOR ethics concerns only"]

**Final Justification:**

Despite more clarifications provided by the authors in the rebuttal, some of my concerns remain. The claim of reproducibility is not fully substantiated without open-sourcing the entire pipeline and training dataset. The interpretability, as emphasized in the rebuttal, leans too heavily on empirical observations. Finally, the limited scale of the released training dataset undermines the ability of others to reproduce the results. Overall, I remain inclined to reject this manuscript.

**Limitations:**

yes

**Quality:**

3

**Strengths And Weaknesses:**

Strengths:
1. The manuscript is well-organised and extensive experiments cover zero-shot image classification, zero-shot video classification, frozen linear probing, VQA and dense prediction tasks.
2. The empirical finding that the “best” embeddings live in intermediate layers.

Weaknesses
1. This work lacks novelty. It does not propose a new model or training objective, and instead builds on established practices such as large-batch CLIP training, progressive resolution, and LLM-based alignment. The core contribution lies in engineering scale and careful recipe design rather than introducing new insightful ideas.
2. The approach shows little differentiation from existing methods like OpenAI’s CLIP. The community already has access to strong open-source models such as OpenCLIP for images and ViCLIP (from InternVideo2) for video, many of which cover similar ground in terms of architecture and training strategies.
3. While the technical execution is impressive and the results are strong, the paper reads more as a report. If the authors plan to release models and datasets, it could be valuable to the community. However, strong empirical results enabled by massive compute and data are not, on their own, sufficient grounds for acceptance at a venue like NeurIPS. Accepting such a paper without clear conceptual and technical contributions would be unfair to submissions that introduce novel methods or insights.

---

> ### Author Rebuttal · Authors · 2025-07-30
>
> > [W1] This work lacks novelty. It does not propose a new model or training objective, and instead builds on established practices such as large-batch CLIP training, progressive resolution, and LLM-based alignment. The core contribution lies in engineering scale and careful recipe design rather than introducing new insightful ideas.
> >
> > [Q2] Are there specific empirical findings, insights, or general principles uncovered through large-scale experiments that advance the field beyond what existing models provide?
>
> We believe that the scope of novelty is broad and covers more than just new algorithms. It can also include extending existing paradigms to new problems, designing new experiments/ablations, drawing new observations/insights, and verifying expected hypotheses in new problems or tasks. From all reviewers’ comments, we have seen that many of these values of our work are recognized. We thank the reviewers for seeing value in our work from that perspective.
>
> It is true that our work makes use of existing techniques. Yet, several high profile works such as GPT3 (NeurIPS 2020 Outstanding Paper Award), ViT (ICLR 2021), and CLIP (ICML 2021) used primarily existing techniques at the time, but offered significant technical and engineering novelty. We believe this work falls in the same category, with many technical and engineering contributions:
>
> 1. PE is the first work to jointly obtain state-of-the-art image and video zero-shot classification and retrieval _with the same model_, primarily enabled by:
>    - A thoroughly ablated robust pretraining recipe (Section 2.1). While many of these techniques are used in other works, they are often just stated as being used without study (e.g., InternViT, EVA-CLIP, SigLIP2). The knowledge of which components are important at scale and their effect on robustness is a significant benefit to the community.
>    - A new video data engine, whose synthetic outputs lead to up to 10+% absolute gains on video benchmarks. Moreover, we will release the data we collected to construct it to the community (PE Video Dataset, see L128-129, Appendix A.1).
>    - A novel CLIP model distillation approach to produce very strong smaller models (L179-182, ablations in Appendix C.3).
>
> 2. As described by reviewer AWet, PE _“demonstrates that a single, simple contrastive pretraining can yield intermediate features competitive with specialized captioning and self-supervised models”_. We expand on this in our response to [Q3] below, but this has not been shown by any CLIP-pretrained model until now, and we show this is a _direct result_ of our robust pretraining recipe in Section 2.1 with ablations in Appendix C.5.  The community has been searching for a single vision pretraining task with good performance on all downstream tasks for a while now, so we believe this finding will be particularly useful.
>
> 3. We fully agree that LLM-based alignment is not new; however, our technique for spatial alignment is _entirely novel_. In fact, not only do we introduce a new loss for this alignment (described in L302-309, Eq. 3 in Appendix B.3.1), but we also use the SAM model for distillation in an _entirely new way_ (SAM mask logits, L306-307 and Figure 19). We expect this to provide significant benefit to future work, especially given existing work has found it hard to distill from a SAM model (as mentioned in L1460-1461). Regardless of the perspective on novelty, this at least qualifies as “a new training objective”, and we've demonstrated its effectiveness across several established benchmarks.
>
> We hope that this helps to clarify our view on the novelty of our work.
>
>
> ---
> > [W2] The approach shows little differentiation from existing methods like OpenAI’s CLIP. The community already has access to strong open-source models such as OpenCLIP for images and ViCLIP (from InternVideo2) for video, many of which cover similar ground in terms of architecture and training strategies.
> >
> > [Q1] How does the proposed approach fundamentally differ from existing open-source models such as OpenCLIP and ViCLIP? A clear comparison highlighting any unique design choices or training insights would help clarify the paper’s novelty.
>
>
> Correct, PE follows in the CLIP line of work, meaning it will naturally share similarities with CLIP-based vision encoders. The main differentiating factors are:
> 1. **Performance**: PE is the most powerful CLIP model to date, for both images and video. PE significantly outperforms other open models like OpenCLIP. The biggest OpenCLIP model (OpenCLIP ViT-bigG) obtains 80.1% IN-1k top-1, while PE core G obtains 85.4% (as well as significantly higher robustness metrics). ViCLIP 6B (InternVideo2) obtains 64.2% Kinetics 700 top-1, while PE core G obtains 69.1%. In both cases, a 5% performance improvement represents a substantial benefit to the community.
>    - Also note the starting point of our ablations (row 1 in Figure 2) _is_ OpenCLIP’s vanilla model trained on our data. Thus, PE’s strong performance is explained by our ablations, not hidden training tricks.
> 2. **Joint Modeling**: PE obtains this strong performance on both image and video _with the same checkpoint_. Not only is this more convenient than using different models for images and videos, it also enables new capabilities that using OpenCLIP / ViCLIP today in tandem couldn’t offer. For instance, image→video or video→image retrieval.
> 3. **General Features**: As shown in Section 3, PE core has strong, general features in its intermediate layers that can match or exceed the performance of encoders specialized to those domains (AIMv2 for MLLM, DINOv2 for spatial). This has not been shown before—in fact, CLIP models are notorious for being bad at spatial tasks. And as mentioned above this is a _direct result_ of robust pretraining strategy in Section 2.1, as shown in Appendix C.5.
>
>
>
>
>
> ---
> > [Q3] The observation that intermediate layers yield the best embeddings is not new. Similar findings have been reported in recent works like "Scaling 4D Representations". Without deeper analysis, this cannot be considered a main contribution.
>
> Thank you, this is a good point, and we would like to clarify that there are two aspects in our claim that “a well-tuned large-scale contrastive model can learn general embeddings in the process of fitting its objective, but it fails to output them”:
>
> 1. Models can have intermediate features that are relevant to a given task but not output them.
> 2. _Our model_ (contrastively pretrained) shows very strong general features, on par with the best models in each domain.
>
> The first part (1), as you point out, is not new. In fact, we dedicate a section of the related work (L372-383) showing other works that share the same finding. What we add to the conversation is controlled, large-scale layer-wise ablations of large models across a diverse set of tasks. In this respect, we show surprising results that all models we test (DINOv2-g, AIMv2 3B, PE core G) have optimal layers for some tasks as deep as the very middle of the network (tracking for AIMv2, OCR for DINOv2), and the difference can be extreme; for example AIMv2’s middle layer has 54.7 J&F on DAVIS vs. 29.3 for the last layer.
>
> However, the main crux of our work is (2): that, when trained in the right way, a fairly vanilla CLIP model has intermediate features on par with the best pretrained models whose losses specialize for that task (AIMv2 for MLLM, DINOv2 for spatial tasks). We explicitly ablate this phenomenon in Appendix C.5, where we show these general features were a direct result of our robust pretraining recipe in Section 2.1 and that this general performance _scales_. As mentioned before, this is the first time this has ever been shown in a CLIP model.
>
> This is an important point to clarify, and so we will update the introduction in the final draft to make this point clear. Thank you.
>
>
> ---
> > [W3] While the technical execution is impressive and the results are strong, the paper reads more as a report. If the authors plan to release models and datasets, it could be valuable to the community. However, strong empirical results enabled by massive compute and data are not, on their own, sufficient grounds for acceptance at a venue like NeurIPS. Accepting such a paper without clear conceptual and technical contributions would be unfair to submissions that introduce novel methods or insights.
>
> We will be releasing both the models, code, and PE video dataset. We are glad that the reviewer agrees that this work can be valuable to the community. We are happy this appears to be a consensus among the reviewers: _“open-source of this work would bring significant benefits to the research community”_ (Reviewer jvg2), _“could benefit a wide range of future research”_ (Reviewer KE9i), _“convincingly establish new SOTA in multiple categories”_ (Reviewer AWet), _“the paper is already top notch”_ (Reviewer KER3). We hope the technical contributions and novel methods listed above are convincing enough to reconsider meeting the bar of NeurIPS acceptance.
>
> ---
> We thank the reviewer for the constructive feedback and hope our response can inspire a reconsideration of the score. Please let us know if anything is unclear or needs further discussion.

---

> ### Comment · Reviewer_pctU · 2025-08-06
> **Comment**
>
> Thanks the authors for their response and clarifications. However, many concerns remain:
>
>
> 1. **Novelty and Contributions:**
>
> I acknowledge the broad definition of novelty and agree that contributions can take various forms, including insightful empirical studies and novel problem formulations. Nevertheless, not all works using existing techniques merit acceptance. Drawing parallels to GPT-3, ViT, and CLIP, as done by the authors, is somewhat misleading. These works significantly impacted their fields as they introduced groundbreaking innovations: GPT-3 for large-scale language pre-training, ViT for shifting vision modeling from convolutional networks (ResNet) to plain Transformers, thereby uncovering scaling laws, and CLIP for pioneering a bridge between vision and language. In contrast, CLIP was published **four years ago**, and numerous follow-up studies (e.g., OpenCLIP, InternVideo  v1 & v2, EVA v1 & v2, SigLIP v1 & v2) have extensively explored similar scaling directions. Therefore, the empirical observations provided here appear less impactful and innovative.
>
> 2. **Ablations:**
>
> I fully appreciate the extensive experiments and detailed ablation studies conducted. However, extensive experiments alone does not guarantee sufficient scientific merit for acceptance. Prior works might have only briefly mentioned similar technical practices (tricks) because these are not their key contributions. By extensively and highlighting these established practices, this paper appears more like a more detailed technical report or an extened journal version, rather than a distinct conceptual or methodological advancement. I do recognize the contribution in such detailed report, but do not see it as sufficient for clear acceptance.
>
> 3. **Video Dataset Contribution:**
>
> The commitment to releasing a new video dataset is potentially beneficial to the community. However, the authors have not clarified some important details such as the size and scale of **the dataset to be released**, and more comparison is needed to existing large-scale open datasets like LAION-5B (OpenCLIP) or InternVid (InternVideo). Without such specifics, the claimed contribution remains ambiguous and mariginal.
>
> Oeverall, while this submission demonstrates good performance on many benchmarks, I do not find sufficient conceptual novelty or impactful insights that deserve clear acceptance.

---

> > ### Author Response · Authors · 2025-08-07
> >
> > We thank the reviewer for their response and understanding that novelty can take various forms, including insightful empirical studies and novel problem formulation. Please note that our work is the first to _successfully combine the performance of image and video CLIP models into one model_ and the first to _produce features general enough to match specialized models on both language (AIMv2) and spatial tasks (DINOv2) with a single CLIP pretraining loss_. As reviewers noted, we comprehensively study the effectiveness of each of the design choices that led to these outcomes, both with extensive study in robust image pretraining and how to build a video data engine, as well as analyzing how PE’s general features came about. As for techniques, we also introduce an entirely novel spatial alignment method to learn spatial features in a general way for downstream tasks. These contributions, in addition to our extensive experimentation, are what the other works mentioned lack and clearly differentiates us from prior or concurrent efforts, including CLIP, OpenCLIP, InterVideo1 & 2, EVA1 & 2, and SigLIP1 & 2.
> >
> > We sincerely thank the reviewer for acknowledging the detailed ablation studies and experimental efforts in our work. While we understand the concern that such depth might resemble a detailed technical report, we respectfully believe that our analysis contributes more than just ablations. In contrast to prior works that often bundle multiple components together or briefly refer to implementation “tricks” without quantitative validation, our work provides reproducible and interpretable evidence that can guide future work. We would like to highlight that, in addition to architectural contributions, NeurIPS values reproducibility, completeness, and clarity, which are important elements of research and are not clearly targeted in prior works.
> >
> >
> > ---
> >
> > As for details about the video dataset we release, the full information and specifications are available in Appendix A.1. The statistics the reviewer is looking for are contained in Table 9. We have reproduced this table below for convenience:
> >
> > | Statistic | Value |
> > |---:|---|
> > | Videos | 998,862 |
> > | Human Captions | 118,862 |
> > | Total Duration | 4625 hrs |
> > | Avg Duration (s) | 16.7±9.8 |
> > | Avg Human Caption Length | 57.1±25.4 |
> > | Avg Model Caption Length | 111.7±43.2 |
> >
> > As described in Appendix A.1, all videos have accompanying metadata and descriptions. Additionally, please note that the 118,862 detailed video captions are all _*human-annotated*_, unlike LAION and InternVid. In section 2.2, we show that training our data engine using these captions and then generating 22M video-caption pairs for jointly training our PE model results in state-of-the-art performance on both image and video zero-shot and retrieval benchmarks using a single model. Incorporating the human annotated data on top of the existing open-source data used to train our captioning model improved downstream video CLIP performance by ~1% on average, as shown in Table 1. This makes it a significant component of the performance of our model, which is why we release it for the benefit of the community.

---

### Decision · Program_Chairs · 2025-09-17

**Decision:**

Accept (oral)

**Comment:**

This paper was reviewed by 5 experts in the field, and 4 of them are enthusiastic about the paper. Reviewer pctU gave borderline reject and is concerned about the technical contribution of the paper.

The AC reads the paper, the reviews, and the authors' responses carefully, especially focusing on the review from pctU and the discussions. The AC appreciates the reviewer's detailed comments and the follow-up discussions that help to clarify the contribution of the paper. The AC also agrees with the weakness pointed out by KE9i in W1. The AC understands the concerns from the reviewer pctU and agrees that the study in the paper may not be sufficient to initiate a paradigm shift in the computer vision community. However, the current paper made significant contributions to the community, providing strong vision encoders with models and data to be released, and great insights about how to better unify the vision encoder for diverse downstream vision tasks. The AC believes the paper made a solid step forward, and the findings in the paper should be presented to a broader audience for better visibility.